# Role of PI3K-AKT Pathway in Ultraviolet Ray and Hydrogen Peroxide-Induced Oxidative Damage and Its Repair by Grain Ferments

**DOI:** 10.3390/foods12040806

**Published:** 2023-02-13

**Authors:** Wenjing Cheng, Xiuqin Shi, Jiachan Zhang, Luyao Li, Feiqian Di, Meng Li, Changtao Wang, Quan An, Dan Zhao

**Affiliations:** 1Beijing Key Laboratory of Plant Resource Research and Development, College of Chemistry and Materials Engineering, Beijing Technology and Business University, Beijing 100048, China; 2Institute of Cosmetic Regulatory Science, Beijing Technology and Business University, Beijing 100048, China; 3Yunnan Baiyao Group Co., Ltd., Kunming 650000, China

**Keywords:** antioxidant, RNA-seq, grain fermentation, PI3K-Akt pathway

## Abstract

UV and external environmental stimuli can cause oxidative damage to skin cells. However, the molecular mechanisms involved in cell damage have not been systematically and clearly elucidated. In our study, an RNA-seq technique was used to determine the differentially expressed genes (DEGs) of the UVA/H_2_O_2_-induced model. Gene Oncology (GO) clustering and the Kyoto Encyclopedia of Genes and Genomes (KEGG) Pathway analysis were performed to determine the core DEGs and key signaling pathway. The PI3K-AKT signaling pathway was selected as playing a part in the oxidative process and was verified by reverse transcription-quantitative polymerase chain reaction (RT-qPCR). We selected three kinds of *Schizophyllum commune* fermented actives to evaluate whether the PI3K-AKT signaling pathway also plays a role in the resistance of active substances to oxidative damage. Results indicated that DEGs were mainly enriched in five categories: external stimulus response, oxidative stress, immunity, inflammation, and skin barrier regulation. *S. commune*-grain ferments can effectively reduce cellular oxidative damage through the PI3K-AKT pathway at both the cellular and molecular levels. Some typical mRNAs (COL1A1, COL1A2, COL4A5, FN1, IGF2, NR4A1, and PIK3R1) were detected, and the results obtained were consistent with those of RNA-seq. These results may give us a common set of standards or criteria for the screen of anti-oxidative actives in the future.

## 1. Introduction

Oxidative damage to the skin can lead to skin aging, which has always been a focus of attention. Excessive reactive oxygen species (ROS) are considered to be the main factor that induces cell damage and apoptosis [1]. A large amount of research has been conducted to identify compounds that can protect against oxidative stress damage. These compounds can be derived from plants, animals, and microorganisms, as well as chemical compounds [2,3,4,5]. Cell model-based screening and mechanism exploration are common research approaches. In these studies, a large proportion of studies are based on UVA induction [6,7,8] and hydrogen peroxide induction [2,6,9,10].

As for research regarding dermatology, cosmetics, and functional foods, human skin fibroblast cells (HSFs) oxidative stress damaged models that are stimulated by either UVA or H_2_O_2_ are widely used [6]. Scholars have some common knowledge both of these models, including that they can break the balance between oxidants and antioxidants and cause oxidative stress damage by increasing the oxides content [6,7,8,9,10]. There may be differences in the internal mechanism of these two models in terms of how they induce free radicals. In spite of this, we still generally believe that the effective ingredients obtained by screening have potential application value. Based on this, if certain criteria can be found that provide a method for raw material screening, future work will experience unprecedented convenience. However, there has been limited information on the similarities and differences in the molecular mechanisms between the models at the RNA or protein levels.

Transcriptomics is the field of studies that evaluates gene expression at the RNA level and is used to study gene transcription and regulation in cells. RNA-seq can be used for to construct a new and complete library [11,12], and its results show repeatability and have low cost [13,14]. The screening of drug targets or the establishment of disease modules using RNA-seq have been used widely in pharmacological research, such as those exerted to analyze the gene pathways involved in the anti-breast cancer effect [15].

In our study, RNA-seq technology was used to analyze the in-depth mechanisms involved in these two oxidative damage models to identify common changes. The core differentially expressed genes (DEGs) were chosen and verified by reverse transcription-quantitative polymerase chain reaction (RT-qPCR) for consistency between high-throughput analysis and RT-qPCR results. Furthermore, several actives were selected to determine if these findings held true.

*Schizophyllum commune* (*S. commune*) is a kind of fungi that exerts excellent antioxidant [16], anti-tumor [17], and anti-aging properties [18]. The *β*-glucan rich schizophyllum can effectively modify the intestinal microbiota, regulate intestinal metabolic disorders, and relieve diseases such as obesity, diabetes, and inflammatory bowel disease [19]. Our previous laboratory study has indicated the anti-oxidative effects of the polysaccharides from *S. commune* (typical data shown in Appendix A). In addition, the most commonly consumed grains, such as rice, highland barley, and oats, may potentially exert skin care and health benefits, such as relieving oxidative stress, scavenging free radicals, and exerting anti-inflammatory capabilities [20,21]. High levels of phenols, including proanthocyanins, flavanols, and their derivatives, are found in rice, highland barley, and oats [20,22,23,24]. Studies have confirmed that most of the phenols contained in highland barley are in bound forms, which have strong antioxidant activities and can also inhibit the proliferation of human hepatoma cells [20]. Additionally, the phenol extraction rates from highland barley or oats using enzymatic or fermented hydrolysis are higher than that of chemical extraction. The extracts also show stronger antioxidant activities [22,25]. The above studies have confirmed that mild extraction methods, such as enzymatic hydrolysis and fermentation, can effectively improve the antioxidant activity of grains. Based on these results, a previous study produced three kinds of *S. commune* fermented broths, including rice, highland barley, and oats [16,26,27] to verify whether they followed the RNA-seq findings.

In doing so, we hope our study will harmonize these distinct methods, and facilitate future experiments and analyses.

## 2. Materials and Methods

### 2.1. Materials

Human skin fibroblasts (HSFs) were purchased from Cell Resource Center, Institute of Basic Medicine, Chinese Academy of Medical Sciences; Dulbecco’s Modified Eagle Medium (DMEM), Fibroblast Medium (FM), fetal bovine serum (FBS), phosphate buffer saline (PBS), and 0.25% trypsin-EDTA and penicillin(1 × 10^5^ U/L)-streptomycin (100 mg/L) were all purchased from GIBCO Life Technologies (Carlsbad, CA, USA); EasyScript^®^ One-Step gDNA Removal and cDNA Synthesis SuperMix reverse transcription kit, TransStart^®^ Top Green qPCR SuperMix kit were obtained from Beijing TransGen Biotech Co., Ltd. (Beijing, China), and Cell Counting Kit-8 (CCK-8) was purchased from Biorigin (Beijing) Inc. (Beijing, China).

### 2.2. Preparation of S. commune Fermented Broths

The rice fermentation broth (RFB), highland barley fermentation broth (HBFB), and oats fermentation broth (OFB) were collected, sterilized, and freeze-dried to be made into powders following the parameters in the literature [16,26,27] with slight modifications. The *S. commune* strain was initially cultured in the seed medium as fermented seed at 28 °C for 3 days. The resulting seed culture was inoculated with a 3% inoculum size (*v*/*v*) in the fermentation medium and cultivation was carried out at 160 rpm and 28 °C for 3 days. The rice/highland barley/oats-supplemented mediums contain 1% rice/highland barley/oats powder (*m*/*v*), respectively.

### 2.3. Cell Culture and the Cell Viability Measurement

The cells were cultured in DMEM supplemented with 10% FBS, 1% fibroblast growth additives, and 1% penicillin-streptomycin. Then, the cells were incubated (Shanghai Shengke Instrument Equipment Co., Ltd., Shanghai, China) in a 5% CO_2_ humidified atmosphere at 37 °C. The culture medium was changed when cell confluence reached 80%.

Cells viability was assayed using the colorimetric CCK-8 method [28,29,30] by following the manufacturer’s instructions. Then, the cells were seeded at a concentration of 1 × 10^5^ cells/mL into a 96-well plate for 12 h and were incubated with various concentrations of RFB (0 to 5 mg/mL), HBFB (0 to 2.5 mg/mL), and OFB (0 to 1.25 mg/mL) for 24 h. Next, the cells were incubated for 4 h by adding 10 μL of CCK8 solution (Biorigin, Beijing, Inc.) into each well. The optical densities of the solutions were quantified at a wavelength of 450 nm using a microplate reader (Thermo Fisher Scientific, Waltham, MA, USA).

### 2.4. UVA/H_2_O_2_-Induced Model Establishment

IC_50_ parameters (50% cell viability) of UVA/H_2_O_2_ were chosen to establish the model. Cell viability was evaluated using the Cell Counting Kit-8 (CCK-8) method.

The UVA-induced model was established under the condition of 25 J/cm^2^. The detailed procedure was as follows: HSF cells were seeded into a 6-well plate (1 × 10^6^ per well) and cultured for 12 h. Then, one milliliter of PBS (pH 7.4, 0.01 M) was added in each well, and the cells were irradiated at a dose of 25 J/cm^2^. UVA (365 nm) irradiation was created using an UVA lamp, and its intensity was measured using a UV power meter (LS125, Shenzhen Shanglin Technology Co., Ltd., Shenzhen, China).

The H_2_O_2_-induced model was established by treating H_2_O_2_ at a concentration of 2000 µmol/L for 30 min. The detailed procedures were as follows: HSF cells were seeded into a 6-well plate (1 × 10^6^ well) and cultured for 12 h. Two milliliters of H_2_O_2_ at the concentration of 2000 µmol/L was added into each well, and the plate was incubated for 30 min.

The control group was treated without UVA/H_2_O_2_, and the cell viability under different conditions were simultaneously measured.

### 2.5. Transcriptome Sequencing

The cells were collected after the different treatments and total RNAs were extracted using the TriQuick Reagent (Beijing Solarbio Science & Technology Co., Ltd., Beijing, China). RNA quality was determined using an Agilent 2100 Bioanalyser (Agilent Technologies, Santa Clara, CA, USA) and quantified using a NanoDrop 2000 system (Thermo Scientific™ Technologies, Wilmington, DE, USA). Only high-quality RNA samples (OD260/280 = 1.8~2.2, OD260/230 ≥ 2.0, RIN ≥ 6.5, 28S:18S ≥ 1.0, > 2 μg) were used to construct the sequencing library. Transcriptome sequencing was performed by Shanghai Majorbio Bio-pharm Technology Co. Ltd. Gene read counts were obtained for each group, and high-quality reads were retained to perform the differential expression analysis of the genes. The *p*-value and fold change (FC) in each gene was calculated to determine the differences in expression between the libraries. In addition, we determined and summarized read frequencies and differences in sequence generation, trimming, and the mapping results of the cell samples before and after UVA and H_2_O_2_ treatments. In the UVA-induced model, approximately 20.6 and 20.3 million raw reads were obtained in the control and model groups, respectively. After removing low-quality reads, 20.4 and 20 million high-quality reads remained in the control and model groups, respectively (>98.5% for both), and were assembled for downstream analysis. In the H_2_O_2_-indued model, approximately 50.5 and 50.46 million raw reads were obtained from the control and model groups, respectively. After removing low-quality reads, 50 and 49.92 million high-quality reads remained in the control and model groups, respectively (> 98.9% for both), and were assembled for downstream analysis. The clean reads were mapped to the reference genome (Genome assembly: GRCh38.p10, http://www.ensembl.org/Homo_sapiens/Info/Index, accessed on 1 February 2023) and the detailed mapping output is summarized in Table 1.

### 2.6. Screening for Differential Expressed Genes (DEGs) and Functional Analysis

Differential expression analysis was performed between control group and model group to identify the DEGs and their functions.

The DEGs between the control and UVA-induced model were screened using |log_2_FC| > 1.2 and *P*-adjust < 0.05 as the thresholds. The DEGs in the H_2_O_2_-induced oxidative damage study were screened using the criteria, |log_2_FC| > 2 and *P*-adjust < 0.05. The DEGs were functionally annotated using Blast2go (Version 2.5) and goatools (Version 0.6.5) for Gene Ontology (GO) annotation and clustering analysis. The clustered GO terms involving biological process (BP), molecular function (MF), and cellular component (CC) were obtained based on the criteria of the enrichment score of > 2. The significantly enriched pathways were identified using the Kyoto Encyclopedia of Genes and Genomes (KEGG) database with *p* < 0.05 as the criterion. The website Database for Annotation, Visualization and Integrated Discovery (DAVID ver 6.8, http://david.ncifcrf.gov/, accessed on 1 February 2023) was also used.

### 2.7. RT-qPCR

RT-qPCR was used to investigate the expression levels of the PI3K-AKT pathway associated genes (COL1A1, COL1A2, COL4A5, FN1, IGF2, and PIK3R1).

Total RNA extraction and cDNA synthesis were performed using the TriQuick Reagent (TRIzol Substitute) (Beijing Solarbio Science & Technology Co., Ltd.) and the UEIris II RT-PCR System for the cDNA Synthesis SuperMix (Beijing TransGen Biotech Co., Ltd.) according to the manufacturers’ instructions. All synthesized cDNA were used for the PCR amplification using the TransStart^®^ Top Green qPCR SuperMix kit (Beijing TransGen Biotech Co., Ltd.). The total reaction system was 20 μL. The primer sequences used were presented in Table 2 [6]. GAPDH was used as a normalization control.

The cyclic parameters used were pre-denaturation at 94 °C for 30 s, followed by the PCR reaction (45 cycles of 94 °C for 15 s, 60 °C for 15 s, and 72 °C for 10 s), while fluorescence data were collected at 72 °C. The relative gene expression levels were analyzed using the 2^−ΔΔCT^ method [31].

### 2.8. Statistics

All experiments were performed in triplicate at least, and the data were expressed as mean ± standard deviation (SD). The data were analyzed using SPSS 17.0 (SPSS, Armonk, New York, NY, USA) and GraphPad Prism 9.0 (GraphPad Software, San Diego, CA, USA). One-way analysis of variance was used for the significance of differences between the groups (#, * *p* < 0.05; ##, ** *p* < 0.01).

## 3. Results

### 3.1. The Establishment of UVA/H_2_O_2_-Induced Models and Identification of DEGs

Special criteria were used to screen for the differences between the DEGs in the model and control in each study. For the UVA-induced oxidative damage study, we collected and filtered out a total of 552 DEGs based on the criteria of |log_2_FC| > 1.2 and P-adjust < 0.05, of which 249 genes were upregulated and 304 genes were downregulated. As shown in Table 3, among the upregulated DEGs, 144 genes were involved in cell proliferation and regulation, with 22 genes encoding proteins that can produce a response to an external stimulus. A total of 15 genes participated in immune and inflammatory response, 9 genes were involved in skin barrier-related responses. In addition, 7 of the genes that encoded proteins were also found to play a role in the oxidative stress response, while the functions of the other 36 genes were unknown. Among the downregulated DEGs, 166 genes were involved in cell metabolism, proliferation, and regulation, while 24 genes participated in intercellular signaling. A total of 16 genes encoded proteins that could play a role in the oxidative stress response, and 11 genes were involved in inflammatory processes. Six genes were involved in the synthesis of the cytoskeleton. Apart from these genes, the functions of eight DEGs were uncharacterized.

UVA/H_2_O_2_ can induce ROS generation, leading to a decrease in cell viability, which lead to a decline in cellular fitness and ultimately, to cell death. We determined the effects of the inducers (UVA/H_2_O_2_) on cell viabilities (Figure 1A,B). Dose dependency was observed in both studies. IC_50_ parameters were chosen to establish damage models. UVA irradiation at a dose of 15 J/cm^2^ was finally selected to be used to establish the UVA-induced oxidative stress damage model for subsequent studies. H_2_O_2_ at a concentration of 1000 µmol/L H_2_O_2_ for 30 min was used as the modeling condition of the H_2_O_2_-induced oxidative stress damage model (Figure 1).

Likewise, as shown in Table 4, we compared the H_2_O_2_-injured Model with the Control, and a total of 607 DEGs were screened based on the criteria, |log_2_FC| > 2 and P-adjusted < 0.05. Among them, 226 DEGs were upregulated and 381 DEGs were downregulated. Among the upregulated DEGs, 19 genes encoded proteins that regulated the oxidative stress response, 10 genes regulated interactions between cells and the surrounding environment, and 6 genes were involved in skin barrier function and lipid anabolism. In addition, there were 42 genes whose functions were unknown. The downregulated DEGs were significantly involved in pathways, cellular matrix composition entry and complement coagulation cascade. Among them, a total of 139 genes encoded proteins that participated in cell proliferation and regulation, 27 genes were responsible for regulating the immune response process and immune system, 24 genes were involved in the positive regulation process of cell chemotaxis and inflammatory response, while 9 genes were responsible for regulating the cell response. In addition, there were 54 genes whose functions were unknown.

The above-mentioned sequencing analysis indicated that some similar functions of DEGs are grouped. There were no common genes between UVA-upregulated and H_2_O_2_-upregulated DEGs (Venn diagram shown in Appendix A), suggesting that different insight mechanisms exist. DAVID analysis showed that UVA-induced upregulated DEGs mainly focused on the regulation of transcription (enrichment score = 3.47) and protein phosphorylation (enrichment score = 1.35), which both had roles in signal transduction. While the H_2_O_2_-induced upregulated DEGs mainly participated in the processes related to the immune response (enrichment score = 3.11), inflammatory responses (enrichment score = 3.09), cell adhesion (enrichment score = 1.93) and regulation of transcription (Enrichment score = 1.11) (data not shown). While 15 common DEGs existed both in UVA and H_2_O_2_ induced downregulated DEGs, 5 existed in the extracellular exosome and participated in cell migration (Data not shown).

### 3.2. Gene Ontology (GO) and KEGG Pathway Analysis

In the UVA induced damage model, the results of the GO enrichment analysis showed that a total of 14 GO terms of the upregulated (Figure 1a) and downregulated (Figure 1b) DEGs were enriched. Two GO terms were enriched in the upregulated DEGs, namely cyclin-dependent protein kinase activity and cyclin-dependent protein serine/threonine kinase activity. On the contrary, a total of 12 categories were enriched in the downregulated DEGs. Among them, DNA-binding transcription activator activity (RNA), negative regulation of phosphorylation, and collagen-containing extracellular matrix were the top 3 GO terms with high gene ratios and significant P-adjust values, which indicated that the processes of cell proliferation, cell communication, and extracellular matrix production were inhibited. Furthermore, extracellular matrix components and their regulated genes were decreased, indicating changes in cellular morphology. At the same time, biological signal transduction induced by phosphorylation were slowed down or even inhibited.

In the H_2_O_2_ oxidative damage model, there were 10 GO terms enriched in upregulated DEGs (Figure 1c) and 12 GO terms enriched in downregulated DEGs (Figure 1d). Among these enriched terms of upregulated DEGs, receptor ligand activity, extracellular matrix organization, and response to molecule of bacterial origin, were the top three terms with high gene ratios and significant P-adjusted values. The terms related to extracellular matrix constituents or collagen production were both found in the UVA and H_2_O_2_-induced damage models, indicating the damaged changes in cellular morphology.

KEGG enrichment analysis was performed. As shown in Figure 2, the UVA-induced upregulated DEGs were mostly enriched in virus infection (herpes simplex virus 1 infection), while the downregulated DEGs were significantly enriched in the TNF signaling pathway and IL-17 signaling pathway (Figure 2A,B). It is worth noting that TNF and IL-17 can activate or regulate the PI3K-AKT signaling pathway, indicating that UVA may damage cells through that pathway. A comprehensive KEGG enrichment analysis was also conducted, as shown in Figure 2C. All the DEGs that were both upregulated and downregulated were enriched in pathway terms, such as ECM-receptor interaction and the PI3K-AKT signaling pathway.

The KEGG enrichment analysis results of the H_2_O_2_-induced damage model (Figure 2D,E) showed that the upregulated DEGs were significantly enriched in cytokine–cytokine receptor interaction and the IL-17 signaling pathway, while the downregulated DEGs were enriched in axon guidance and complement and coagulation cascades. The H_2_O_2_-induced and UVA-induced DEGs were both involved in several common terms regardless of the method of regulation. These terms were ECM-receptor interaction, PI3K-AKT signaling pathway, focal adhesion, and TNF signaling pathway. In addition, the H_2_O_2_-induced DEGs were also involved in the other pathway terms, such as complement and coagulation cascades and cytokine–cytokine receptor interaction, which are associated with immune response (Figure 2F).

### 3.3. Key DEGs Functioned in PI3K-AKT Pathway

According to Figure 2C,F, several common KEGG terms were found to be enriched in both UVA and H_2_O_2_ induced cell damage. The pathways involved in the occurrence, development, and response of inflammation were both enriched in these two models. TNF can combine with tumor necrosis factor receptor 1 (TNFR1) to form complex 1, which promotes the ubiquitination of E3, the ubiquitin-connected protein inhibitor of apoptosis (cIAP), which in turn activates the NF-κB signaling pathway, leading to keratinocyte inflammatory response and even keratinocyte inflammatory response. NF-κB inhibition of the keratinocytes causes RIPK1-mediated necroptosis and skin inflammation. In addition, the extracellular matrix (ECM)-related gene expression process has also been identified as an important factor that affects the level of cellular inflammation. Based on gene counts and P-value into account, we found that many DEGs induced by UVA (70) or H_2_O_2_ (55) were involved in the PI3K-AKT pathway based on a relatively high value of -log10(P), as shown in Figure 2C,F.

Based on the findings presented here, the PI3K-AKT signaling pathway was the focus of our ongoing studies. As shown in Figure 3, the expression changes of the genes involved in the PI3K-AKT signaling pathway induced by UVA (Figure 3A) and H_2_O_2_ (Figure 3B) are clearly demonstrated.

In the UVA-induced model, most of the DEGs involved in PI3K-AKT pathway were downregulated (60 downregulated DEGs out of 70 DEGs) and mainly participated in three biological processes, such as cell adhesion, regulation of MAPK cascade, and cell proliferation and differentiation (Appendix A). The expression of receptor tyrosine (RTK)-associated protein on the membrane was downregulated, which was induced by the decrease of population stimulating factor (GF) in the extracellular matrix. RTK subsequently changed the expression of SOS and Ras, which finally induced the decrease of cell proliferation, angiogenesis, and DNA repair. Furthermore, phosphatidylinositol 3 kinase (PI3K) was also downregulated by Ras. Some cytokines and ECM in the extracellular matrix were also repressed in expression, causing a downregulation of PI3K. As previously stated, PI3K is a hub node that can influence how the expression of downstream proteins affects protein synthesis, glycolysis, apoptosis, and crosstalk with other signaling pathways. For example, the downregulation of CREB (cAMP-response element binding protein), and FOXO (Forkhead box O) in this study may affect cell survival and cell cycle progression. The levels of PKCs also decreased, affecting glucose uptake and vesicle transport. The accelerated DNA damaging-induced transcript (REDD1) and serine threonine kinase (LKB1) also resulted in an unbalancing of protein synthesis.

Figure 3B showed the DEGs involved in PI3K-AKT signaling pathway in the H_2_O_2_-induced oxidative damaged model, which is different from that of the UVA model. A total of 55 DEGs, including 28 upregulated DEGs and 27 downregulated DEGs, were enriched in the PI3K-AKT signaling pathway. Three GO terms were enriched (Appendix A). Cell adhesion, for example, showed a high enrichment score of 11.81, which played an important part in the inflammatory response.

In the H_2_O_2_-induced model, the downregulation of PI3K was mainly due to the downregulation of toll-like receptor (TLR2/4) and integrin (ITGB) on cell membrane and extracellular matrix (ECM), which ultimately affected cell survival, cell cycle, metabolism, and protein synthesis. Additionally, the reduced expression of Adenosine 5 ‘-Monophosphate (AMP)-activated protein kinase (AMPK) also influenced the synthesis of proteins. The expression level of the protein encoded by phosphatase type 2 phosphatase activator (PP2A) decreased, which induced an increase in the CREB and MDM2 prototype oncogene, indirectly. The increased expression of the MDM2 inhibited tumor suppressor p53, which played a positive role in cell survival. Furthermore, the downregulation of PI3K and PP2A also affected cell metabolism, indirectly. We also found that the transmembrane 4 L six family member 1 (TM4SF1), E2F transcription factor 7 (E2F7), and BCL2 related protein A1 (BCL2A1) showed a high level of expression.

Overall, although differences were found between these two oxidative damage models, there were certain common DEGs. A total of 11 DEGs with a similar trend were identified in the pathway: PIK3R1, IGF2, ITGA4, LAMA2, LAMA5, LAMB2, NR4A1, COL1A1, COL1A2, COL4A5, and FN1. Some of them were regulatory hub nodes, while others encoded for the cellular matrix. We finally chose 6 of them in the subsequent verification (COL1A1, COL1A2, COL4A5, FN1, IGF2, and PIK3R1) by qRT-PCR, and found the consistency with the RNA-seq results.

### 3.4. Composition Analysis and Protective Effects of RFB, HBFB and OFB

S. commune is an edible fungus with good nutritive value that is often used in food as a functional additive. Since the functional effects of S. commune-fermented grain additives have been developed and studied in the lab for many years, we selected three kinds of fermented actives in this study to verify whether they were suitable for the criteria obtained from the above-mentioned RNA-seq results.

The physicochemical properties of the RFB, HBFB, OFB are shown in detail in Table 5 and Table 6. The carbohydrate content of RFB, HBFB, and OFB was 83.9%, 73.3% and 59.2%, respectively. Among them, total sugar content of RFB was highest, about 43.36%, and the β-glucan content of OFB was the highest (2.74%). The reducing sugar was highest in RFB (3.18%).

The cell viabilities to HSFs of RFB (Figure 4A), HBFB (Figure 4B), and OFB (Figure 4C) were examined. RFB at a concentration of 5 mg/mL had a relatively high cell viability with a cell survival rate of 69.55% ± 2.30%. OFB at a concentration of 1.25 mg/mL led to a cell survival rate of 57.83% ± 3.81%. Apart from these results, other concentrations showed more than 80% cell survival rates, indicating almost no toxicity to cells. Furthermore, at a concentration ranging from 0.16 to 0.625 mg/mL, RFB showed the most excellent proliferation promoting effect among these three samples. We finally chose a concentration of 0.625 mg/mL of RFB, 0.625 mg/mL of HBFB, and 0.32 mg/mL OFB to be used in further studies.

The proliferation effects on the damaged cells caused by UVA and H_2_O_2_ were then studied. As shown in Figure 4D–F, all of the samples can proliferate the decreased cell viabilities induced by UVA and H_2_O_2_ (*p* < 0.05), which indicated the excellent proliferation effects of RFB, HBFB and OFB.

### 3.5. Validation of Key DEGs by RT-qPCR

RT-qPCR was performed to validate the expression levels of the selected DEGs. The results obtained are presented in Figure 5. The PI3K-AKT pathway in the cytoplasm plays an important role in protecting cells from oxidative stress. The decreased expression levels of COL1A1, COL1A2, COL4A5, FN1, IGF2, and PIK3R1 were detected in the cells induced by UVA/H_2_O_2_ (all *p* < 0.01, Figure 5). RFB could significantly upregulate the expression levels of all genes that were downregulated due to UVA/H_2_O_2_ injury (all *p* < 0.01, compared with each model, Figure 5A). Except for COL1A2, HBFB showed similar effects on the expression of COL1A1, COL4A5, FN1, IGF2, and PIK3R1 (all *p* < 0.01, compared with the model separately, Figure 5B). OFB could significantly increase the expression of COL1A1, FN1, IGF2, and PIK3R1 (all *p* < 0.01, compared with the model separately, Figure 5C), but could not promote the expression of COL1A2 in either of the models. Furthermore, the expression of COL4A5 in the H_2_O_2_-induced model can be accelerated by OFB but was suppressed in the UVA-induced model.

## 4. Discussion

Oxidative stress is a disorder between oxidative molecules and insufficient defense of antioxidants, resulting in tissue damage and systemic injury. Oxidative damage leads to elevated levels of ROS in cells. ROS are an oxygen-containing small species, such as singlet oxygen (^1^O_2_), ozone (O_3_), hydroxy radical (OH•), hydrogen peroxide (H_2_O_2_), and superoxide anion radical (O_2_ •−) [6]. Melatonin acts as an antioxidant by scavenging free radicals [32]. There are close relationships between oxidation and skin aging. External agents, such as ultraviolet light and environmental pollution, together with endogenous metabolic irregularities, affect skin cell oxidative damages, resulting in skin aging [33].

However, low levels of oxidative stress activate the transcription of genes encoding proteins which are involved in the defense against oxidative damage, oxidative damage repair mechanisms, and apoptosis. On the contrary, a high level of oxidative stress is a severe threat to lots of macromolecules, such as lipid peroxidation, DNA oxidative damage, protein oxidation, and monosaccharide oxidation [34].

Many efforts have been made to fight against skin aging as people’s cosmetic desires increase, and many studies have examined the functional actives that can prevent oxidative damage [35,36]. UVA and H_2_O_2_ are traditional inducers used to establish oxidative damage cell models in the exploration of the internal mechanism and to screen functional components. It is generally agreed that H_2_O_2_ can induce the production of ROS in cells, leading to oxidative damage [37]. Exogenous H_2_O_2_ treatment is a simple and feasible cell model for studying the mechanism of oxidative damage and actives screening, which can effectively simulate the process of oxidative damage. Similarly, a UVA-induced photoaging model can penetrate more deeply into the skin, elevate ROS, and thus cause DNA damage [38]. In addition, inflammatory reactions are directly associated with the pathogenesis of photoaging [39]. It has been reported that *Rhodiola rosea* fermented by *lactobacillus plantarum*, ectoin, *Laminaria japonica* fermentation broth, and *Lacticaseibacillus paracasei* Subsp. *paracasei SS-01* strain exopolysaccharide all have potential roles in anti-oxidative damage effects [2,6,33,40], no matter which oxidative damage models are used in the studies. However, limited studies have been performed to compare different oxidative stress models in phenotypes and mechanisms.

In the study, RNA-seq technology was performed to examine the similarities and differences of the two models in the function of the involved DEGs and pathways. DEGs of UVA-induced HSFs were mainly focused on the biological processes, such as the regulation of cell proliferation, external stimulus response, and cellular inflammatory response. The negative regulation of DNA-binding transcriptional activity and phosphorylation were inhibited as shown by the GO enrichment analysis. PI3K-AKT signaling pathway was selected as one of the important terms with a high number of genes involved and a significant *P*-value. In the other model of this study, which was H_2_O_2_-induced, processes such as receptor ligand activity, extracellular matrix organization, and response to lipopolysaccharide were significantly upregulated, and collagen-containing extracellular matrix and extracellular matrix structural components were significantly downregulated. Although differences were obtained by the analysis of GO annotation and clustering, the same pathway (PI3K-AKT signaling pathway) obviously changed. Furthermore, the IL-17 signaling pathway was significantly influenced but with a different tendency.

The results obtained partly verified the previous conclusions. Alafiatayo et al. found that UV-irradiated HDF cells showed a lower level of proliferation, more morphological changes, and lower activity compared with the control group. Additionally, the MAPK pathway was found to have been accelerated as antioxidant-related genes were downregulated [41]. In our study, some DEGs in the UVA-induced damage model participated in the PI3K-AKT signaling pathway, the downregulated DEGs of which were mainly involved in cell adhesion, MAPK cascade regulation, and cell proliferation and differentiation. Another transcriptome deep sequencing analysis study conducted by Zheng et al. reported that the differential expression genes altered by UVA were involved in a variety of biological processes, such as the regulation of elastin and phosphoglucomutase encoding levels, and cellular inflammation. Our study found that the functional annotation of the DEGs in UVA-induced HSFs appeared in several terms, such as the regulation of cell proliferation, DNA-binding transcriptional activity, response to external stimuli, and cellular inflammation, which showed consistencies with Zheng et al. [42]. In addition, the expression of ECM and protein complexes in the extracellular matrix and the PI3K-related DEGs decreased simultaneously. As a pivotal node of this signaling pathway, the downregulation of PI3K can interfere with cell survival, apoptosis, DNA damage induction, and downstream protein synthesis, even as it affects cell cycle progression.

Many research studies have examined H_2_O_2_-induced damage to cells. In a previous report published by Barandalla et al., the overall gene expression profile of H_2_O_2_ -induced HUES3 cells were analyzed using an Illumina HT-12 V4 chip, and it was found that the most affected genes were associated with RNA processing and splicing, redox reduction, and sterol metabolic processes [43]. Studies have shown that IL-17 may inhibit apoptosis through the PI3K-AKT signaling pathway, and that TNF could activate the PI3K-AKT signaling pathway [44,45]. Our study also reached a similar conclusion that, when cells were stimulated by H_2_O_2_, the upregulated DEGs were significantly enriched in the cytokine–cytokine receptor interaction and IL-17 signaling pathway, which in turn activated the PI3K signaling pathway. We also performed an enrichment analysis of the PI3K signaling pathway genes involved in the H_2_O_2_ injury model. The DEGs were mainly enriched in cell adhesion, which was found to play an important role in inflammatory responses. The downregulation of the PI3K was mainly due to the downregulation of TLR2/4 and ECM.

Zou et al. systematically mapped single cell transcriptomes involved in human skin aging and revealed that the matriculated expression of transcription factors drove human skin aging. They found that the downregulation of KLF6 and HES1 were the driving force involved in skin aging [46]. In our study, the sequencing results of the UVA-induced model showed that KLF6 was significantly downregulated, too. Therefore, it can be suggested that UVA radiation can lead to skin aging by the downregulation of KLF6, and that the activation of KLF6 may be a major anti-aging approach. HES1 factor was also identified in the sequencing results of H_2_O_2_-induced model but was differentially upregulated. Therefore, it can be suggested that, upon stimulation by H_2_O_2,_ HSFs may activate a self-protection mechanism that activates HES1 factor to prevent cell senescence, which then plays a role in resisting changes caused by external stimuli.

In a previously published report, the underlying signaling pathway and biological processes involved in UVB-induced skin injury were identified [47]. The IL-6 gene of the interleukin family was also identified in our results. ECM is an integral component of all organs and plays a pivotal role in tissue homeostasis and repair [48]. CCN1 is a secreted ECM protein, and research has shown that blocking CCN1 function in vivo can effectively alleviate epidermal hyperplasia and inflammation in psoriasis-like mice [49,50]. In the study, the decline in ECM was a common factor for the activation of PI3K. It indicated that both light and oxidative damage could damage the secretion, synthesis, and normal operation of the cell matrix, and that the activation of the PI3K pathway was an effective regulatory pathway to restore such damage.

Our study showed that both the UVA and H_2_O_2_ injury models were able to regulate the expression levels of the PI3K-AKT signaling pathway genes. The two lesions together affected 11 genes, including PIK3R1, IGF2, ITGA4, LAMA2, LAMA5, LAMB2, NR4A1, COL1A1, COL1A2, COL4A5, and FN1, with similar expression trends. The above-mentioned genes were partly responsible for regulating the central node and partly for encoding the cytoplasmic matrix.

However, the regulation of PI3K-AKT pathway in the UVA and H_2_O_2_ injury models was not completely consistent. In the DEGs analysis results of the UVA injury model, the main inducing factors leading to the downregulation of PI3K signaling were the decrease in the expression of cytokines, protein complexes, and population-stimulating factors in the extracellular matrix. The chain reaction caused by the decrease in PI3K signaling led to the responses in the extracellular matrix, manifested as increased transcriptional activity of DNA damage-inducing transcript (REDD1) and serine threonine kinase (LKB1), which affected normal protein synthesis and hindered cell cycle progression and survival. Although ECM factors were involved in the regulation of PI3K in the H_2_O_2_ injury model, the regulation was mostly proceeded by downregulating TLR2/4 and ITGB genes. Therefore, we can infer that the main action pathway of skin aging damage caused by light and oxidation are concentrated in the PI3K-AKT signaling pathway.

Both of the damaged models suppressed PI3K gene expression and decreased the content of ECM components. H_2_O_2_ was not limited to the regulation of ECM; membrane receptors and integrins were also involved. The results indicated that H_2_O_2_-induced skin oxidative aging was stronger and more complicated than light damage.

*S. commune* is a fungus composed of mycelium and fruiting bodies, which is rich in organic acids and polysaccharides, such as chitosan, glucosamine, and lignocellulolytic enzymes [51,52]. Studies have shown that Schizophyllum exerts anticancer, antitumor, and immunomodulatory activities [27,53,54]. Existing studies have shown that Schizophyllan can effectively restore the brain function of aging mice, increase the SOD content in serum and brain, and reduce the accumulation of malondialdehyde (MDA), thus leading to significant anti-aging activity [55]. *S. commune* fermented broths were also reported to have potential benefits for humans [16,26].

Rice, highland barley, and oats are some of the main food crops used worldwide and are rich in carbohydrates, such as starch, cellulose, and protein. Among them, highland barley and oats contain unique substances, such as arabinoxylan and oat polyphenols, which have been widely applied in food and pharmaceutical industries [56,57,58,59,60]. In the study, three kinds of *S. commune* fermented broths (RFB, HBFB, and OFB) were obtained. Previous studies have demonstrated their anti-oxidative effects in our lab (Some data shown in Appendix A). Here, RFB, HBFB, and OFB were used to verify whether the genes screened out could be accelerated by the broths, and the results were confirmed as expected. It can be concluded that RFB, HBFB, and OFB might play roles in the synthesis of insulin growth factor, cell proliferation, and DNA repair. They may also affect cell survival by promoting the formation of anti-apoptotic protein (Bcl-XL), as well as the upregulation of collagen and fibronectin, thus protecting human skin fibroblasts from UVA or H_2_O_2_ oxidative damage (Figure 6).

RNA-seq analysis and verification results of our study showed that the expression of key genes involved in the PI3K-AKT signaling pathway could be the screening standard to understand the potential antioxidant actives.

There are still some limitations. Although three replicates were performed when we established these two types of models, more parallel tests are needed to obtain results with good stability and repeatability. The criteria used in RNA-seq analysis are empirical, which have great relationships with data analysis. We hope to determine a common screening method to identify active ingredients that can prevent oxidative damage, and plenty of standard antioxidants should be used to verify the results. Further studies should be performed to validate the results in this study.

## 5. Conclusions

In this study, oxidative stress injury models of human skin fibroblasts were established using H_2_O_2_ and UVA stimulation. RNA-seq technology was used to conduct GO and KEGG functional enrichment analysis of the DEGs in the two oxidative injury models. The PI3K-AKT signaling pathway was selected as the common criteria of the two kinds of models. *S. commune*-fermented RFB, HBFB, and OFB were used for further mRNA validation. Results showed that these three broths can protect against oxidative damage through this pathway, indicating that it might provide a research direction for future studies on the oxidative damage mechanisms and the screening of functional actives.

## Figures and Tables

**Figure 1 foods-12-00806-f001:**
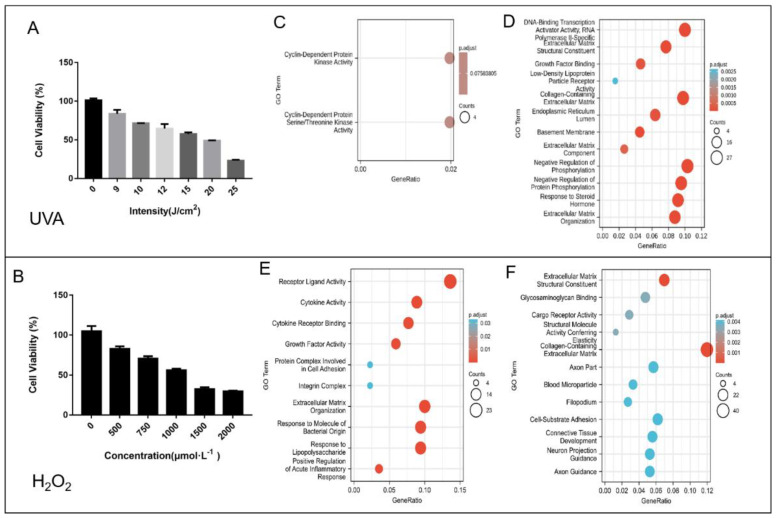
The toxicity of UVA (**A**), H_2_O_2_ (**B**) to HSF cells, and the diagrams of the DEG GO terms (**C**–**F**). The results were expressed as mean ± SD (*n* = 6). The Model in (**A**) was established by UVA (15 J/cm^2^) irradiation; and the Model in (**B**) was established by H_2_O_2_ (1000 μmol/L for 30 min) treatment. In the bubble diagram, the vertical axis indicated GO terms and the horizontal axis represented gene ratio involved in each term. The size of dots indicated the number of genes in the GO term. The color of dots exhibited the significance. (**C**,**E**), differentially upregulated genes; (**D**,**F**), differently downregulated genes.

**Figure 2 foods-12-00806-f002:**
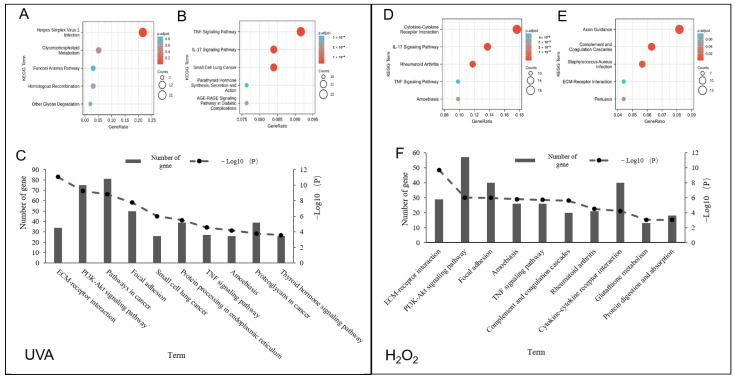
Bar diagrams and bubble diagrams of KEGG pathway enrichment analysis of UVA-induced (**A**–**C**) and H_2_O_2_-induced (**D**–**F**) DEGs. (**A**, **D**), differentially upregulated genes in each study; (**B**,**E**), differently downregulated genes in each study. In the bubble diagram, the vertical axis indicated GO terms and the horizontal axis represented gene ratio involved in each term. The size of dots indicated the number of genes in the GO term. The color of dots exhibited the significance. (**C**,**F**), the top 10 ranked KEGG terms of both upregulated and downregulated DEGs. In the bar diagram, the right vertical axis indicated the significance of each term by exhibiting a value of −log10(P), and the left vertical axis showed the number of genes involved in each term. The horizontal axis represented the top 10 ranked KEGG terms.

**Figure 3 foods-12-00806-f003:**
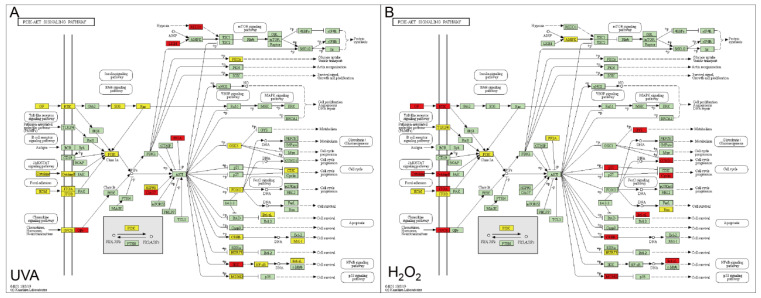
PI3K-AKT Signaling Pathway in UVA-induced(**A**) and H_2_O_2_-induced(**B**) model. Red: upregulated gene; Yellow: downregulated gene; Green: not regulated significantly gene.

**Figure 4 foods-12-00806-f004:**
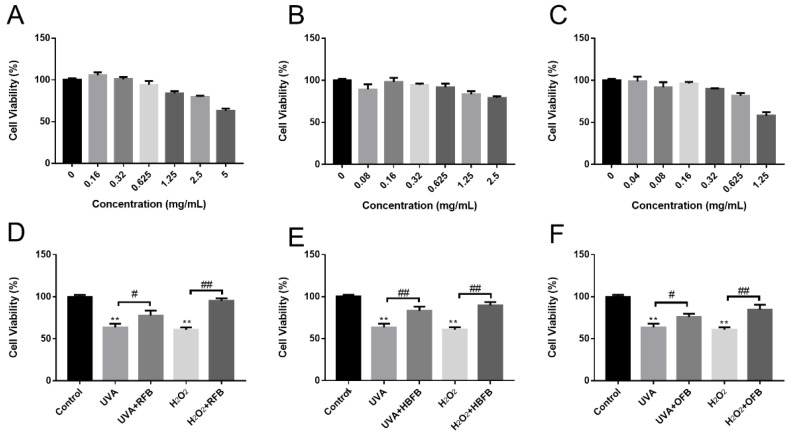
The toxicity of RFB (**A**), HBFB (**B**) and OFB (**C**) to HSF cells and the repair effects of RFB (**D**), HBFB (**E**), and OFB (**F**) on UVA-induced and H_2_O_2_-induced oxidative stress damage in HSF cells. The results were expressed as mean ± SD (*n* = 6). The discussed concentrations of RFB, HBFB and OFB were 0.625 mg/mL, 0.625 mg/mL, and 0.32 mg/mL, respectively. **, *p* < 0.01, UVA/H_2_O_2_-induced model compared with the DMEM-treated control. # *p* < 0.05, and ## *p* < 0.01, RFB, HBFB, OFB compared with the Model.

**Figure 5 foods-12-00806-f005:**
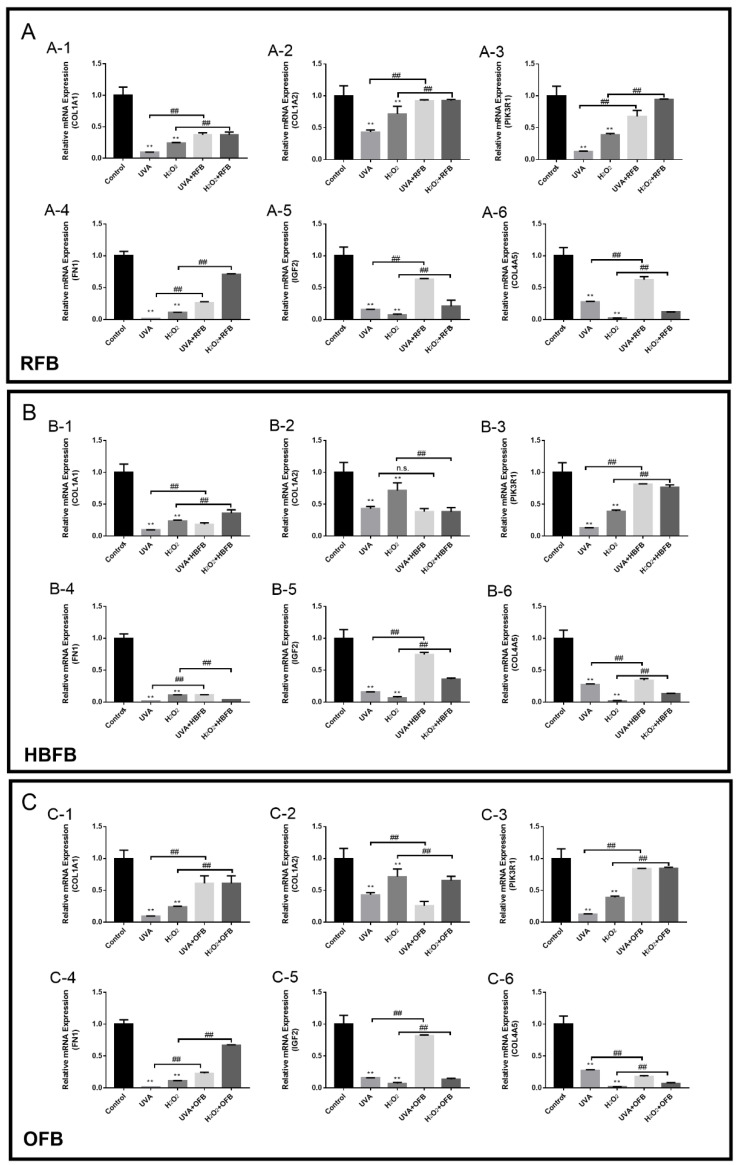
The relative gene expression changes after the repair of rice fermentation broth (RFB, **A**), highland barley fermentation broth (HBFB, **B**) and oats fermented broth (OFB, **C**) on UVA and H_2_O_2_ injury. RFB, HBFB, and OFB are three kinds of *S. commune* fermentation broths**.** Genes, COL1A1 (**A-1**, **B-1**, **C-1**), COL1A2 (**A-2**, **B-2**, **C-2**), PIK3R1 (**A-3**, **B-3**, **C-3**), FN1 (**A-4**, **B-4**, **C-4**), IGF2 (**A-5**, **B-5**, **C-5**) and COL4A5 (**A-6**, **B-6**, **C-6**) were measured. The UVA bar represented the model established by UVA (15 J/cm^2^) treatment; the H_2_O_2_ bar represented the model established by H_2_O_2_ (1000 µmol/L for 30 min) treatment. **, *p* < 0.01, the UVA/H_2_O_2_-induced model compared with the DMEM-treated control. ## *p* < 0.01 and n.s. *p* > 0.05, RFB, HBFB, and OFB treated cells compared with the Model.

**Figure 6 foods-12-00806-f006:**
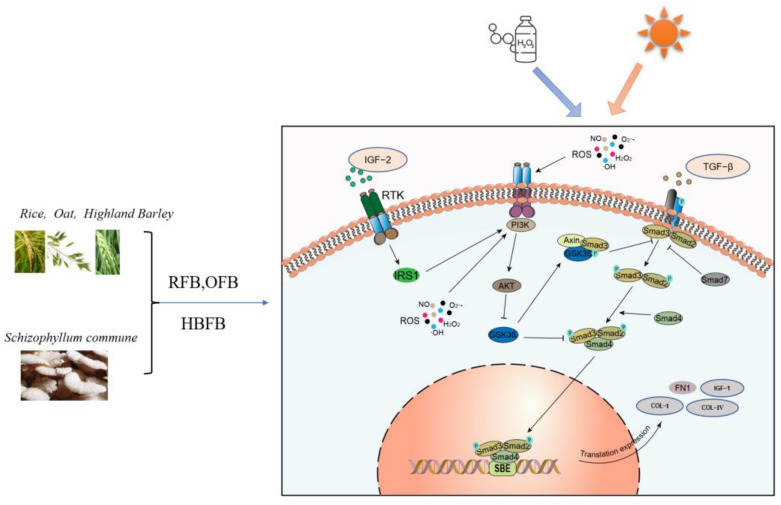
RFB, HBFB and OFB influence the signaling pathways of UVA/H_2_O_2_-induced skin oxidative damage. RFB, HBFB, and OFB are three kinds of *S. commune* fermentation broths. RFB, rice fermentation broth; HBFB, highland barley fermentation broth; OFB, oats fermented broth.

**Table 1 foods-12-00806-t001:** Summary of read screening and mapping results of the sequences generated from cells with different treatments.

Term	Sample	Raw Reads	Clean Reads	Total Reads	Multiple Mapped	Uniquely Mapped
H_2_O_2_	control1	52,006,260	51,489,806	51,489,806	1,585,446 (3.08%)	48,550,893 (94.29%)
	control2	51,434,366	51,007,382	51,007,382	1,656,100 (3.25%)	48,073,211(94.25%)
	contro13	47,992,480	47,570,278	47,570,278	1,767,599 (3.72%)	44,528,485 (93.61%)
	model1	42,375,544	41,851,304	41,851,304	2,523,225 (6.03%)	37,561,116 (89.75%)
	model2	55,834,672	55,256,324	55,256,324	1,946,513 (3.52%)	51,879,913 (93.89%)
	model3	53,168,944	52,654,864	52,654,864	1,852,768 (3.52%)	49,508,846 (94.03%)
UVA	control1	203,492,014	201,140,460	201,140,460	9,220,602 (4.58%)	108,745,918 (54.06%)
	control2	214,473,458	212,140,016	212,140,016	10,108,543 (4.77%)	114,327,168 (53.89%)
	control3	201,264,476	199,240,206	199,240,206	8,290,552 (4.16%)	103,655,047 (52.03%)
	model1	183,171,608	180,873,264	180,873,264	10,508,759 (5.81%)	117542871 (64.99%)
	model2	219,499,948	216,959,018	216,959,018	12,593,146 (5.8%)	137,233,440 (63.25%)
	model3	207,223,010	204,370,646	204,370,646	12,256,006 (6.0%)	131,192,428 (64.19%)

Notes: Control represents the cell sample without the UVA/H_2_O_2_; Model represents the cell sample with the UVA/H_2_O_2_. Three replicates of Control (Control 1, 2 and 3) and Model (Model 1, 2 and 3) treatments were carried out in RNA-seq analysis.

**Table 2 foods-12-00806-t002:** Primer sequences for real-time PCR.

Gene	Direction	Primer Pair Sequence (5′→3′)
GAPDH	F	TCAGACACCATGGGGAAGGT
	R	TCCCGTTCTCAGCCATGTAG
COL4A5	F	CAAGGTCTACCAGGTCCAGAA
	R	TCATTCCATTGAGACCCGGC
FN1	F	CCCAATTGAGTGCTTCATGCC
	R	CCTCCAGAGCAAAGGGCTTA
IGF2	F	TCCTGTGAAAGAGACTTCCAG
	R	GTCTCACTGGGGCGGTAAG
COL1A1	F	GAGGGCCAAGACGAAGACATC
	R	CAGATCACGTCATCGCACAAC
COL1A2	F	GTTGCTGCTTGCAGTAACCTT
	R	AGGGCCAAGTCCAACTCCTT
PIK3R1	F	ACCACTACCGGAATGAATCTCT
	R	GGGATGTGCGGGTATATTCTTC

Notes: F, forward primer; R, reverse primer; the primer organism is homo sapiens.

**Table 3 foods-12-00806-t003:** Functional characterization of up/downregulated differential expressed genes in UVA-induced model.

Gene_Id	Gene Name	Gene Description	Log2FC	P-Adjust	Function
ENSG00000187961	KLHL17	kelch like family member 17	0.457	8.97 × 10^−5^	Cytoskeleton organization
ENSG00000125817	CENPB	centromere protein B	0.342	1.87 × 10^−10^	Cytoskeleton organization
ENSG00000011426	ANLN	anillin actin binding protein	−0.299	3.31 × 10^−2^	Cytoskeleton organization
ENSG00000123384	LRP1	LDL receptor related protein 1	−0.435	2.99 × 10^−16^	Cytoskeleton organization
ENSG00000288380	CRIPAK	cysteine rich PAK1 inhibitor	−0.401	1.97 × 10^−2^	Cytoskeleton organization
ENSG00000275993	SIK1B	salt inducible kinase 1B (putative)	−2.211	1.70 × 10^−22^	Cytoskeleton organization
ENSG00000147202	DIAPH2	diaphanous related formin 2	−0.334	1.34 × 10^−8^	Cytoskeleton organization
ENSG00000160551	TAOK1	TAO kinase 1	−0.264	4.71 × 10^−9^	Cytoskeleton organization
ENSG00000075826	SEC31B	SEC31 homolog B, COPII coat complex component	0.370	6.71 × 10^−5^	Transport
ENSG00000137700	SLC37A4	solute carrier family 37 member 4	0.485	1.26 × 10^−3^	Transport
ENSG00000225697	SLC26A6	solute carrier family 26 member 6	0.448	3.28 × 10^−12^	Transport
ENSG00000155287	SLC25A28	solute carrier family 25 member28	0.403	3.14 × 10^−6^	Transport
ENSG00000153291	SLC25A27	solute carrier family 25 member 27	0.278	6.30 × 10^−4^	Transport
ENSG00000213901	SLC23A3	solute carrier family 23 member 3	0.340	3.43 × 10^−2^	Transport
ENSG00000197208	SLC22A4	solute carrier family 22 member4	0.292	2.76 × 10^−5^	Transport
ENSG00000101194	SLC17A9	solute carrier family 17 member 9	0.421	2.03 × 10^−5^	Transport
ENSG00000105643	ARRDC2	arrestin domain containing 2	0.276	4.49 × 10^−2^	Transport
ENSG00000205593	DENND6B	DENN domain containing 6B	0.293	4.04 × 10^−2^	Transport
ENSG00000157514	TSC22D3	TSC22 domain family member 3	0.285	7.88 × 10^−4^	Transport
ENSG00000140104	CLBA1	clathrin binding box of aftiphilin containing 1	0.328	2.62 × 10^−2^	Transport
ENSG00000106266	SNX8	sorting nexin 8	0.294	2.33 × 10^−4^	Transport
ENSG00000274512	TBC1D3L	TBC1 domain family member 3L	0.377	2.85 × 10^−3^	Transport
ENSG00000104886	PLEKHJ1	pleckstrin homology domain containing J1	0.311	9.15 × 10^−3^	Transport
ENSG00000177096	PHETA2	PH domain containing endocytic trafficking adaptor 2	0.305	1.35 × 10^−2^	Transport
ENSG00000257390	AC023055.1	novel protein	0.467	1.20 × 10^−3^	Transport
ENSG00000254852	NPIPA2	nuclear pore complex interacting protein family member A2	0.279	4.98 × 10^−3^	Transport
ENSG00000095066	HOOK2	hook microtubule tethering protein 2	0.297	1.92 × 10^−2^	Transport
ENSG00000101199	ARFGAP1	ADP ribosylation factor GTPase activating protein 1	0.280	9.43 × 10^−7^	Transport
ENSG00000213983	AP1G2	adaptor related protein complex 1 subunit gamma 2	0.318	1.78 × 10^−2^	Transport
ENSG00000181404	WASHC1	WASH complex subunit 1	0.450	6.60 × 10^−18^	Transport
ENSG00000139190	VAMP1	vesicle associated membrane protein 1	0.601	5.83 × 10^−6^	Transport
ENSG00000162341	TPCN2	two pore segment channel 2	0.292	1.43 × 10^−2^	Transport
ENSG00000114268	PFKFB4	6-phosphofructo-2-kinase/fructose-2,6-biphosphatase 4	0.632	1.33 × 10^−3^	Transport
ENSG00000196655	TRAPPC4	trafficking protein particle complex 4	0.275	4.52 × 10^−2^	Transport
ENSG00000102287	GABRE	gamma-aminobutyric acid type A receptor epsilon subunit	0.317	4.13 × 10^−4^	Transport
ENSG00000225663	MCRIP1	MAPK regulated corepressor interacting protein 1	0.270	4.63 × 10^−3^	Transport
ENSG00000081692	JMJD4	jumonji domain containing 4	0.353	2.60 × 10^−2^	Transport
ENSG00000168026	TTC21A	tetratricopeptide repeat domain 21A	0.383	4.65 × 10^−2^	Transport
ENSG00000108932	SLC16A6	solute carrier family 16 member 6	−0.416	1.95 × 10^−4^	Transport
ENSG00000117479	SLC19A2	solute carrier family 19 member 2	−1.360	6.72 × 10^−42^	Transport
ENSG00000140199	SLC12A6	solute carrier family 12 member6	−0.279	2.21 × 10^−5^	Transport
ENSG00000171488	LRRC8C	leucine rich repeat containing 8 VRAC subunit C	−0.269	2.37 × 10^−2^	Transport
ENSG00000169446	MMGT1	membrane magnesium transporter 1	−0.306	4.40 × 10^−4^	Transport
ENSG00000228253	MT-ATP8	mitochondrially encoded ATP synthase membrane subunit 8	−0.567	6.75 × 10^−4^	Transport
ENSG00000198899	MT-ATP6	mitochondrially encoded ATP synthase membrane subunit 6	−0.457	2.44 × 10^−3^	Transport
ENSG00000165714	BORCS5	BLOC-1 related complex subunit 5	−0.285	4.97 × 10^−3^	Transport
ENSG00000181333	HEPHL1	hephaestin like 1	−0.285	3.27 × 10^−3^	Transport
ENSG00000107771	CCSER2	coiled-coil serine rich protein 2	−0.272	1.36 × 10^−6^	Transport
ENSG00000115657	ABCB6	ATP binding cassette subfamily B member 6 (Langereis blood group)	0.303	7.37 × 10^−4^	Cellular and extracellular matrix
ENSG00000117425	PTCH2	patched 2	0.285	1.16 × 10^−2^	Cellular and extracellular matrix
ENSG00000125551	PLGLB2	plasminogen like B2	0.383	1.82 × 10^−2^	Cellular and extracellular matrix
ENSG00000164877	MICALL2	MICAL like 2	0.285	9.72 × 10^−3^	Cellular and extracellular matrix
ENSG00000156535	CD109	CD109 molecule	−0.399	1.50 × 10^−25^	Cellular and extracellular matrix
ENSG00000185022	MAFF	MAF bZIP transcription factor F	−0.968	5.78 × 10^−80^	Cellular and extracellular matrix
ENSG00000166147	FBN1	fibrillin 1	−0.421	4.89 × 10^−30^	Cellular and extracellular matrix
ENSG00000165240	ATP7A	ATPase copper transporting alpha	−0.277	2.43 × 10^−4^	Cellular and extracellular matrix
ENSG00000142871	CCN1	cellular communication network factor 1	−0.410	4.51 × 10^−25^	Cellular and extracellular matrix
ENSG00000038427	VCAN	versican	−0.492	2.64 × 10^−13^	Cellular and extracellular matrix
ENSG00000116962	NID1	nidogen 1	−0.356	1.04 × 10^−21^	Cellular and extracellular matrix
ENSG00000187955	COL14A1	collagen type XIV alpha 1 chain	−0.303	1.32 × 10^−3^	Cellular and extracellular matrix
ENSG00000196569	LAMA2	laminin subunit alpha 2	−0.407	5.66 × 10^−28^	Cellular and extracellular matrix
ENSG00000142798	HSPG2	heparan sulfate proteoglycan 2	−0.440	8.61 × 10^−14^	Cellular and extracellular matrix
ENSG00000123500	COL10A1	collagen type X alpha 1 chain	−0.282	2.54 × 10^−2^	Cellular and extracellular matrix
ENSG00000187498	COL4A1	collagen type IV alpha 1 chain	−0.265	7.12 × 10^−12^	Cellular and extracellular matrix
ENSG00000115414	FN1	fibronectin 1	−0.446	1.23 × 10^−34^	Cellular and extracellular matrix
ENSG00000163359	COL6A3	collagen type VI alpha 3 chain	−0.412	3.41 × 10^−38^	Cellular and extracellular matrix
ENSG00000103196	CRISPLD2	cysteine rich secretory protein LCCL domain containing 2	−0.321	6.09 × 10^−4^	Cellular and extracellular matrix
ENSG00000111799	COL12A1	collagen type XII alpha 1 chain	−0.388	7.55 × 10^−28^	Cellular and extracellular matrix
ENSG00000080561	MID2	midline 2	−0.275	8.28 × 10^−4^	Cellular and extracellular matrix
ENSG00000171223	JUNB	JunB proto-oncogene, AP-1 transcription factor subunit	−1.704	9.13 × 10^−173^	Cellular and extracellular matrix
ENSG00000101825	MXRA5	matrix remodeling associated 5	−0.387	8.91 × 10^−32^	Cellular and extracellular matrix
ENSG00000113369	ARRDC3	arrestin domain containing 3	−0.289	6.30 × 10^−4^	Cellular and extracellular matrix
ENSG00000138759	FRAS1	Fraser extracellular matrix complex subunit 1	−0.447	4.82 × 10^−14^	Cellular and extracellular matrix
ENSG00000106780	MEGF9	multiple EGF like domains 9	−0.284	3.72 × 10^−3^	Cellular and extracellular matrix
ENSG00000119681	LTBP2	latent transforming growth factor beta binding protein 2	−0.270	3.80 × 10^−10^	Cellular and extracellular matrix
ENSG00000182179	UBA7	ubiquitin like modifier activating enzyme 7	0.278	6.06 × 10^−4^	Metabolism/Cell Proliferation/Regulation
ENSG00000148399	DPH7	diphthamide biosynthesis 7	0.345	5.72 × 10^−3^	Metabolism/Cell Proliferation/Regulation
ENSG00000167733	HSD11B1L	hydroxysteroid 11-beta dehydrogenase 1 like	0.284	4.11 × 10^−2^	Metabolism/Cell Proliferation/Regulation
ENSG00000104852	SNRNP70	small nuclear ribonucleoprotein U1 subunit 70	0.340	6.65 × 10^−17^	Metabolism/Cell Proliferation/Regulation
ENSG00000125901	MRPS26	mitochondrial ribosomal protein S26	0.289	3.13 × 10^−2^	Metabolism/Cell Proliferation/Regulation
ENSG00000183513	COA5	cytochrome c oxidase assembly factor 5	0.273	4.15 × 10^−3^	Metabolism/Cell Proliferation/Regulation
ENSG00000165792	METTL17	methyltransferase like 17	0.401	5.91 × 10^−5^	Metabolism/Cell Proliferation/Regulation
ENSG00000100429	HDAC10	histone deacetylase 10	0.274	1.11 × 10^−2^	Metabolism/Cell Proliferation/Regulation
ENSG00000141519	CCDC40	coiled-coil domain containing 40	0.383	3.27 × 10^−2^	Metabolism/Cell Proliferation/Regulation
ENSG00000172828	CES3	carboxylesterase 3	0.506	2.88 × 10^−2^	Metabolism/Cell Proliferation/Regulation
ENSG00000139631	CSAD	cysteine sulfinic acid decarboxylase	0.311	8.25 × 10^−6^	Metabolism/Cell Proliferation/Regulation
ENSG00000112578	BYSL	bystin like	0.331	1.17 × 10^−2^	Metabolism/Cell Proliferation/Regulation
ENSG00000213398	LCAT	lecithin-cholesterol acyltransferase	0.488	3.70 × 10^−3^	Metabolism/Cell Proliferation/Regulation
ENSG00000232653	GOLGA8N	golgin A8 family member N	0.348	2.84 × 10^−2^	Metabolism/Cell Proliferation/Regulation
ENSG00000145020	AMT	aminomethyltransferase	0.388	4.57 × 10^−2^	Metabolism/Cell Proliferation/Regulation
ENSG00000178038	ALS2CL	ALS2 C-terminal like	0.409	1.90 × 10^−2^	Metabolism/Cell Proliferation/Regulation
ENSG00000179886	TIGD5	tigger transposable element derived 5	0.313	4.31 × 10^−2^	Metabolism/Cell Proliferation/Regulation
ENSG00000128710	HOXD10	homeobox D10	0.319	3.71 × 10^−3^	Metabolism/Cell Proliferation/Regulation
ENSG00000108641	B9D1	B9 domain containing 1	0.286	3.29 × 10^−2^	Metabolism/Cell Proliferation/Regulation
ENSG00000197774	EME2	essential meiotic structure-specific endonuclease subunit 2	0.385	8.64 × 10^−5^	Metabolism/Cell Proliferation/Regulation
ENSG00000172732	MUS81	MUS81 structure-specific endonuclease subunit	0.276	7.00 × 10^−3^	Metabolism/Cell Proliferation/Regulation
ENSG00000039650	PNKP	polynucleotide kinase 3′-phosphatase	0.284	1.49 × 10^−3^	Metabolism/Cell Proliferation/Regulation
ENSG00000129250	KIF1C	kinesin family member 1C	0.366	1.72 × 10^−17^	Metabolism/Cell Proliferation/Regulation
ENSG00000010292	NCAPD2	non-SMC condensin I complex subunit D2	0.328	2.36 × 10^−6^	Metabolism/Cell Proliferation/Regulation
ENSG00000146063	TRIM41	tripartite motif containing 41	0.332	1.10 × 10^−6^	Metabolism/Cell Proliferation/Regulation
ENSG00000140983	RHOT2	ras homolog family member T2	0.328	5.44 × 10^−6^	Metabolism/Cell Proliferation/Regulation
ENSG00000163482	STK36	serine/threonine kinase 36	0.273	4.31 × 10^−5^	Metabolism/Cell Proliferation/Regulation
ENSG00000138834	MAPK8IP3	mitogen-activated protein kinase 8 interacting protein 3	0.309	1.58 × 10^−6^	Metabolism/Cell Proliferation/Regulation
ENSG00000148120	AOPEP	aminopeptidase O (putative)	0.469	1.77 × 10^−9^	Metabolism/Cell Proliferation/Regulation
ENSG00000167100	SAMD14	sterile alpha motif domain containing 14	0.270	4.18 × 10^−2^	Metabolism/Cell Proliferation/Regulation
ENSG00000110455	ACCS	1-aminocyclopropane-1-carboxylate synthase homolog (inactive)	0.357	3.47 × 10^−4^	Metabolism/Cell Proliferation/Regulation
ENSG00000158062	UBXN11	UBX domain protein 11	0.303	1.60 × 10^−2^	Metabolism/Cell Proliferation/Regulation
ENSG00000107829	FBXW4	F-box and WD repeat domain containing 4	0.347	1.00 × 10^−4^	Metabolism/Cell Proliferation/Regulation
ENSG00000138835	RGS3	regulator of G protein signaling 3	0.343	9.63 × 10^−7^	Metabolism/Cell Proliferation/Regulation
ENSG00000232119	MCTS1	MCTS1 re-initiation and release factor	0.278	1.81 × 10^−3^	Metabolism/Cell Proliferation/Regulation
ENSG00000167280	ENGASE	endo-beta-N-acetylglucosaminidase	0.409	1.64 × 10^−5^	Metabolism/Cell Proliferation/Regulation
ENSG00000149716	LTO1	LTO1 maturation factor of ABCE1	0.307	4.33 × 10^−3^	Metabolism/Cell Proliferation/Regulation
ENSG00000250151	ARPC4-TTLL3	ARPC4-TTLL3 readthrough	0.354	1.51 × 10^−4^	Metabolism/Cell Proliferation/Regulation
ENSG00000073605	GSDMB	gasdermin B	0.448	1.17 × 10^−8^	Metabolism/Cell Proliferation/Regulation
ENSG00000148824	MTG1	mitochondrial ribosome associated GTPase 1	0.303	1.28 × 10^−2^	Metabolism/Cell Proliferation/Regulation
ENSG00000137504	CREBZF	CREB/ATF bZIP transcription factor	0.287	3.31 × 10^−4^	Metabolism/Cell Proliferation/Regulation
ENSG00000180902	D2HGDH	D-2-hydroxyglutarate dehydrogenase	0.274	3.10 × 10^−2^	Metabolism/Cell Proliferation/Regulation
ENSG00000042429	MED17	mediator complex subunit 17	0.408	1.86 × 10^−3^	Metabolism/Cell Proliferation/Regulation
ENSG00000149930	TAOK2	TAO kinase 2	0.456	8.73 × 10^−17^	Metabolism/Cell Proliferation/Regulation
ENSG00000139546	TARBP2	TARBP2 subunit of RISC loading complex	0.310	9.44 × 10^−3^	Metabolism/Cell Proliferation/Regulation
ENSG00000116001	TIA1	TIA1 cytotoxic granule associated RNA binding protein	0.332	8.51 × 10^−6^	Metabolism/Cell Proliferation/Regulation
ENSG00000115053	NCL	nucleolin	0.275	1.29 × 10^−12^	Metabolism/Cell Proliferation/Regulation
ENSG00000173559	NABP1	nucleic acid binding protein 1	0.285	4.36 × 10^−10^	Metabolism/Cell Proliferation/Regulation
ENSG00000198585	NUDT16	nudix hydrolase 16	0.317	4.87 × 10^−7^	Metabolism/Cell Proliferation/Regulation
ENSG00000167393	PPP2R3B	protein phosphatase 2 regulatory subunit B’’beta	0.331	2.50 × 10^−2^	Metabolism/Cell Proliferation/Regulation
ENSG00000141456	PELP1	proline, glutamate and leucine rich protein 1	0.337	3.84 × 10^−6^	Metabolism/Cell Proliferation/Regulation
ENSG00000128159	TUBGCP6	tubulin gamma complex associated protein 6	0.283	1.47 × 10^−5^	Metabolism/Cell Proliferation/Regulation
ENSG00000078399	HOXA9	homeobox A9	0.355	4.10 × 10^−4^	Metabolism/Cell Proliferation/Regulation
ENSG00000213339	QTRT1	queuine tRNA-ribosyltransferase catalytic subunit 1	0.292	4.27 × 10^−3^	Metabolism/Cell Proliferation/Regulation
ENSG00000177192	PUS1	pseudouridine synthase 1	0.275	4.58 × 10^−2^	Metabolism/Cell Proliferation/Regulation
ENSG00000144785	AC073896.1	novel protein	0.382	4.61 × 10^−3^	Metabolism/Cell Proliferation/Regulation
ENSG00000185024	BRF1	BRF1 RNA polymerase III transcription initiation factor subunit	0.301	7.22 × 10^−4^	Metabolism/Cell Proliferation/Regulation
ENSG00000228049	POLR2J2	RNA polymerase II subunit J2	0.339	3.50 × 10^−2^	Metabolism/Cell Proliferation/Regulation
ENSG00000197782	ZNF780A	zinc finger protein 780A	0.281	1.47 × 10^−2^	Metabolism/Cell Proliferation/Regulation
ENSG00000147789	ZNF7	zinc finger protein 7	0.282	1.60 × 10^−3^	Metabolism/Cell Proliferation/Regulation
ENSG00000171163	ZNF692	zinc finger protein 692	0.390	1.49 × 10^−5^	Metabolism/Cell Proliferation/Regulation
ENSG00000176024	ZNF613	zinc finger protein 613	0.405	3.87 × 10^−2^	Metabolism/Cell Proliferation/Regulation
ENSG00000180626	ZNF594	zinc finger protein 594	0.552	8.53 × 10^−6^	Metabolism/Cell Proliferation/Regulation
ENSG00000161551	ZNF577	zinc finger protein 577	0.309	9.79 × 10^−3^	Metabolism/Cell Proliferation/Regulation
ENSG00000167785	ZNF558	zinc finger protein 558	0.330	6.43 × 10^−3^	Metabolism/Cell Proliferation/Regulation
ENSG00000152433	ZNF547	zinc finger protein 547	0.503	2.07 × 10^−2^	Metabolism/Cell Proliferation/Regulation
ENSG00000144026	ZNF514	zinc finger protein 514	0.270	4.19 × 10^−3^	Metabolism/Cell Proliferation/Regulation
ENSG00000196653	ZNF502	zinc finger protein 502	0.320	4.78 × 10^−2^	Metabolism/Cell Proliferation/Regulation
ENSG00000083817	ZNF416	zinc finger protein 416	0.337	4.62 × 10^−2^	Metabolism/Cell Proliferation/Regulation
ENSG00000175213	ZNF408	zinc finger protein 408	0.394	1.40 × 10^−2^	Metabolism/Cell Proliferation/Regulation
ENSG00000130684	ZNF337	zinc finger protein 337	0.340	6.05 × 10^−7^	Metabolism/Cell Proliferation/Regulation
ENSG00000083812	ZNF324	zinc finger protein 324	0.378	1.21 × 10^−4^	Metabolism/Cell Proliferation/Regulation
ENSG00000182986	ZNF320	zinc finger protein 320	0.406	6.47 × 10^−5^	Metabolism/Cell Proliferation/Regulation
ENSG00000205903	ZNF316	zinc finger protein 316	0.313	1.86 × 10^−5^	Metabolism/Cell Proliferation/Regulation
ENSG00000174652	ZNF266	zinc finger protein 266	0.433	1.55 × 10^−10^	Metabolism/Cell Proliferation/Regulation
ENSG00000167380	ZNF226	zinc finger protein 226	0.337	1.45 × 10^−4^	Metabolism/Cell Proliferation/Regulation
ENSG00000197841	ZNF181	zinc finger protein 181	0.298	3.56 × 10^−2^	Metabolism/Cell Proliferation/Regulation
ENSG00000154957	ZNF18	zinc finger protein 18	0.339	2.32 × 10^−2^	Metabolism/Cell Proliferation/Regulation
ENSG00000179909	ZNF154	zinc finger protein 154	0.341	3.55 × 10^−3^	Metabolism/Cell Proliferation/Regulation
ENSG00000213762	ZNF134	zinc finger protein 134	0.280	2.08 × 10^−3^	Metabolism/Cell Proliferation/Regulation
ENSG00000125846	ZNF133	zinc finger protein 133	0.325	1.48 × 10^−3^	Metabolism/Cell Proliferation/Regulation
ENSG00000204946	ZNF783	zinc finger family member 783	0.265	1.58 × 10^−3^	Metabolism/Cell Proliferation/Regulation
ENSG00000197114	ZGPAT	zinc finger CCCH-type and G-patch domain containing	0.270	1.89 × 10^−2^	Metabolism/Cell Proliferation/Regulation
ENSG00000140987	ZSCAN32	zinc finger and SCAN domain containing 32	0.486	6.71 × 10^−5^	Metabolism/Cell Proliferation/Regulation
ENSG00000114853	ZBTB47	zinc finger and BTB domain containing 47	0.372	4.54 × 10^−9^	Metabolism/Cell Proliferation/Regulation
ENSG00000099899	TRMT2A	tRNA methyltransferase 2 homolog A	0.293	7.14 × 10^−4^	Metabolism/Cell Proliferation/Regulation
ENSG00000100038	TOP3B	DNA topoisomerase III beta	0.295	1.09 × 10^−2^	Metabolism/Cell Proliferation/Regulation
ENSG00000198056	PRIM1	DNA primase subunit 1	0.461	1.32 × 10^−3^	Metabolism/Cell Proliferation/Regulation
ENSG00000178028	DMAP1	DNA methyltransferase 1 associated protein 1	0.311	4.49 × 10^−4^	Metabolism/Cell Proliferation/Regulation
ENSG00000168209	DDIT4	DNA damage inducible transcript 4	0.278	7.35 × 10^−5^	Metabolism/Cell Proliferation/Regulation
ENSG00000221978	CCNL2	cyclin L2	0.429	2.31 × 10^−23^	Metabolism/Cell Proliferation/Regulation
ENSG00000250506	CDK3	cyclin dependent kinase 3	0.535	6.59 × 10^−4^	Metabolism/Cell Proliferation/Regulation
ENSG00000156345	CDK20	cyclin dependent kinase 20	0.287	3.81 × 10^−2^	Metabolism/Cell Proliferation/Regulation
ENSG00000248333	CDK11B	cyclin dependent kinase 11B	0.299	8.71 × 10^−4^	Metabolism/Cell Proliferation/Regulation
ENSG00000185324	CDK10	cyclin dependent kinase 10	0.344	1.84 × 10^−3^	Metabolism/Cell Proliferation/Regulation
ENSG00000130305	NSUN5	NOP2/Sun RNA methyltransferase 5	0.264	4.48 × 10^−2^	Metabolism/Cell Proliferation/Regulation
ENSG00000134186	PRPF38B	pre-mRNA processing factor 38B	0.269	6.68 × 10^−4^	Metabolism/Cell Proliferation/Regulation
ENSG00000103168	TAF1C	TATA-box binding protein associated factor, RNA polymerase I subunit C	0.324	7.71 × 10^−5^	Metabolism/Cell Proliferation/Regulation
ENSG00000178718	RPP25	ribonuclease P and MRP subunit p25	0.280	3.45 × 10^−2^	Metabolism/Cell Proliferation/Regulation
ENSG00000264668	AC138696.1	novel protein	0.384	3.75 × 10^−7^	Metabolism/Cell Proliferation/Regulation
ENSG00000041988	THAP3	THAP domain containing 3	0.347	3.09 × 10^−2^	Metabolism/Cell Proliferation/Regulation
ENSG00000104129	DNAJC17	DnaJ heat shock protein family (Hsp40) member C17	0.364	3.96 × 10^−2^	Metabolism/Cell Proliferation/Regulation
ENSG00000187531	SIRT7	sirtuin 7	0.386	1.73 × 10^−2^	Metabolism/Cell Proliferation/Regulation
ENSG00000108479	GALK1	galactokinase 1	0.311	9.61 × 10^−3^	Metabolism/Cell Proliferation/Regulation
ENSG00000140400	MAN2C1	mannosidase alpha class 2C member 1	0.384	3.49 × 10^−12^	Metabolism/Cell Proliferation/Regulation
ENSG00000142102	PGGHG	protein-glucosylgalactosylhydroxylysine glucosidase	0.388	6.40 × 10^−7^	Metabolism/Cell Proliferation/Regulation
ENSG00000181274	FRAT2	FRAT regulator of WNT signaling pathway 2	0.331	4.77 × 10^−2^	Metabolism/Cell Proliferation/Regulation
ENSG00000215788	TNFRSF25	TNF receptor superfamily member 25	0.558	1.63 × 10^−2^	Metabolism/Cell Proliferation/Regulation
ENSG00000153179	RASSF3	Ras association domain family member 3	0.326	5.71 × 10^−8^	Metabolism/Cell Proliferation/Regulation
ENSG00000115875	SRSF7	serine and arginine rich splicing factor 7	0.293	1.90 × 10^−4^	Metabolism/Cell Proliferation/Regulation
ENSG00000162910	MRPL55	mitochondrial ribosomal protein L55	0.295	4.41 × 10^−2^	Metabolism/Cell Proliferation/Regulation
ENSG00000172586	CHCHD1	coiled-coil-helix-coiled-coil-helix domain containing 1	0.415	2.99 × 10^−2^	Metabolism/Cell Proliferation/Regulation
ENSG00000136938	ANP32B	acidic nuclear phosphoprotein 32 family member B	0.309	1.46 × 10^−7^	Metabolism/Cell Proliferation/Regulation
ENSG00000177595	PIDD1	p53-induced death domain protein 1	0.300	2.42 × 10^−4^	Metabolism/Cell Proliferation/Regulation
ENSG00000129473	BCL2L2	BCL2 like 2	0.273	1.67 × 10^−4^	Metabolism/Cell Proliferation/Regulation
ENSG00000184207	PGP	phosphoglycolate phosphatase	0.409	5.30 × 10^−4^	Metabolism/Cell Proliferation/Regulation
ENSG00000136271	DDX56	DEAD-box helicase 56	0.292	5.37 × 10^−7^	Metabolism/Cell Proliferation/Regulation
ENSG00000106404	CLDN15	claudin 15	0.436	2.44 × 10^−3^	Metabolism/Cell Proliferation/Regulation
ENSG00000130734	ATG4D	autophagy related 4D cysteine peptidase	0.360	4.83 × 10^−2^	Metabolism/Cell Proliferation/Regulation
ENSG00000256825	AC026786.1	novel protein	0.466	1.84 × 10^−2^	Metabolism/Cell Proliferation/Regulation
ENSG00000113240	CLK4	CDC like kinase 4	0.348	8.35 × 10^−4^	Metabolism/Cell Proliferation/Regulation
ENSG00000176444	CLK2	CDC like kinase 2	0.319	4.36 × 10^−5^	Metabolism/Cell Proliferation/Regulation
ENSG00000114735	HEMK1	HemK methyltransferase family member 1	0.313	9.02 × 10^−5^	Metabolism/Cell Proliferation/Regulation
ENSG00000135414	GDF11	growth differentiation factor 11	0.306	2.35 × 10^−6^	Metabolism/Cell Proliferation/Regulation
ENSG00000141013	GAS8	growth arrest specific 8	0.395	1.02 × 10^−2^	Metabolism/Cell Proliferation/Regulation
ENSG00000065268	WDR18	WD repeat domain 18	0.321	9.31 × 10^−3^	Metabolism/Cell Proliferation/Regulation
ENSG00000071246	VASH1	vasohibin 1	0.374	2.50 × 10^−4^	Metabolism/Cell Proliferation/Regulation
ENSG00000005007	UPF1	UPF1 RNA helicase and ATPase	0.275	4.60 × 10^−6^	Metabolism/Cell Proliferation/Regulation
ENSG00000126790	L3HYPDH	trans-L-3-hydroxyproline dehydratase	0.376	9.63 × 10^−3^	Metabolism/Cell Proliferation/Regulation
ENSG00000102871	TRADD	TNFRSF1A associated via death domain	0.399	2.50 × 10^−5^	Metabolism/Cell Proliferation/Regulation
ENSG00000146109	ABT1	activator of basal transcription 1	0.263	1.99 × 10^−2^	Metabolism/Cell Proliferation/Regulation
ENSG00000264343	NOTCH2NLA	notch 2 N-terminal like A	0.540	1.22 × 10^−2^	Metabolism/Cell Proliferation/Regulation
ENSG00000258674	AC011448.1	novel protein	0.432	3.06 × 10^−5^	Metabolism/Cell Proliferation/Regulation
ENSG00000149451	ADAM33	ADAM metallopeptidase domain 33	0.382	5.58 × 10^−20^	Metabolism/Cell Proliferation/Regulation
ENSG00000215041	NEURL4	neuralized E3 ubiquitin protein ligase 4	0.308	1.02 × 10^−4^	Metabolism/Cell Proliferation/Regulation
ENSG00000150401	DCUN1D2	defective in cullin neddylation 1 domain containing 2	0.282	1.23 × 10^−3^	Metabolism/Cell Proliferation/Regulation
ENSG00000251287	ALG1L2	ALG1 chitobiosyldiphosphodolichol beta-mannosyltransferase like 2	0.356	6.01 × 10^−7^	Metabolism/Cell Proliferation/Regulation
ENSG00000073169	SELENOO	selenoprotein O	0.364	1.01 × 10^−6^	Metabolism/Cell Proliferation/Regulation
ENSG00000234616	JRK	Jrk helix-turn-helix protein	0.374	3.11 × 10^−5^	Metabolism/Cell Proliferation/Regulation
ENSG00000105559	PLEKHA4	pleckstrin homology domain containing A4	0.442	5.49 × 10^−6^	Metabolism/Cell Proliferation/Regulation
ENSG00000007392	LUC7L	LUC7 like	0.322	8.83 × 10^−5^	Metabolism/Cell Proliferation/Regulation
ENSG00000131584	ACAP3	ArfGAP with coiled-coil, ankyrin repeat and PH domains 3	0.334	3.35 × 10^−7^	Metabolism/Cell Proliferation/Regulation
ENSG00000175137	SH3BP5L	SH3 binding domain protein 5 like	0.422	5.89 × 10^−6^	Metabolism/Cell Proliferation/Regulation
ENSG00000159692	CTBP1	C-terminal binding protein 1	0.342	2.13 × 10^−13^	Metabolism/Cell Proliferation/Regulation
ENSG00000173706	HEG1	heart development protein with EGF like domains 1	−0.277	5.78 × 10^−14^	Metablism/proliferation/regulation
ENSG00000155008	APOOL	apolipoprotein O like	−0.267	3.99 × 10^−3^	Metablism/proliferation/regulation
ENSG00000178385	PLEKHM3	pleckstrin homology domain containing M3	−0.264	1.23 × 10^−3^	Metablism/proliferation/regulation
ENSG0000017207	EIF2AK3	eukaryotic translation initiation factor 2 alpha kinase 3	−0.307	2.27 × 10^−5^	Metablism/proliferation/regulation
ENSG00000106392	C1GALT1	core 1 synthase, glycoprotein-N-acetylgalactosamine 3-beta-galactosyltransferase 1	−0.268	1.02 × 10^−2^	Metablism/proliferation/regulation
ENSG00000165195	PIGA	phosphatidylinositol glycan anchor biosynthesis class A	−0.726	1.08 × 10^−5^	Metablism/proliferation/regulation
ENSG00000155090	KLF10	Kruppel like factor 10	−1.136	1.11 × 10^−43^	Metablism/proliferation/regulation
ENSG00000144655	CSRNP1	cysteine and serine rich nuclear protein 1	−1.225	4.13 × 10^−35^	Metablism/proliferation/regulation
ENSG00000255112	CHMP1B	charged multivesicular body protein 1B	−0.526	1.54 × 10^−16^	Metablism/proliferation/regulation
ENSG00000179241	LDLRAD3	low density lipoprotein receptor class A domain containing 3	−0.272	1.15 × 10^−2^	Metablism/proliferation/regulation
ENSG00000108582	CPD	carboxypeptidase D	−0.308	1.62 × 10^−15^	Metablism/proliferation/regulation
ENSG00000176641	RNF152	ring finger protein 152	−0.326	7.13 × 10^−5^	Metablism/proliferation/regulation
ENSG00000136542	GALNT5	polypeptide N-acetylgalactosaminyltransferase 5	−0.264	7.28 × 10^−10^	Metablism/proliferation/regulation
ENSG00000203814	HIST2H2BF	histone cluster 2 H2B family member f	−0.332	3.21 × 10^−4^	Metablism/proliferation/regulation
ENSG00000162924	REL	REL proto-oncogene, NF-kB subunit	−0.591	5.43 × 10^−11^	Metablism/proliferation/regulation
ENSG00000160888	IER2	immediate early response 2	−1.137	6.07 × 10^−73^	Metablism/proliferation/regulation
ENSG00000087074	PPP1R15A	protein phosphatase 1 regulatory subunit 15A	−0.598	1.91 × 10^−38^	Metablism/proliferation/regulation
ENSG00000136158	SPRY2	sprouty RTK signaling antagonist 2	−0.356	8.30 × 10^−5^	Metablism/proliferation/regulation
ENSG00000137075	RNF38	ring finger protein 38	−0.266	3.06 × 10^−3^	Metablism/proliferation/regulation
ENSG00000166340	TPP1	tripeptidyl peptidase 1	−0.341	1.59 × 10^−9^	Metablism/proliferation/regulation
ENSG00000130164	LDLR	low density lipoprotein receptor	−0.282	9.25 × 10^−6^	Metablism/proliferation/regulation
ENSG00000112245	PTP4A1	protein tyrosine phosphatase 4A1	−0.303	1.01 × 10^−6^	Metablism/proliferation/regulation
ENSG00000119138	KLF9	Kruppel like factor 9	−0.409	7.12 × 10^−12^	Metablism/proliferation/regulation
ENSG00000134107	BHLHE40	basic helix-loop-helix family member e40	−0.597	7.53 × 10^−11^	Metablism/proliferation/regulation
ENSG00000115520	COQ10B	coenzyme Q10B	−0.414	2.48 × 10^−7^	Metablism/proliferation/regulation
ENSG00000049323	LTBP1	latent transforming growth factor beta binding protein 1	−0.308	1.58 × 10^−15^	Metablism/proliferation/regulation
ENSG00000138166	DUSP5	dual specificity phosphatase 5	−0.738	6.55 × 10^−23^	Metablism/proliferation/regulation
ENSG00000130513	GDF15	growth differentiation factor 15	−0.915	1.32 × 10^−35^	Metablism/proliferation/regulation
ENSG00000119986	AVPI1	arginine vasopressin induced 1	−0.734	1.12 × 10^−21^	Metablism/proliferation/regulation
ENSG00000157168	NRG1	neuregulin 1	−0.295	2.04 × 10^−7^	Metablism/proliferation/regulation
ENSG00000166225	FRS2	fibroblast growth factor receptor substrate 2	−0.292	7.11 × 10^−5^	Metablism/proliferation/regulation
ENSG00000197081	IGF2R	insulin like growth factor 2 receptor	−0.264	3.17 × 10^−11^	Metablism/proliferation/regulation
ENSG00000136527	TRA2B	transformer 2 beta homolog	−0.421	2.33 × 10^−14^	Metablism/proliferation/regulation
ENSG00000283782	AC116366.3	novel protein	−0.435	5.49 × 10^−3^	Metablism/proliferation/regulation
ENSG00000164220	F2RL2	coagulation factor II thrombin receptor like 2	−0.303	1.82 × 10^−10^	Metablism/proliferation/regulation
ENSG00000185483	ROR1	receptor tyrosine kinase like orphan receptor 1	−0.272	2.88 × 10^−5^	Metablism/proliferation/regulation
ENSG00000080200	CRYBG3	crystallin beta-gamma domain containing 3	−0.468	3.60 × 10^−16^	Metablism/proliferation/regulation
ENSG00000162433	AK4	adenylate kinase 4	−0.266	1.58 × 10^−2^	Metablism/proliferation/regulation
ENSG00000178607	ERN1	endoplasmic reticulum to nucleus signaling 1	−0.334	4.90 × 10^−6^	Metablism/proliferation/regulation
ENSG00000004799	PDK4	pyruvate dehydrogenase kinase 4	−1.212	9.25 × 10^−10^	Metablism/proliferation/regulation
ENSG00000155816	FMN2	formin 2	−0.275	3.38 × 10^−6^	Metablism/proliferation/regulation
ENSG00000132475	H3F3B	H3 histone family member 3B	−0.400	6.12 × 10^−11^	Metablism/proliferation/regulation
ENSG00000184260	HIST2H2AC	histone cluster 2 H2A family member c	−0.362	6.93 × 10^−3^	Metablism/proliferation/regulation
ENSG00000272196	HIST2H2AA4	histone cluster 2 H2A family member a4	−0.754	4.41 × 10^−2^	Metablism/proliferation/regulation
ENSG00000105835	NAMPT	nicotinamide phosphoribosyltransferase	−0.700	1.23 × 10^−34^	Metablism/proliferation/regulation
ENSG00000143384	MCL1	MCL1 apoptosis regulator, BCL2 family member	−0.436	1.54 × 10^−22^	Metablism/proliferation/regulation
ENSG00000067082	KLF6	Kruppel like factor 6	−0.645	2.26 × 10^−44^	Metablism/proliferation/regulation
ENSG00000177606	JUN	Jun proto-oncogene, AP-1 transcription factor subunit	−1.114	2.64 × 10^−153^	Metablism/proliferation/regulation
ENSG00000153936	HS2ST1	heparan sulfate 2-O-sulfotransferase 1	−0.264	4.02 × 10^−4^	Metablism/proliferation/regulation
ENSG00000168621	GDNF	glial cell derived neurotrophic factor	−0.518	3.74 × 10^−7^	Metablism/proliferation/regulation
ENSG00000175592	FOSL1	FOS like 1, AP-1 transcription factor subunit	−0.568	2.02 × 10^−19^	Metablism/proliferation/regulation
ENSG00000108306	FBXL20	F-box and leucine rich repeat protein 20	−0.264	6.71 × 10^−5^	Metablism/proliferation/regulation
ENSG00000174010	KLHL15	kelch like family member 15	−0.339	8.75 × 10^−3^	Metablism/proliferation/regulation
ENSG00000118263	KLF7	Kruppel like factor 7	−0.415	2.68 × 10^−11^	Metablism/proliferation/regulation
ENSG00000144959	NCEH1	neutral cholesterol ester hydrolase 1	−0.401	1.07 × 10^−5^	Metablism/proliferation/regulation
ENSG00000167470	MIDN	midnolin	−0.473	1.19 × 10^−15^	Metablism/proliferation/regulation
ENSG00000069667	RORA	RAR related orphan receptor A	−0.292	1.71 × 10^−3^	Metablism/proliferation/regulation
ENSG00000075213	SEMA3A	semaphorin 3A	−0.321	2.38 × 10^−5^	Metablism/proliferation/regulation
ENSG00000176542	USF3	upstream transcription factor family member 3	−0.359	2.91 × 10^−7^	Metablism/proliferation/regulation
ENSG00000100354	TNRC6B	trinucleotide repeat containing adaptor 6B	−0.265	3.40 × 10^−10^	Metablism/proliferation/regulation
ENSG00000185650	ZFP36L1	ZFP36 ring finger protein like 1	−0.377	3.07 × 10^−20^	Metablism/proliferation/regulation
ENSG00000122641	INHBA	inhibin subunit beta A	−0.301	9.16 × 10^−5^	Metablism/proliferation/regulation
ENSG00000092969	TGFB2	transforming growth factor beta 2	−0.309	7.30 × 10^−3^	Metablism/proliferation/regulation
ENSG00000165997	ARL5B	ADP ribosylation factor like GTPase 5B	−0.828	2.28 × 10^−25^	Metablism/proliferation/regulation
ENSG00000221869	CEBPD	CCAAT enhancer binding protein delta	−0.326	1.21 × 10^−4^	Metablism/proliferation/regulation
ENSG00000136731	UGGT1	UDP-glucose glycoprotein glucosyltransferase 1	−0.366	1.03 × 10^−17^	Metablism/proliferation/regulation
ENSG00000148737	TCF7L2	transcription factor 7 like 2	−0.339	2.90 × 10^−6^	Metablism/proliferation/regulation
ENSG00000152377	SPOCK1	SPARC (osteonectin), cwcv and kazal like domains proteoglycan 1	−0.323	7.57 × 10^−14^	Metablism/proliferation/regulation
ENSG00000124813	RUNX2	RUNX family transcription factor 2	−0.313	1.99 × 10^−8^	Metablism/proliferation/regulation
ENSG00000143190	POU2F1	POU class 2 homeobox 1	−0.331	1.05 × 10^−4^	Metablism/proliferation/regulation
ENSG00000184384	MAML2	mastermind like transcriptional coactivator 2	−0.314	3.02 × 10^−12^	Metablism/proliferation/regulation
ENSG00000128342	LIF	LIF interleukin 6 family cytokine	−0.763	2.73 × 10^−10^	Metablism/proliferation/regulation
ENSG00000120616	EPC1	enhancer of polycomb homolog 1	−0.321	9.31 × 10^−6^	Metablism/proliferation/regulation
ENSG00000175197	DDIT3	DNA damage inducible transcript 3	−0.427	6.45 × 10^−6^	Metablism/proliferation/regulation
ENSG00000171681	ATF7IP	activating transcription factor 7 interacting protein	−0.277	9.25 × 10^−10^	Metablism/proliferation/regulation
ENSG00000162772	ATF3	activating transcription factor 3	−2.877	3.76 × 10^−175^	Metablism/proliferation/regulation
ENSG00000003989	SLC7A2	solute carrier family 7 member2	−0.359	4.70 × 10^−2^	Metablism/proliferation/regulation
ENSG00000131389	SLC6A6	solute carrier family 6 member6	−0.269	2.71 × 10^−7^	Metablism/proliferation/regulation
ENSG00000059804	SLC2A3	solute carrier family 2 member3	−1.000	6.82 × 10^−60^	Metablism/proliferation/regulation
ENSG00000155850	SLC26A2	solute carrier family 26 member2	−0.266	1.54 × 10^−4^	Metablism/proliferation/regulation
ENSG00000118596	SLC16A7	solute carrier family 16 member7	−0.389	1.14 × 10^−8^	Metablism/proliferation/regulation
ENSG00000163660	CCNL1	cyclin L1	−1.027	4.70 × 10^−78^	Metablism/proliferation/regulation
ENSG00000182263	FIGN	fidgetin, microtubule severing factor	−0.360	2.43 × 10^−3^	Metablism/proliferation/regulation
ENSG00000137331	IER3	immediate early response 3	−0.884	1.62 × 10^−10^	Metablism/proliferation/regulation
ENSG00000143878	RHOB	ras homolog family member B	−1.048	2.20 × 10^−87^	Metablism/proliferation/regulation
ENSG00000123975	CKS2	CDC28 protein kinase regulatory subunit 2	−0.448	1.76 × 10^−7^	Metablism/proliferation/regulation
ENSG00000102781	KATNAL1	katanin catalytic subunit A1 like 1	−0.274	3.80 × 10^−5^	Metablism/proliferation/regulation
ENSG00000185621	LMLN	leishmanolysin like peptidase	−0.293	4.91 × 10^−3^	Metablism/proliferation/regulation
ENSG00000204131	NHSL2	NHS like 2	−0.292	1.07 × 10^−6^	Metablism/proliferation/regulation
ENSG00000134954	ETS1	ETS proto-oncogene 1, transcription factor	−0.298	8.39 × 10^−7^	Metablism/proliferation/regulation
ENSG00000105810	CDK6	cyclin dependent kinase 6	−0.306	2.07 × 10^−8^	Metablism/proliferation/regulation
ENSG00000204524	ZNF805	zinc finger protein 805	−0.310	4.10 × 10^−4^	Metablism/proliferation/regulation
ENSG00000197483	ZNF628	zinc finger protein 628	−0.439	1.94 × 10^−2^	Metablism/proliferation/regulation
ENSG00000197714	ZNF460	zinc finger protein 460	−0.305	3.43 × 10^−7^	Metablism/proliferation/regulation
ENSG00000130844	ZNF331	zinc finger protein 331	−0.277	5.71 × 10^−4^	Metablism/proliferation/regulation
ENSG00000285253	AC090517.4	zinc finger protein 280D	−0.444	2.01 × 10^−5^	Metablism/proliferation/regulation
ENSG00000185947	ZNF267	zinc finger protein 267	−0.284	3.90 × 10^−2^	Metablism/proliferation/regulation
ENSG00000091656	ZFHX4	zinc finger homeobox 4	−0.313	1.06 × 10^−9^	Metablism/proliferation/regulation
ENSG00000180776	ZDHHC20	zinc finger DHHC-type containing 20	−0.287	5.38 × 10^−7^	Metablism/proliferation/regulation
ENSG00000163874	ZC3H12A	zinc finger CCCH-type containing 12A	−0.274	3.54 × 10^−3^	Metablism/proliferation/regulation
ENSG00000173276	ZBTB21	zinc finger and BTB domain containing 21	−0.288	1.05 × 10^−5^	Metablism/proliferation/regulation
ENSG00000030419	IKZF2	IKAROS family zinc finger 2	−0.412	4.37 × 10^−4^	Metablism/proliferation/regulation
ENSG00000164122	ASB5	ankyrin repeat and SOCS box containing 5	−0.269	6.59 × 10^−3^	Metablism/proliferation/regulation
ENSG00000113448	PDE4D	phosphodiesterase 4D	−0.374	1.92 × 10^−3^	Metablism/proliferation/regulation
ENSG00000184588	PDE4B	phosphodiesterase 4B	−0.474	2.45 × 10^−7^	Metablism/proliferation/regulation
ENSG00000142892	PIGK	phosphatidylinositol glycan anchor biosynthesis class K	−0.264	9.96 × 10^−4^	Metablism/proliferation/regulation
ENSG00000118523	CCN2	cellular communication network factor 2	−0.313	3.41 × 10^−11^	Metablism/proliferation/regulation
ENSG00000112419	PHACTR2	phosphatase and actin regulator 2	−0.281	1.95 × 10^−6^	Metablism/proliferation/regulation
ENSG00000179094	PER1	period circadian regulator 1	−0.443	1.38 × 10^−11^	Metablism/proliferation/regulation
ENSG00000106460	TMEM106B	transmembrane protein 106B	−0.311	8.77 × 10^−8^	Metablism/proliferation/regulation
ENSG00000182752	PAPPA	pappalysin 1	−0.354	1.08 × 10^−26^	Metablism/proliferation/regulation
ENSG00000139496	NUP58	nucleoporin 58	−0.328	2.66 × 10^−7^	Metablism/proliferation/regulation
ENSG00000119508	NR4A3	nuclear receptor subfamily 4 group A member 3	−3.256	9.06 × 10^−169^	Metablism/proliferation/regulation
ENSG00000153234	NR4A2	nuclear receptor subfamily 4 group A member 2	−3.941	1.48 × 10^−78^	Metablism/proliferation/regulation
ENSG00000123358	NR4A1	nuclear receptor subfamily 4 group A member 1	−3.834	4.26 × 10^−130^	Metablism/proliferation/regulation
ENSG00000165030	NFIL3	nuclear factor, interleukin 3 regulated	−0.605	9.27 × 10^−18^	Metablism/proliferation/regulation
ENSG00000102908	NFAT5	nuclear factor of activated T cells 5	−0.312	1.58 × 10^−15^	Metablism/proliferation/regulation
ENSG00000162599	NFIA	nuclear factor I A	−0.312	1.38 × 10^−4^	Metablism/proliferation/regulation
ENSG00000141458	NPC1	NPC intracellular cholesterol transporter 1	−0.649	6.52 × 10^−53^	Metablism/proliferation/regulation
ENSG00000284057	AP001273.2	novel protein, C11orf54-MED17 readthrough	−0.356	2.99 × 10^−2^	Metablism/proliferation/regulation
ENSG00000138336	TET1	tet methylcytosine dioxygenase 1	−0.348	3.53 × 10^−5^	Metablism/proliferation/regulation
ENSG00000163960	UBXN7	UBX domain protein 7	−0.265	1.61 × 10^−8^	Metablism/proliferation/regulation
ENSG00000172059	KLF11	Kruppel like factor 11	−0.437	8.40 × 10^−8^	Metablism/proliferation/regulation
ENSG00000196233	LCOR	ligand dependent nuclear receptor corepressor	−0.382	2.07 × 10^−11^	Metablism/proliferation/regulation
ENSG00000004776	HSPB6	heat shock protein family B (small) member 6	−0.286	1.62 × 10^−3^	Metablism/proliferation/regulation
ENSG00000177570	SAMD12	sterile alpha motif domain containing 12	−0.334	7.44 × 10^−6^	Metablism/proliferation/regulation
ENSG00000197620	CXorf40A	chromosome X open reading frame 40A	−0.320	8.98 × 10^−3^	Metablism/proliferation/regulation
ENSG00000198142	SOWAHC	sosondowah ankyrin repeat domain family member C	−0.427	6.40 × 10^−7^	Metablism/proliferation/regulation
ENSG00000154175	ABI3BP	ABI family member 3 binding protein	−0.352	8.25 × 10^−6^	Metablism/proliferation/regulation
ENSG00000280987	MATR3	matrin 3	−0.323	4.21 × 10^−5^	Metablism/proliferation/regulation
ENSG00000164296	TIGD6	tigger transposable element derived 6	−0.272	2.37 × 10^−2^	Metablism/proliferation/regulation
ENSG00000205189	ZBTB10	zinc finger and BTB domain containing 10	−0.266	6.94 × 10^−3^	Metablism/proliferation/regulation
ENSG00000151967	SCHIP1	schwannomin interacting protein 1	−0.628	3.95 × 10^−3^	Metablism/proliferation/regulation
ENSG00000130962	PRRG1	proline rich and Gla domain 1	−0.307	1.52 × 10^−5^	Metablism/proliferation/regulation
ENSG00000169908	TM4SF1	transmembrane 4 L six family member 1	−0.945	7.67 × 10^−5^	Metablism/proliferation/regulation
ENSG00000236383	CCDC200	coiled-coil domain containing 200	−0.411	1.06 × 10^−16^	Metablism/proliferation/regulation
ENSG00000219481	NBPF1	NBPF member 1	−0.850	4.05 × 10^−8^	Metablism/proliferation/regulation
ENSG00000041982	TNC	tenascin C	−0.466	1.86 × 10^−26^	Metablism/proliferation/regulation
ENSG00000130702	LAMA5	laminin subunit alpha 5	−0.285	6.65 × 10^−5^	Metablism/proliferation/regulation
ENSG00000150907	FOXO1	forkhead box O1	−0.283	2.58 × 10^−3^	Metablism/proliferation/regulation
ENSG00000135842	NIBAN1	niban apoptosis regulator 1	−0.310	2.05 × 10^−7^	Metablism/proliferation/regulation
ENSG00000181827	RFX7	regulatory factor X7	−0.264	2.26 × 10^−7^	Metablism/proliferation/regulation
ENSG00000156030	ELMSAN1	ELM2 and Myb/SANT domain containing 1	−0.295	6.51 × 10^−8^	Metablism/proliferation/regulation
ENSG00000167244	IGF2	insulin like growth factor 2	−0.409	6.22 × 10^−3^	Metablism/proliferation/regulation
ENSG00000152409	JMY	junction mediating and regulatory protein, p53 cofactor	−0.265	4.94 × 10^−6^	Metablism/proliferation/regulation
ENSG00000059728	MXD1	MAX dimerization protein 1	−0.392	1.74 × 10^−3^	Metablism/proliferation/regulation
ENSG00000170653	ATF7	activating transcription factor 7	−0.340	1.15 × 10^−8^	Metablism/proliferation/regulation
ENSG00000118922	KLF12	Kruppel like factor 12	−0.272	3.37 × 10^−4^	Metablism/proliferation/regulation
ENSG00000158711	ELK4	ETS transcription factor ELK4	−0.266	1.36 × 10^−4^	Metablism/proliferation/regulation
ENSG00000095951	HIVEP1	human immunodeficiency virus type I enhancer binding protein 1	−0.305	1.59 × 10^−6^	Metablism/proliferation/regulation
ENSG00000119314	PTBP3	polypyrimidine tract binding protein 3	−0.309	1.49 × 10^−7^	Metablism/proliferation/regulation
ENSG00000173889	PHC3	polyhomeotic homolog 3	−0.265	3.70 × 10^−10^	Metablism/proliferation/regulation
ENSG00000099250	NRP1	neuropilin 1	−0.302	1.84 × 10^−12^	Metablism/proliferation/regulation
ENSG00000091409	ITGA6	integrin subunit alpha 6	−0.277	4.73 × 10^−4^	Metablism/proliferation/regulation
ENSG00000148841	ITPRIP	inositol 1,4,5-trisphosphate receptor interacting protein	−0.961	1.51 × 10^−44^	Metablism/proliferation/regulation
ENSG00000118503	TNFAIP3	TNF alpha induced protein 3	−0.658	3.99 × 10^−25^	Metablism/proliferation/regulation
ENSG00000277075	HIST1H2AE	histone cluster 1 H2A family member e	−0.403	2.84 × 10^−4^	Metablism/proliferation/regulation
ENSG00000173611	SCAI	suppressor of cancer cell invasion	−0.281	3.22 × 10^−3^	Metablism/proliferation/regulation
ENSG00000197594	ENPP1	ectonucleotide pyrophosphatase/phosphodiesterase 1	−0.293	1.62 × 10^−4^	Metablism/proliferation/regulation
ENSG00000213064	SFT2D2	SFT2 domain containing 2	−0.363	3.39 × 10^−11^	Metablism/proliferation/regulation
ENSG00000184897	H1FX	H1 histone family member X	−0.307	1.90 × 10^−4^	Metablism/proliferation/regulation
ENSG00000168298	HIST1H1E	histone cluster 1 H1 family member e	−0.314	2.88 × 10^−3^	Metablism/proliferation/regulation
ENSG00000163125	RPRD2	regulation of nuclear pre-mRNA domain containing 2	−0.265	1.20 × 10^−6^	Metablism/proliferation/regulation
ENSG00000148773	MKI67	marker of proliferation Ki-67	−0.339	6.61 × 10^−3^	Metablism/proliferation/regulation
ENSG00000164307	ERAP1	endoplasmic reticulum aminopeptidase 1	−0.280	9.38 × 10^−7^	Metablism/proliferation/regulation
ENSG00000076770	MBNL3	muscleblind like splicing regulator 3	−0.269	1.35 × 10^−2^	Metablism/proliferation/regulation
ENSG00000139636	LMBR1L	limb development membrane protein 1 like	0.349	3.97 × 10^−5^	Immune response
ENSG00000107281	NPDC1	neural proliferation, differentiation and control 1	0.279	3.67 × 10^−4^	Immune response
ENSG00000121716	PILRB	paired immunoglobin like type 2 receptor beta	0.512	3.18 × 10^−3^	Immune response
ENSG00000160360	GPSM1	G protein signaling modulator 1	0.316	1.77 × 10^−9^	Immune response
ENSG00000104522	TSTA3	tissue specific transplantation antigen P35B	0.289	1.26 × 10^−2^	Immune response
ENSG00000214279	SCART1	scavenger receptor family member expressed on T cells 1	0.588	1.35 × 10^−7^	Immune response
ENSG00000169228	RAB24	RAB24, member RAS oncogene family	0.286	3.64 × 10^−3^	Immune response
ENSG0000010877	DHX58	DExH-box helicase 58	0.361	1.47 × 10^−3^	Immune response
ENSG00000160072	ATAD3B	ATPase family AAA domain containing 3B	0.276	4.14 × 10^−3^	Immune response
ENSG00000112715	VEGFA	vascular endothelial growth factor A	0.263	3.33 × 10^−4^	Immune response
ENSG00000110719	TCIRG1	T cell immune regulator 1, ATPase H+ transporting V0 subunit a3	0.278	2.19 × 10^−6^	Immune response
ENSG00000204104	TRAF3IP1	TRAF3 interacting protein 1	0.313	2.78 × 10^−4^	Immune response
ENSG00000173531	MST1	macrophage stimulating 1	0.575	2.35 × 10^−7^	Immune response
ENSG00000273802	HIST1H2BG	histone cluster 1 H2B family member g	−0.339	5.97 × 10^−4^	Immune response
ENSG00000277224	HIST1H2BF	histone cluster 1 H2B family member f	−0.332	2.65 × 10^−2^	Immune response
ENSG00000135870	RC3H1	ring finger and CCCH-type domains 1	−0.270	4.05 × 10^−7^	Immune response
ENSG00000144802	NFKBIZ	NFKB inhibitor zeta	−1.500	1.51 × 10^−32^	Immune response
ENSG00000125730	C3	complement C3	−0.321	2.55 × 10^−17^	Immune response
ENSG00000184678	HIST2H2BE	histone cluster 2 H2B family member e	−0.615	1.72 × 10^−20^	Immune response
ENSG00000164949	GEM	GTP binding protein overexpressed in skeletal muscle	−1.068	2.18 × 10^−56^	Immune response
ENSG00000113494	PRLR	prolactin receptor	−0.372	3.94 × 10^−3^	Immune response
ENSG00000069702	TGFBR3	transforming growth factor beta receptor 3	−0.279	2.29 × 10^−11^	Immune response
ENSG00000121210	TMEM131L	transmembrane 131 like	−0.281	1.44 × 10^−2^	Immune response
ENSG00000120738	EGR1	early growth response 1	−2.195	0.00	Immune response
ENSG00000181722	ZBTB20	zinc finger and BTB domain containing 20	−0.511	4.80 × 10^−22^	Immune response
ENSG00000067992	PDK3	pyruvate dehydrogenase kinase 3	−0.294	2.41 × 10^−3^	Immune response
ENSG00000157764	BRAF	B-Raf proto-oncogene, serine/threonine kinase	−0.277	1.91 × 10^−5^	Immune response
ENSG00000249437	NAIP	NLR family apoptosis inhibitory protein	−0.318	7.20 × 10^−3^	Immune response
ENSG00000131016	AKAP12	A-kinase anchoring protein 12	−0.494	1.87 × 10^−20^	Immune response
ENSG00000106089	STX1A	syntaxin 1A	0.438	2.44 × 10^−2^	inflammatory response
ENSG00000185338	SOCS1	suppressor of cytokine signaling 1	0.351	6.41 × 10^−4^	inflammatory response
ENSG00000136244	IL6	interleukin 6	−2.301	4.19 × 10^−65^	inflammatory response
ENSG00000172292	CERS6	ceramide synthase 6	−0.286	5.45 × 10^−5^	inflammatory response
ENSG00000170345	FOS	Fos proto-oncogene, AP-1 transcription factor subunit	−2.530	1.24 × 10^−62^	inflammatory response
ENSG00000101966	XIAP	X-linked inhibitor of apoptosis	−0.276	3.96 × 10^−6^	inflammatory response
ENSG00000169429	CXCL8	C-X-C motif chemokine ligand 8	−0.572	3.49 × 10^−4^	inflammatory response
ENSG00000081041	CXCL2	C-X-C motif chemokine ligand 2	−1.212	3.32 × 10^−29^	inflammatory response
ENSG00000128016	ZFP36	ZFP36 ring finger protein	−1.515	1.97 × 10^−82^	inflammatory response
ENSG00000179954	SSC5D	scavenger receptor cysteine rich family member with 5 domains	−0.307	3.19 × 10^−7^	inflammatory response
ENSG00000100906	NFKBIA	NFKB inhibitor alpha	−0.406	3.02 × 10^−14^	inflammatory response
ENSG00000034152	MAP2K3	mitogen-activated protein kinase kinase 3	−0.402	3.85 × 10^−12^	inflammatory response
ENSG00000172817	CYP7B1	cytochrome P450 family 7 subfamily B member 1	−0.377	7.82 × 10^−3^	inflammatory response
ENSG00000173992	CCS	copper chaperone for superoxide dismutase	0.283	5.84 × 10^−3^	Response to oxidative stress
ENSG00000002016	RAD52	RAD52 homolog, DNA repair protein	0.297	1.52 × 10^−2^	Response to oxidative stress
ENSG00000172780	RAB43	RAB43, member RAS oncogene family	0.311	5.51 × 10^−3^	Response to oxidative stress
ENSG00000103245	CIAO3	cytosolic iron-sulfur assembly component 3	0.324	2.22 × 10^−2^	Response to oxidative stress
ENSG00000163516	ANKZF1	ankyrin repeat and zinc finger peptidyl tRNA hydrolase 1	0.292	1.87 × 10^−4^	Response to oxidative stress
ENSG00000140398	NEIL1	nei like DNA glycosylase 1	0.459	4.10 × 10^−5^	Response to oxidative stress
ENSG00000159363	ATP13A2	ATPase cation transporting 13A2	0.284	3.52 × 10^−2^	Response to oxidative stress
ENSG00000151012	SLC7A11	solute carrier family 7 member11	−0.327	2.98 × 10^−4^	Response to oxidative stress
ENSG00000172115	CYCS	cytochrome c, somatic	−0.323	3.86 × 10^−4^	Response to oxidative stress
ENSG00000146648	EGFR	epidermal growth factor receptor	−0.303	3.91 × 10^−19^	Response to oxidative stress
ENSG00000198840	MT-ND3	mitochondrially encoded NADH: ubiquinone oxidoreductase core subunit 3	−0.477	6.34 × 10^−14^	Response to oxidative stress
ENSG00000158615	PPP1R15B	protein phosphatase 1 regulatory subunit 15B	−0.631	1.15 × 10^−36^	Response to oxidative stress
ENSG00000109819	PPARGC1A	PPARG coactivator 1 alpha	−0.285	1.53 × 10^−2^	Response to oxidative stress
ENSG00000137449	CPEB2	cytoplasmic polyadenylation element binding protein 2	−0.313	9.29 × 10^−6^	Response to oxidative stress
ENSG00000198763	MT-ND2	mitochondrially encoded NADH: ubiquinone oxidoreductase core subunit 2	−0.463	3.74 × 10^−18^	Response to oxidative stress
ENSG00000198786	MT-ND5	mitochondrially encoded NADH: ubiquinone oxidoreductase core subunit 5	−0.493	3.60 × 10^−4^	Response to oxidative stress
ENSG00000198886	MT-ND4	mitochondrially encoded NADH: ubiquinone oxidoreductase core subunit 4	−0.556	4.78 × 10^−4^	Response to oxidative stress
ENSG00000198888	MT-ND1	mitochondrially encoded NADH: ubiquinone oxidoreductase core subunit 1	−0.511	1.15 × 10^−5^	Response to oxidative stress
ENSG00000212907	MT-ND4L	mitochondrially encoded NADH: ubiquinone oxidoreductase core subunit 4L	−0.467	2.65 × 10^−3^	Response to oxidative stress
ENSG00000132510	KDM6B	lysine demethylase 6B	−0.543	1.21 × 10^−18^	Response to oxidative stress
ENSG00000198712	MT-CO2	mitochondrially encoded cytochrome c oxidase II	−0.397	2.79 × 10^−3^	Response to oxidative stress
ENSG00000143322	ABL2	ABL proto-oncogene 2, non-receptor tyrosine kinase	−0.359	9.88 × 10^−14^	Response to oxidative stress
ENSG00000127528	KLF2	Kruppel like factor 2	−1.221	4.51 × 10^−14^	Response to oxidative stress
ENSG00000286268	AF196969.1	novel protein	0.712	9.99 × 10^−3^	Lipids
ENSG00000100288	CHKB	choline kinase beta	0.359	6.43 × 10^−5^	Lipids
ENSG00000132793	LPIN3	lipin 3	0.342	2.57 × 10^−8^	Lipids
ENSG00000243708	PLA2G4B	phospholipase A2 group IVB	0.565	1.56 × 10^−2^	Lipids
ENSG00000240303	ACAD11	acyl-CoA dehydrogenase family member 11	0.284	9.86 × 10^−3^	Lipids
ENSG00000072778	ACADVL	acyl-CoA dehydrogenase very long chain	0.268	7.46 × 10^−10^	Lipids
ENSG00000197943	PLCG2	phospholipase C gamma 2	0.310	5.49 × 10^−3^	Lipids
ENSG00000102125	TAZ	tafazzin	0.408	5.69 × 10^−4^	Lipids
ENSG00000111664	GNB3	G protein subunit beta 3	0.506	3.20 × 10^−2^	Lipids
ENSG00000123689	G0S2	G0/G1 switch 2	−0.470	4.89 × 10^−2^	Lipids
ENSG00000073756	PTGS2	prostaglandin-endoperoxide synthase 2	−2.220	1.77 × 10^−13^	Lipids
ENSG00000151176	PLBD2	phospholipase B domain containing 2	−0.446	2.90 × 10^−18^	Lipids
ENSG00000137642	SORL1	sortilin related receptor 1	−0.407	3.23 × 10^−2^	Lipids
ENSG00000172954	LCLAT1	lysocardiolipin acyltransferase 1	−0.319	9.16 × 10^−5^	Lipids
ENSG00000171132	PRKCE	protein kinase C epsilon	−0.267	4.13 × 10^−4^	Lipids
ENSG00000165029	ABCA1	ATP binding cassette subfamily A member 1	−0.299	1.82 × 10^−13^	Lipids
ENSG00000117600	PLPPR4	phospholipid phosphatase related 4	−0.311	1.43 × 10^−3^	Lipids
ENSG00000147872	PLIN2	perilipin 2	−0.340	8.19 × 10^−15^	Lipids
ENSG00000140474	ULK3	unc-51 like kinase 3	0.302	4.43 × 10^−3^	Communication
ENSG00000204084	INPP5B	inositol polyphosphate-5-phosphatase B	0.295	3.88 × 10^−5^	Communication
ENSG00000170153	RNF150	ring finger protein 150	−0.340	4.94 × 10^−9^	Communication
ENSG00000198753	PLXNB3	plexin B3	−0.372	1.56 × 10^−3^	Communication
ENSG00000184226	PCDH9	protocadherin 9	−0.439	2.47 × 10^−4^	Communication
ENSG00000253537	PCDHGA7	protocadherin gamma subfamily A, 7	−0.264	4.29 × 10^−3^	Communication
ENSG00000253873	PCDHGA11	protocadherin gamma subfamily A, 11	−0.302	9.94 × 10^−4^	Communication
ENSG00000189152	GRAPL	GRB2 related adaptor protein like	−0.270	1.68 × 10^−5^	Communication
ENSG00000187678	SPRY4	sprouty RTK signaling antagonist 4	−0.310	1.38 × 10^−4^	Communication
ENSG00000075568	TMEM131	transmembrane protein 131	−0.268	1.03 × 10^−8^	Communication
ENSG00000013588	GPRC5A	G protein-coupled receptor class C group 5 member A	−0.422	1.06 × 10^−10^	Communication
ENSG00000113441	LNPEP	leucyl and cystinyl aminopeptidase	−0.336	1.61 × 10^−9^	Communication
ENSG00000158258	CLSTN2	calsyntenin 2	−0.388	9.23 × 10^−10^	Communication
ENSG00000091129	NRCAM	neuronal cell adhesion molecule	−0.346	1.21 × 10^−3^	Communication
ENSG00000083857	FAT1	FAT atypical cadherin 1	−0.387	4.48 × 10^−33^	Communication
ENSG00000070018	LRP6	LDL receptor related protein 6	−0.284	1.70 × 10^−8^	Communication
ENSG00000253953	PCDHGB4	protocadherin gamma subfamily B, 4	−0.269	5.57 × 10^−4^	Communication
ENSG00000134982	APC	APC regulator of WNT signaling pathway	−0.282	2.50 × 10^−7^	Communication
ENSG00000112902	SEMA5A	semaphorin 5A	−0.341	1.31 × 10^−25^	Communication
ENSG00000218336	TENM3	teneurin transmembrane protein 3	−0.304	9.06 × 10^−4^	Communication
ENSG00000165124	SVEP1	sushi, von Willebrand factor type A, EGF and pentraxin domain containing 1	−0.481	7.92 × 10^−36^	Communication
ENSG00000123096	SSPN	sarcospan	−0.315	1.10 × 10^−4^	Communication
ENSG00000145012	LPP	LIM domain containing preferred translocation partner in lipoma	−0.319	2.31 × 10^−15^	Communication
ENSG00000067141	NEO1	neogenin 1	−0.272	2.21 × 10^−7^	Communication
ENSG00000196159	FAT4	FAT atypical cadherin 4	−0.318	2.88 × 10^−12^	Communication
ENSG00000143341	HMCN1	hemicentin 1	−0.421	2.95 × 10^−4^	Communication
ENSG00000078401	EDN1	endothelin 1	−0.429	3.86 × 10^−3^	Response to stimulus
ENSG00000135999	EPC2	enhancer of polycomb homolog 2	−0.284	7.13 × 10^−4^	Response to stimulus
ENSG00000283321	AC019117.3	novel protein	−0.337	5.32 × 10^−3^	Response to stimulus
ENSG00000110395	CBL	Cbl proto-oncogene	−0.303	2.57 × 10^−8^	Response to stimulus
ENSG00000114933	INO80D	INO80 complex subunit D	−0.303	6.43 × 10^−7^	Response to stimulus
ENSG00000130522	JUND	JunD proto-oncogene, AP-1 transcription factor subunit	−0.681	5.90 × 10^−30^	Response to stimulus
ENSG00000120129	DUSP1	dual specificity phosphatase 1	−2.123	1.05 × 10^−267^	Response to stimulus
ENSG00000148339	SLC25A25	solute carrier family 25 member25	−0.303	6.18 × 10^−4^	Response to stimulus
ENSG00000125740	FOSB	FosB proto-oncogene, AP-1 transcription factor subunit	−4.090	0.00	Response to stimulus
ENSG00000198727	MT-CYB	mitochondrially encoded cytochrome b	−0.320	3.34 × 10^−8^	Response to stimulus
ENSG00000177694	NAALADL2	N-acetylated alpha-linked acidic dipeptidase like 2	−0.320	1.80 × 10^−4^	Response to stimulus
ENSG00000184557	SOCS3	suppressor of cytokine signaling 3	−0.506	6.46 × 10^−7^	Response to stimulus
ENSG00000159167	STC1	stanniocalcin 1	−0.534	1.49 × 10^−4^	Response to stimulus
ENSG00000164603	BMT2	base methyltransferase of 25S rRNA 2 homolog	−0.323	1.14 × 10^−2^	Response to stimulus
ENSG00000181072	CHRM2	cholinergic receptor muscarinic 2	−0.273	2.59 × 10^−3^	Response to stimulus
ENSG00000132635	PCED1A	PC-esterase domain containing 1A	0.280	1.88 × 10^−3^	Unknown
ENSG00000186166	CCDC84	coiled-coil domain containing 84	0.282	7.88 × 10^−7^	Unknown
ENSG00000117616	RSRP1	arginine and serine rich protein 1	0.347	1.04 × 10^−9^	Unknown
ENSG00000214193	SH3D21	SH3 domain containing 21	0.284	1.03 × 10^−2^	Unknown
ENSG00000134884	ARGLU1	arginine and glutamate rich 1	0.289	9.33 × 10^−6^	Unknown
ENSG00000268350	FAM156A	family with sequence similarity 156 member A	0.449	9.92 × 10^−5^	Unknown
ENSG00000182700	IGIP	IgA inducing protein	0.400	3.33 × 10^−4^	Unknown
ENSG00000140365	COMMD4	COMM domain containing 4	0.319	1.83 × 10^−4^	Unknown
ENSG00000146067	FAM193B	family with sequence similarity 193 member B	0.285	7.51 × 10^−3^	Unknown
ENSG00000014914	MTMR11	myotubularin related protein 11	0.286	1.90 × 10^−2^	Unknown
ENSG00000148341	SH3GLB2	SH3 domain containing GRB2 like, endophilin B2	0.278	1.62 × 10^−4^	Unknown
ENSG00000188483	IER5L	immediate early response 5 like	0.325	1.54 × 10^−2^	Unknown
ENSG00000135637	CCDC142	coiled-coil domain containing 142	0.407	1.10 × 10^−4^	Unknown
ENSG00000213563	C8orf82	chromosome 8 open reading frame 82	0.275	5.43 × 10^−3^	Unknown
ENSG00000143469	SYT14	synaptotagmin 14	−0.272	8.77 × 10^−4^	Unknown
ENSG00000215472	RPL17-C18orf32	RPL17-C18orf32 readthrough	−0.298	6.96 × 10^−3^	Unknown

Notes: FC, fold change. The differentially expressed genes (DEGs) between the control and UVA-induced model were screened using |log_2_FC| > 1.2 and *P*-adjust < 0.05 as the thresholds. The DEGs were functionally annotated using Blast2go (Version 2.5) and goatools (Version 0.6.5) for gene ontology (GO) annotation and clustering analysis.

**Table 4 foods-12-00806-t004:** Functional characterization of up/downregulated differential expressed genes in H_2_O_2_-induced model.

Gene_Id	Gene Name	Gene Description	Log_2_FC	P-Adjust	Function
ENSG00000113494	PRLR	prolactin receptor	1.509	2.51 × 10^−4^	immune response
ENSG00000124882	EREG	epiregulin	1.029	1.75 × 10^−10^	immune response
ENSG00000128271	ADORA2A	adenosine A2a receptor	1.938	1.64 × 10^−10^	immune response
ENSG00000160224	AIRE	autoimmune regulator	1.490	1.30 × 10^−2^	immune response
ENSG00000169508	GPR183	G protein-coupled receptor 183	2.593	2.80 × 10^−3^	immune response
ENSG00000198805	PNP	purine nucleoside phosphorylase	1.397	1.63 × 10^−4^	immune response
ENSG00000105088	OLFM2	olfactomedin 2	1.794	6.44 × 10^−9^	immune response
ENSG00000243509	TNFRSF6B	TNF receptor superfamily member 6b	1.875	3.16 × 10^−5^	immune response
ENSG00000113070	HBEGF	heparin binding EGF like growth factor	1.237	4.11 × 10^−2^	immune response
ENSG00000122861	PLAU	plasminogen activator, urokinase	1.872	5.61 × 10^−35^	immune response
ENSG00000284337	AC013271.1	LIM and senescent cell antigen-like-containing domain protein 3	1.835	6.10 × 10^−4^	immune response
ENSG00000198535	C2CD4A	C2 calcium dependent domain containing 4A	2.260	2.90 × 10^−7^	immune response
ENSG00000237264	FTH1P11	ferritin heavy chain 1 pseudogene 11	4.468	4.70 × 10^−2^	immune response
ENSG00000219507	FTH1P8	ferritin heavy chain 1 pseudogene 8	6.317	1.68 × 10^−2^	immune response
ENSG00000160223	ICOSLG	inducible T-cell costimulator ligand	1.226	1.85 × 10^−2^	immune response
ENSG00000211772	TRBC2	T-cell receptor beta constant 2	1.185	1.65 × 10^−2^	immune response
ENSG00000110031	LPXN	leupaxin	2.068	9.53 × 10^−12^	immune response
ENSG00000028277	POU2F2	POU class 2 homeobox 2	1.447	7.46 × 10^−21^	immune response
ENSG00000076641	PAG1	phosphoprotein membrane anchor with glycosphingolipid microdomains 1	1.396	8.91 × 10^−7^	immune response
ENSG00000173457	PPP1R14B	protein phosphatase 1 regulatory inhibitor subunit 14B	1.476	2.28 × 10^−2^	immune response
ENSG00000131187	F12	coagulation factor XII	1.808	4.49 × 10^−2^	immune response
ENSG00000131015	ULBP2	UL16 binding protein 2	1.169	3.47 × 10^−2^	immune response
ENSG00000069702	TGFBR3	transforming growth factor beta receptor 3	−1.074	4.18 × 10^−3^	Immune response
ENSG00000138735	PDE5A	phosphodiesterase 5A	−1.754	3.76 × 10^−13^	Immune response
ENSG00000164764	SBSPON	somatomedin B and thrombospondin type 1 domain containing	−1.973	3.06 × 10^−3^	Immune response
ENSG00000089127	OAS1	2′-5′-oligoadenylate synthetase 1	−1.399	4.55 × 10^−4^	Immune response
ENSG00000066735	KIF26A	kinesin family member 26A	−1.743	1.91 × 10^−4^	Immune response
ENSG00000120885	CLU	clusterin	−1.346	1.07 × 10^−11^	Immune response
ENSG00000136869	TLR4	toll like receptor 4	−1.989	7.89 × 10^−6^	Immune response
ENSG00000164342	TLR3	toll like receptor 3	−2.064	2.85 × 10^−4^	Immune response
ENSG00000159403	C1R	complement C1r	−1.476	6.54 × 10^−10^	Immune response
ENSG00000182326	C1S	complement C1s	−1.331	6.39 × 10^−14^	Immune response
ENSG00000120899	PTK2B	protein tyrosine kinase 2 beta	−2.970	6.43 × 10^−31^	Immune response
ENSG00000130303	BST2	bone marrow stromal cell antigen 2	−1.112	7.88 × 10^−3^	Immune response
ENSG00000158270	COLEC12	collectin subfamily member 12	−1.382	2.11 × 10^−7^	Immune response
ENSG00000244731	C4A	complement C4A (Rodgers blood group)	−2.248	3.26 × 10^−6^	Immune response
ENSG00000224389	C4B	complement C4B (Chido blood group)	−1.643	2.40 × 10^−3^	Immune response
ENSG00000205403	CFI	complement factor I	−1.319	7.62 × 10^−5^	Immune response
ENSG00000243649	CFB	complement factor B	−2.221	1.40 × 10^−26^	Immune response
ENSG00000197766	CFD	complement factor D	−1.657	2.00 × 10^−2^	Immune response
ENSG00000000971	CFH	complement factor H	−1.106	1.44 × 10^−12^	Immune response
ENSG00000125730	C3	complement C3	−1.190	3.38 × 10^−4^	Immune response
ENSG00000185745	IFIT1	interferon induced protein with tetratricopeptide repeats 1	−1.393	2.81 × 10^−7^	Immune response
ENSG00000091972	CD200	CD200 molecule	−1.694	2.28 × 10^−3^	Immune response
ENSG00000111331	OAS3	2′-5′-oligoadenylate synthetase 3	−1.246	3.53 × 10^−2^	Immune response
ENSG00000101347	SAMHD1	SAM and HD domain containing deoxynucleoside triphosphate triphosphohydrolase 1	−1.048	1.57 × 10^−2^	Immune response
ENSG00000154188	ANGPT1	angiopoietin 1	−1.648	7.33 × 10^−9^	Immune response
ENSG00000107562	CXCL12	C-X-C motif chemokine ligand 12	−2.798	5.70 × 10^−42^	Immune response
ENSG00000142910	TINAGL1	tubulointerstitial nephritis antigen like 1	−3.217	7.01 × 10^−25^	Immune response
ENSG00000106537	TSPAN13	tetraspanin 13	1.228	3.73 × 10^−2^	Transport
ENSG00000108932	SLC16A6	solute carrier family 16 member 6	1.447	1.76 × 10^−8^	Transport
ENSG00000134955	SLC37A2	solute carrier family 37 member 2	1.488	6.86 × 10^−5^	Transport
ENSG00000139508	SLC46A3	solute carrier family 46 member3	1.007	3.41 × 10^−5^	Transport
ENSG00000101187	SLCO4A1	solute carrier organic anion transporter family member 4A1	1.354	1.02 × 10^−7^	Transport
ENSG00000188643	S100A16	S100 calcium binding protein A16	1.023	6.96 × 10^−3^	Transport
ENSG00000157445	CACNA2D3	calcium voltage-gated channel auxiliary subunit alpha2delta 3	1.659	1.37 × 10^−2^	Transport
ENSG00000069849	ATP1B3	ATPase Na+/K+ transporting subunit beta 3	1.100	3.43 × 10^−2^	Transport
ENSG00000166592	RRAD	RRAD, Ras related glycolysis inhibitor and calcium channel regulator	1.262	2.60 × 10^−3^	Transport
ENSG00000159199	ATP5G1	ATP synthase, H+ transporting, mitochondrial Fo complex subunit C1 (subunit 9)	1.329	4.87 × 10^−2^	Transport
ENSG00000166510	CCDC68	coiled-coil domain containing 68	1.072	1.66 × 10^−4^	Transport
ENSG00000157551	KCNJ15	potassium voltage-gated channel subfamily J member 15	1.993	1.26 × 10^−3^	Transport
ENSG00000157542	KCNJ6	potassium voltage-gated channel subfamily J member 6	2.391	3.83 × 10^−5^	Transport
ENSG00000173114	LRRN3	leucine rich repeat neuronal 3	1.515	7.22 × 10^−8^	Transport
ENSG00000171246	NPTX1	neuronal pentraxin 1	2.029	1.12 × 10^−7^	Transport
ENSG00000259863	SH3RF3-AS1	SH3RF3 antisense RNA 1	2.013	6.87 × 10^−7^	Transport
ENSG00000102287	GABRE	gamma-aminobutyric acid type A receptor epsilon subunit	−1.296	2.76 × 10^−6^	Transport
ENSG00000198691	ABCA4	ATP binding cassette subfamily A member 4	−1.740	1.07 × 10^−11^	Transport
ENSG00000221955	SLC12A8	solute carrier family 12 member 8	−1.424	5.81 × 10^−7^	Transport
ENSG00000162383	SLC1A7	solute carrier family 1 member 7	−1.588	1.17 × 10^−3^	Transport
ENSG00000146411	SLC2A12	solute carrier family 2 member 12	−1.807	1.49 × 10^−6^	Transport
ENSG00000169507	SLC38A11	solute carrier family 38 member 11	−2.235	7.62 × 10^−10^	Transport
ENSG00000139209	SLC38A4	solute carrier family 38 member 4	−1.638	1.79 × 10^−4^	Transport
ENSG00000138821	SLC39A8	solute carrier family 39 member 8	−1.053	6.51 × 10^−7^	Transport
ENSG00000138449	SLC40A1	solute carrier family 40 member 1	−1.479	7.54 × 10^−8^	Transport
ENSG00000013293	SLC7A14	solute carrier family 7 member 14	−1.849	2.46 × 10^−6^	Transport
ENSG00000181804	SLC9A9	solute carrier family 9 member A9	−1.018	4.45 × 10^−3^	Transport
ENSG00000168356	SCN11A	sodium voltage-gated channel alpha subunit 11	−1.637	4.59 × 10^−2^	Transport
ENSG00000144285	SCN1A	sodium voltage-gated channel alpha subunit 1	−4.979	1.95 × 10^−3^	Transport
ENSG00000151572	ANO4	anoctamin 4	−1.308	5.69 × 10^−11^	Transport
ENSG00000145362	ANK2	ankyrin 2	−2.002	3.71 × 10^−6^	Transport
ENSG00000111859	NEDD9	neural precursor cell expressed, developmentally down-regulated 9	−1.118	5.58 × 10^−6^	Transport
ENSG00000144645	OSBPL10	oxysterol binding protein like 10	−1.097	1.63 × 10^−4^	Transport
ENSG00000162687	KCNT2	potassium sodium-activated channel subfamily T member 2	−2.348	1.96 × 10^−3^	Transport
ENSG00000135643	KCNMB4	potassium calcium-activated channel subfamily M regulatory beta subunit 4	−1.242	1.75 × 10^−4^	Transport
ENSG00000143028	SYPL2	synaptophysin like 2	−1.197	2.42 × 10^−5^	Transport
ENSG00000089472	HEPH	hephaestin	−1.250	2.79 × 10^−6^	Transport
ENSG00000166473	PKD1L2	polycystin 1 like 2 (gene/pseudogene)	−1.796	1.07 × 10^−11^	Transport
ENSG00000143416	SELENBP1	selenium binding protein 1	−1.411	1.75 × 10^−19^	Transport
ENSG00000175538	KCNE3	potassium voltage-gated channel subfamily E regulatory subunit 3	−2.405	1.46 × 10^−5^	Transport
ENSG00000055732	MCOLN3	mucolipin 3	−1.027	1.32 × 10^−3^	Transport
ENSG00000172548	NIPAL4	NIPA like domain containing 4	−1.999	4.27 × 10^−2^	Transport
ENSG00000003137	CYP26B1	cytochrome P450 family 26 subfamily B member 1	1.217	9.43 × 10^−5^	Intracellular and extracellular matrix
ENSG00000058085	LAMC2	laminin subunit gamma 2	1.484	1.02 × 10^−4^	Intracellular and extracellular matrix
ENSG00000138316	ADAMTS14	ADAM metallopeptidase with thrombospondin type 1 motif 14	1.501	3.47 × 10^−5^	Intracellular and extracellular matrix
ENSG00000135346	CGA	glycoprotein hormones, alpha polypeptide	3.014	2.16 × 10^−5^	Intracellular and extracellular matrix
ENSG00000163347	CLDN1	claudin 1	1.065	1.97 × 10^−6^	Intracellular and extracellular matrix
ENSG00000123500	COL10A1	collagen type X alpha 1 chain	1.206	4.22 × 10^−2^	Intracellular and extracellular matrix
ENSG00000197467	COL13A1	collagen type XIII alpha 1 chain	1.299	2.71 × 10^−7^	Intracellular and extracellular matrix
ENSG00000080573	COL5A3	collagen type V alpha 3 chain	1.535	3.48 × 10^−7^	Intracellular and extracellular matrix
ENSG00000143127	ITGA10	integrin subunit alpha 10	1.596	9.82 × 10^−5^	Intracellular and extracellular matrix
ENSG00000137809	ITGA11	integrin subunit alpha 11	1.284	2.46 × 10^−3^	Intracellular and extracellular matrix
ENSG00000164171	ITGA2	integrin subunit alpha 2	1.034	9.75 × 10^−3^	Intracellular and extracellular matrix
ENSG00000005884	ITGA3	integrin subunit alpha 3	1.065	3.37 × 10^−6^	Intracellular and extracellular matrix
ENSG00000196611	MMP1	matrix metallopeptidase 1	2.713	3.61 × 10^−68^	Intracellular and extracellular matrix
ENSG00000166670	MMP10	matrix metallopeptidase 10	3.191	2.09 × 10^−4^	Intracellular and extracellular matrix
ENSG00000156103	MMP16	matrix metallopeptidase 16	1.229	8.60 × 10^−6^	Intracellular and extracellular matrix
ENSG00000149968	MMP3	matrix metallopeptidase 3	3.076	9.37 × 10^−44^	Intracellular and extracellular matrix
ENSG00000104368	PLAT	plasminogen activator, tissue type	1.692	1.03 × 10^−32^	Intracellular and extracellular matrix
ENSG00000196581	AJAP1	adherens junctions associated protein 1	1.032	8.97 × 10^−5^	Intracellular and extracellular matrix
ENSG00000182667	NTM	neurotrimin	2.160	2.03 × 10^−25^	Intracellular and extracellular matrix
ENSG00000261371	PECAM1	platelet and endothelial cell adhesion molecule 1	2.965	1.05 × 10^−7^	Intracellular and extracellular matrix
ENSG00000235649	MXRA5Y	matrix remodeling associated 5, Y-linked (pseudogene)	−1.849	3.32 × 10^−2^	Cellular and extracellular matrix
ENSG00000124749	COL21A1	collagen type XXI alpha 1 chain	−3.204	7.20 × 10^−22^	Cellular and extracellular matrix
ENSG00000049089	COL9A2	collagen type IX alpha 2 chain	−1.606	1.01 × 10^−3^	Cellular and extracellular matrix
ENSG00000060718	COL11A1	collagen type XI alpha 1 chain	−1.577	1.44 × 10^−4^	Cellular and extracellular matrix
ENSG00000187955	COL14A1	collagen type XIV alpha 1 chain	−2.288	8.17 × 10^−16^	Cellular and extracellular matrix
ENSG00000168077	SCARA3	scavenger receptor class A member 3	−1.422	1.51 × 10^−18^	Cellular and extracellular matrix
ENSG00000106823	ECM2	extracellular matrix protein 2	−2.612	3.11 × 10^−3^	Cellular and extracellular matrix
ENSG00000242600	MBL1P	mannose binding lectin 1, pseudogene	−1.941	1.06 × 10^−2^	Cellular and extracellular matrix
ENSG00000166482	MFAP4	microfibrillar associated protein 4	−1.966	1.51 × 10^−26^	Cellular and extracellular matrix
ENSG00000165272	AQP3	aquaporin 3 (Gill blood group)	−1.789	4.63 × 10^−11^	Cellular and extracellular matrix
ENSG00000138829	FBN2	fibrillin 2	−1.036	1.55 × 10^−2^	Cellular and extracellular matrix
ENSG00000123243	ITIH5	inter-alpha-trypsin inhibitor heavy chain family member 5	−1.312	7.45 × 10^−4^	Cellular and extracellular matrix
ENSG00000049540	ELN	elastin	−1.271	2.11 × 10^−4^	Cellular and extracellular matrix
ENSG00000183798	EMILIN3	elastin microfibril interfacer 3	−1.626	8.81 × 10^−4^	Cellular and extracellular matrix
ENSG00000091986	CCDC80	coiled-coil domain containing 80	−1.606	9.31 × 10^−7^	Cellular and extracellular matrix
ENSG00000135424	ITGA7	integrin subunit alpha 7	−1.024	1.29 × 10^−4^	Cellular and extracellular matrix
ENSG00000196569	LAMA2	laminin subunit alpha 2	−1.840	1.50 × 10^−5^	Cellular and extracellular matrix
ENSG00000132470	ITGB4	integrin subunit beta 4	−1.792	1.98 × 10^−11^	Cellular and extracellular matrix
ENSG00000105855	ITGB8	integrin subunit beta 8	−1.390	1.53 × 10^−6^	Cellular and extracellular matrix
ENSG00000205221	VIT	vitrin	−2.675	4.97 × 10^−3^	Cellular and extracellular matrix
ENSG00000122707	RECK	reversion inducing cysteine rich protein with kazal motifs	−1.066	7.99 × 10^−6^	Cellular and extracellular matrix
ENSG00000169908	TM4SF1	transmembrane 4 L six family member 1	1.509	4.14 × 10^−6^	Metabolism/Cell Proliferation/Regulation
ENSG00000157064	NMNAT2	nicotinamide nucleotide adenylyltransferase 2	1.159	5.85 × 10^−3^	Metabolism/Cell Proliferation/Regulation
ENSG00000165891	E2F7	E2F transcription factor 7	1.276	4.50 × 10^−5^	Metabolism/Cell Proliferation/Regulation
ENSG00000128965	CHAC1	ChaC glutathione specific gamma-glutamylcyclotransferase 1	1.667	1.34 × 10^−7^	Metabolism/Cell Proliferation/Regulation
ENSG00000185697	MYBL1	MYB proto-oncogene like 1	1.393	9.20 × 10^−12^	Metabolism/Cell Proliferation/Regulation
ENSG00000168405	CMAHP	cytidine monophospho-N-acetylneuraminic acid hydroxylase, pseudogene	1.120	1.46 × 10^−3^	Metabolism/Cell Proliferation/Regulation
ENSG00000137331	IER3	immediate early response 3	1.390	4.21 × 10^−7^	Metabolism/Cell Proliferation/Regulation
ENSG00000124802	EEF1E1	eukaryotic translation elongation factor 1 epsilon 1	1.325	2.52 × 10^−2^	Metabolism/Cell Proliferation/Regulation
ENSG00000170498	KISS1	KiSS-1 metastasis-suppressor	2.954	1.23 × 10^−2^	Metabolism/Cell Proliferation/Regulation
ENSG00000118523	CTGF	connective tissue growth factor	1.010	6.38 × 10^−4^	Metabolism/Cell Proliferation/Regulation
ENSG00000140379	BCL2A1	BCL2 related protein A1	1.898	2.86 × 10^−2^	Metabolism/Cell Proliferation/Regulation
ENSG00000169372	CRADD	CASP2 and RIPK1 domain containing adaptor with death domain	1.186	1.22 × 10^−2^	Metabolism/Cell Proliferation/Regulation
ENSG00000160013	PTGIR	prostaglandin I2 (prostacyclin) receptor (IP)	1.162	3.31 × 10^−18^	Metabolism/Cell Proliferation/Regulation
ENSG00000183691	NOG	noggin	2.225	9.39 × 10^−4^	Metabolism/Cell Proliferation/Regulation
ENSG00000124762	CDKN1A	cyclin dependent kinase inhibitor 1A	1.073	8.29 × 10^−11^	Metabolism/Cell Proliferation/Regulation
ENSG00000232977	LINC00327	long intergenic non-protein coding RNA 327	1.356	6.49 × 10^−7^	Metabolism/Cell Proliferation/Regulation
ENSG00000255073	ZFP91-CNTF	ZFP91-CNTF readthrough (NMD candidate)	1.284	3.17 × 10^−2^	Metabolism/Cell Proliferation/Regulation
ENSG00000179862	CITED4	Cbp/p300 interacting transactivator with Glu/Asp rich carboxy-terminal domain 4	2.519	6.13 × 10^−3^	Metabolism/Cell Proliferation/Regulation
ENSG00000128917	DLL4	delta like canonical Notch ligand 4	1.356	3.41 × 10^−9^	Metabolism/Cell Proliferation/Regulation
ENSG00000205683	DPF3	double PHD fingers 3	1.292	1.03 × 10^−7^	Metabolism/Cell Proliferation/Regulation
ENSG00000006468	ETV1	ETS variant 1	1.229	8.84 × 10^−14^	Metabolism/Cell Proliferation/Regulation
ENSG00000175592	FOSL1	FOS like 1, AP-1 transcription factor subunit	1.145	5.17 × 10^−6^	Metabolism/Cell Proliferation/Regulation
ENSG00000164379	FOXQ1	forkhead box Q1	1.395	1.63 × 10^−2^	Metabolism/Cell Proliferation/Regulation
ENSG00000134363	FST	follistatin	1.174	3.03 × 10^−2^	Metabolism/Cell Proliferation/Regulation
ENSG00000177283	FZD8	frizzled class receptor 8	1.495	7.35 × 10^−7^	Metabolism/Cell Proliferation/Regulation
ENSG00000114315	HES1	hes family bHLH transcription factor 1	2.075	1.27 × 10^−6^	Metabolism/Cell Proliferation/Regulation
ENSG00000188290	HES4	hes family bHLH transcription factor 4	1.563	1.16 × 10^−2^	Metabolism/Cell Proliferation/Regulation
ENSG00000164683	HEY1	hes related family bHLH transcription factor with YRPW motif 1	1.461	5.80 × 10^−4^	Metabolism/Cell Proliferation/Regulation
ENSG00000163909	HEYL	hes related family bHLH transcription factor with YRPW motif-like	1.649	5.79 × 10^−5^	Metabolism/Cell Proliferation/Regulation
ENSG00000187837	HIST1H1C	histone cluster 1 H1 family member c	1.075	2.17 × 10^−2^	Metabolism/Cell Proliferation/Regulation
ENSG00000137309	HMGA1	high mobility group AT-hook 1	1.299	2.87 × 10^−6^	Metabolism/Cell Proliferation/Regulation
ENSG00000115274	INO80B	INO80 complex subunit B	1.318	3.48 × 10^−2^	Metabolism/Cell Proliferation/Regulation
ENSG00000108551	RASD1	ras related dexamethasone induced 1	1.168	4.82 × 10^−4^	Metabolism/Cell Proliferation/Regulation
ENSG00000196460	RFX8	RFX family member 8, lacking RFX DNA binding domain	1.059	3.06 × 10^−4^	Metabolism/Cell Proliferation/Regulation
ENSG00000147509	RGS20	regulator of G protein signaling 20	1.409	2.83 × 10^−2^	Metabolism/Cell Proliferation/Regulation
ENSG00000095752	IL11	interleukin 11	1.511	1.46 × 10^−12^	Metabolism/Cell Proliferation/Regulation
ENSG00000133169	BEX1	brain expressed X-linked 1	1.579	1.14 × 10^−2^	Metabolism/Cell Proliferation/Regulation
ENSG00000168621	GDNF	glial cell derived neurotrophic factor	1.029	1.69 × 10^−6^	Metabolism/Cell Proliferation/Regulation
ENSG00000122641	INHBA	inhibin beta A subunit	1.188	6.30 × 10^−5^	Metabolism/Cell Proliferation/Regulation
ENSG00000254858	MPV17L2	MPV17 mitochondrial inner membrane protein like 2	1.060	4.39 × 10^−2^	Metabolism/Cell Proliferation/Regulation
ENSG00000169994	MYO7B	myosin VIIB	2.121	2.99 × 10^−2^	Metabolism/Cell Proliferation/Regulation
ENSG00000239672	NME1	NME/NM23 nucleoside diphosphate kinase 1	1.300	2.18 × 10^−2^	Metabolism/Cell Proliferation/Regulation
ENSG00000139289	PHLDA1	pleckstrin homology like domain family A member 1	1.113	4.84 × 10^−11^	Metabolism/Cell Proliferation/Regulation
ENSG00000181649	PHLDA2	pleckstrin homology like domain family A member 2	1.679	1.79 × 10^−4^	Metabolism/Cell Proliferation/Regulation
ENSG00000176641	RNF152	ring finger protein 152	1.800	7.61 × 10^−16^	Metabolism/Cell Proliferation/Regulation
ENSG00000154133	ROBO4	roundabout guidance receptor 4	2.382	2.98 × 10^−5^	Metabolism/Cell Proliferation/Regulation
ENSG00000197632	SERPINB2	serpin family B member 2	1.263	3.41 × 10^−5^	Metabolism/Cell Proliferation/Regulation
ENSG00000006327	TNFRSF12A	TNF receptor superfamily member 12A	1.168	5.04 × 10^−3^	Metabolism/Cell Proliferation/Regulation
ENSG00000130513	GDF15	growth differentiation factor 15	1.930	5.16 × 10^−10^	Metabolism/Cell Proliferation/Regulation
ENSG00000092345	DAZL	deleted in azoospermia like	1.516	2.85 × 10^−2^	Metabolism/Cell Proliferation/Regulation
ENSG00000268439	EMG1	EMG1, N1-specific pseudouridine methyltransferase	1.048	2.25 × 10^−2^	Metabolism/Cell Proliferation/Regulation
ENSG00000003249	DBNDD1	dysbindin domain containing 1	1.267	1.40 × 10^−3^	Metabolism/Cell Proliferation/Regulation
ENSG00000077348	EXOSC5	exosome component 5	1.275	2.99 × 10^−2^	Metabolism/Cell Proliferation/Regulation
ENSG00000162669	HFM1	HFM1, ATP dependent DNA helicase homolog	2.020	9.62 × 10^−3^	Metabolism/Cell Proliferation/Regulation
ENSG00000156265	MAP3K7CL	MAP3K7 C-terminal like	1.040	3.11 × 10^−3^	Metabolism/Cell Proliferation/Regulation
ENSG00000158747	NBL1	neuroblastoma 1, DAN family BMP antagonist	1.065	1.69 × 10^−5^	Metabolism/Cell Proliferation/Regulation
ENSG00000135919	SERPINE2	serpin family E member 2	1.245	7.42 × 10^−6^	Metabolism/Cell Proliferation/Regulation
ENSG00000128805	ARHGAP22	Rho GTPase activating protein 22	1.187	3.55 × 10^−11^	Metabolism/Cell Proliferation/Regulation
ENSG00000130702	LAMA5	laminin subunit alpha 5	−1.325	6.48 × 10^−3^	Metabolism/Cell Proliferation/Regulation
ENSG00000110455	ACCS	1-aminocyclopropane-1-carboxylate synthase homolog (inactive)	−1.435	2.85 × 10^−5^	Metabolism/Cell Proliferation/Regulation
ENSG00000138772	ANXA3	annexin A3	−1.922	8.29 × 10^−5^	Metabolism/Cell Proliferation/Regulation
ENSG00000151632	AKR1C2	aldo-keto reductase family 1 member C2	−1.608	1.64 × 10^−10^	Metabolism/Cell Proliferation/Regulation
ENSG00000170962	PDGFD	platelet derived growth factor D	−1.790	1.64 × 10^−43^	Metabolism/Cell Proliferation/Regulation
ENSG00000128606	LRRC17	leucine rich repeat containing 17	−1.332	5.81 × 10^−3^	Metabolism/Cell Proliferation/Regulation
ENSG00000102466	FGF14	fibroblast growth factor 14	−1.333	3.65 × 10^−4^	Metabolism/Cell Proliferation/Regulation
ENSG00000109339	MAPK10	mitogen-activated protein kinase 10	−1.329	1.44 × 10^−2^	Metabolism/Cell Proliferation/Regulation
ENSG00000108984	MAP2K6	mitogen-activated protein kinase kinase 6	−1.954	1.33 × 10^−9^	Metabolism/Cell Proliferation/Regulation
ENSG00000064687	ABCA7	ATP binding cassette subfamily A member 7	−1.603	1.39 × 10^−3^	Metabolism/Cell Proliferation/Regulation
ENSG00000173706	HEG1	heart development protein with EGF like domains 1	−1.524	1.95 × 10^−5^	Metabolism/Cell Proliferation/Regulation
ENSG00000174348	PODN	podocan	−1.891	4.35 × 10^−10^	Metabolism/Cell Proliferation/Regulation
ENSG00000003096	KLHL13	kelch like family member 13	−2.855	4.79 × 10^−7^	Metabolism/Cell Proliferation/Regulation
ENSG00000146021	KLHL3	kelch like family member 3	−1.014	2.99 × 10^−3^	Metabolism/Cell Proliferation/Regulation
ENSG00000145819	ARHGAP26	Rho GTPase activating protein 26	−2.155	2.00 × 10^−13^	Metabolism/Cell Proliferation/Regulation
ENSG00000139354	GAS2L3	growth arrest specific 2 like 3	−1.578	5.53 × 10^−3^	Metabolism/Cell Proliferation/Regulation
ENSG00000198796	ALPK2	alpha kinase 2	−1.018	2.28 × 10^−3^	Metabolism/Cell Proliferation/Regulation
ENSG00000082438	COBLL1	cordon-bleu WH2 repeat protein like 1	−1.576	1.97 × 10^−8^	Metabolism/Cell Proliferation/Regulation
ENSG00000176971	FIBIN	fin bud initiation factor homolog (zebrafish)	−1.382	9.25 × 10^−10^	Metabolism/Cell Proliferation/Regulation
ENSG00000183098	GPC6	glypican 6	−1.037	3.16 × 10^−6^	Metabolism/Cell Proliferation/Regulation
ENSG00000139263	LRIG3	leucine rich repeats and immunoglobulin like domains 3	−1.084	2.12 × 10^−7^	Metabolism/Cell Proliferation/Regulation
ENSG00000170500	LONRF2	LON peptidase N-terminal domain and ring finger 2	−2.063	3.13 × 10^−4^	Metabolism/Cell Proliferation/Regulation
ENSG00000163017	ACTG2	actin, gamma 2, smooth muscle, enteric	−1.699	5.71 × 10^−5^	Metabolism/Cell Proliferation/Regulation
ENSG00000111684	LPCAT3	lysophosphatidylcholine acyltransferase 3	−1.021	1.61 × 10^−6^	Metabolism/Cell Proliferation/Regulation
ENSG00000182575	NXPH3	neurexophilin 3	−1.451	7.04 × 10^−4^	Metabolism/Cell Proliferation/Regulation
ENSG00000243970	PPIEL	peptidylprolyl isomerase E like pseudogene	−1.019	1.09 × 10^−2^	Metabolism/Cell Proliferation/Regulation
ENSG00000113231	PDE8B	phosphodiesterase 8B	−1.113	6.47 × 10^−6^	Metabolism/Cell Proliferation/Regulation
ENSG00000185483	ROR1	receptor tyrosine kinase like orphan receptor 1	−1.344	6.02 × 10^−4^	Metabolism/Cell Proliferation/Regulation
ENSG00000187164	SHTN1	shootin 1	−1.080	2.96 × 10^−7^	Metabolism/Cell Proliferation/Regulation
ENSG00000177409	SAMD9L	sterile alpha motif domain containing 9 like	−1.003	1.80 × 10^−2^	Metabolism/Cell Proliferation/Regulation
ENSG00000196562	SULF2	sulfatase 2	−1.218	2.03 × 10^−5^	Metabolism/Cell Proliferation/Regulation
ENSG00000186854	TRABD2A	TraB domain containing 2A	−1.386	9.04 × 10^−9^	Metabolism/Cell Proliferation/Regulation
ENSG00000182179	UBA7	ubiquitin like modifier activating enzyme 7	−1.234	7.71 × 10^−6^	Metabolism/Cell Proliferation/Regulation
ENSG00000112303	VNN2	vanin 2	−3.082	2.31 × 10^−2^	Metabolism/Cell Proliferation/Regulation
ENSG00000115339	GALNT3	polypeptide N-acetylgalactosaminyltransferase 3	−1.284	2.38 × 10^−3^	Metabolism/Cell Proliferation/Regulation
ENSG00000167311	ART5	ADP-ribosyltransferase 5	−1.826	3.84 × 10^−2^	Metabolism/Cell Proliferation/Regulation
ENSG00000088882	CPXM1	carboxypeptidase X, M14 family member 1	−1.278	1.34 × 10^−6^	Metabolism/Cell Proliferation/Regulation
ENSG00000138180	CEP55	centrosomal protein 55	−1.272	1.45 × 10^−2^	Metabolism/Cell Proliferation/Regulation
ENSG00000064886	CHI3L2	chitinase 3 like 2	−1.354	2.34 × 10^−11^	Metabolism/Cell Proliferation/Regulation
ENSG00000115468	EFHD1	EF-hand domain family member D1	−1.752	5.70 × 10^−5^	Metabolism/Cell Proliferation/Regulation
ENSG00000106565	TMEM176B	transmembrane protein 176B	−1.559	4.58 × 10^−3^	Metabolism/Cell Proliferation/Regulation
ENSG00000143869	GDF7	growth differentiation factor 7	−2.047	2.63 × 10^−3^	Metabolism/Cell Proliferation/Regulation
ENSG00000139269	INHBE	inhibin beta E subunit	−2.466	7.03 × 10^−3^	Metabolism/Cell Proliferation/Regulation
ENSG00000105989	WNT2	Wnt family member 2	−3.133	2.67 × 10^−32^	Metabolism/Cell Proliferation/Regulation
ENSG00000139304	PTPRQ	protein tyrosine phosphatase, receptor type Q	−2.790	5.05 × 10^−7^	Metabolism/Cell Proliferation/Regulation
ENSG00000128045	RASL11B	RAS like family 11 member B	−2.577	1.04 × 10^−4^	Metabolism/Cell Proliferation/Regulation
ENSG00000138615	CILP	cartilage intermediate layer protein	−1.854	3.03 × 10^−2^	Metabolism/Cell Proliferation/Regulation
ENSG00000002933	TMEM176A	transmembrane protein 176A	−1.660	4.32 × 10^−4^	Metabolism/Cell Proliferation/Regulation
ENSG00000121858	TNFSF10	TNF superfamily member 10	−1.790	9.42 × 10^−4^	Metabolism/Cell Proliferation/Regulation
ENSG00000171462	DLK2	delta like non-canonical Notch ligand 2	−1.329	1.62 × 10^−2^	Metabolism/Cell Proliferation/Regulation
ENSG00000178662	CSRNP3	cysteine and serine rich nuclear protein 3	−2.392	8.91 × 10^−3^	Metabolism/Cell Proliferation/Regulation
ENSG00000090006	LTBP4	latent transforming growth factor beta binding protein 4	−1.122	1.82 × 10^−4^	Metabolism/Cell Proliferation/Regulation
ENSG00000100626	GALNT16	polypeptide N-acetylgalactosaminyltransferase 16	−1.780	2.60 × 10^−4^	Metabolism/Cell Proliferation/Regulation
ENSG00000168453	HR	HR, lysine demethylase and nuclear receptor corepressor	−2.686	1.55 × 10^−18^	Metabolism/Cell Proliferation/Regulation
ENSG00000054654	SYNE2	spectrin repeat containing nuclear envelope protein 2	−1.497	9.55 × 10^−5^	Metabolism/Cell Proliferation/Regulation
ENSG00000064692	SNCAIP	synuclein alpha interacting protein	−1.993	3.08 × 10^−20^	Metabolism/Cell Proliferation/Regulation
ENSG00000101938	CHRDL1	chordin like 1	−1.622	1.48 × 10^−4^	Metabolism/Cell Proliferation/Regulation
ENSG00000132846	ZBED3	zinc finger BED-type containing 3	−1.501	2.00 × 10^−13^	Metabolism/Cell Proliferation/Regulation
ENSG00000187764	SEMA4D	semaphorin 4D	−1.142	1.53 × 10^−4^	Metabolism/Cell Proliferation/Regulation
ENSG00000104081	BMF	Bcl2 modifying factor	−1.067	7.29 × 10^−7^	Metabolism/Cell Proliferation/Regulation
ENSG00000170214	ADRA1B	adrenoceptor alpha 1B	−3.324	4.07 × 10^−3^	Metabolism/Cell Proliferation/Regulation
ENSG00000151692	RNF144A	ring finger protein 144A	−1.343	1.09 × 10^−6^	Metabolism/Cell Proliferation/Regulation
ENSG00000111885	MAN1A1	mannosidase alpha class 1A member 1	−1.117	1.09 × 10^−10^	Metabolism/Cell Proliferation/Regulation
ENSG00000048740	CELF2	CUGBP Elav-like family member 2	−1.545	2.49 × 10^−2^	Metabolism/Cell Proliferation/Regulation
ENSG00000134245	WNT2B	Wnt family member 2B	−1.467	5.53 × 10^−6^	Metabolism/Cell Proliferation/Regulation
ENSG00000136859	ANGPTL2	angiopoietin like 2	−1.707	9.06 × 10^−22^	Metabolism/Cell Proliferation/Regulation
ENSG00000170624	SGCD	sarcoglycan delta	−1.300	8.69 × 10^−7^	Metabolism/Cell Proliferation/Regulation
ENSG00000135472	FAIM2	Fas apoptotic inhibitory molecule 2	−1.531	2.10 × 10^−6^	Metabolism/Cell Proliferation/Regulation
ENSG00000065717	TLE2	transducin like enhancer of split 2	−1.275	2.37 × 10^−5^	Metabolism/Cell Proliferation/Regulation
ENSG00000064205	WISP2	WNT1 inducible signaling pathway protein 2	−2.029	5.00 × 10^−3^	Metabolism/Cell Proliferation/Regulation
ENSG00000184347	SLIT3	slit guidance ligand 3	−1.835	7.13 × 10^−5^	Metabolism/Cell Proliferation/Regulation
ENSG00000137834	SMAD6	SMAD family member 6	−1.545	9.50 × 10^−5^	Metabolism/Cell Proliferation/Regulation
ENSG00000178573	MAF	MAF bZIP transcription factor	−1.077	2.32 × 10^−4^	Metabolism/Cell Proliferation/Regulation
ENSG00000248746	ACTN3	actinin alpha 3 (gene/pseudogene)	−1.987	1.15 × 10^−2^	Metabolism/Cell Proliferation/Regulation
ENSG00000154734	ADAMTS1	ADAM metallopeptidase with thrombospondin type 1 motif 1	−1.667	4.48 × 10^−6^	Metabolism/Cell Proliferation/Regulation
ENSG00000166106	ADAMTS15	ADAM metallopeptidase with thrombospondin type 1 motif 15	−1.395	2.83 × 10^−2^	Metabolism/Cell Proliferation/Regulation
ENSG00000158555	GDPD5	glycerophosphodiester phosphodiesterase domain containing 5	−2.275	1.82 × 10^−8^	Metabolism/Cell Proliferation/Regulation
ENSG00000173599	PC	pyruvate carboxylase	−1.123	4.07 × 10^−3^	Metabolism/Cell Proliferation/Regulation
ENSG00000122877	EGR2	early growth response 2	−2.430	3.34 × 10^−6^	Metabolism/Cell Proliferation/Regulation
ENSG00000179388	EGR3	early growth response 3	−1.946	1.08 × 10^−10^	Metabolism/Cell Proliferation/Regulation
ENSG00000052850	ALX4	ALX homeobox 4	−1.188	4.54 × 10^−6^	Metabolism/Cell Proliferation/Regulation
ENSG00000168502	MTCL1	microtubule crosslinking factor 1	−1.144	3.19 × 10^−4^	Metabolism/Cell Proliferation/Regulation
ENSG00000276600	RAB7B	RAB7B, member RAS oncogene family	−1.749	1.91 × 10^−6^	Metabolism/Cell Proliferation/Regulation
ENSG00000118849	RARRES1	retinoic acid receptor responder 1	−1.041	7.15 × 10^−3^	Metabolism/Cell Proliferation/Regulation
ENSG00000213626	LBH	limb bud and heart development	−1.890	1.54 × 10^−25^	Metabolism/Cell Proliferation/Regulation
ENSG00000169744	LDB2	LIM domain binding 2	−1.115	1.09 × 10^−9^	Metabolism/Cell Proliferation/Regulation
ENSG00000071282	LMCD1	LIM and cysteine rich domains 1	−1.747	1.80 × 10^−22^	Metabolism/Cell Proliferation/Regulation
ENSG00000163431	LMOD1	leiomodin 1	−1.048	1.94 × 10^−2^	Metabolism/Cell Proliferation/Regulation
ENSG00000170312	CDK1	cyclin dependent kinase 1	−1.104	4.72 × 10^−2^	Metabolism/Cell Proliferation/Regulation
ENSG00000123080	CDKN2C	cyclin dependent kinase inhibitor 2C	−1.247	1.27 × 10^−3^	Metabolism/Cell Proliferation/Regulation
ENSG00000066279	ASPM	abnormal spindle microtubule assembly	−1.597	7.20 × 10^−4^	Metabolism/Cell Proliferation/Regulation
ENSG00000116741	RGS2	regulator of G protein signaling 2	−1.023	7.35 × 10^−4^	Metabolism/Cell Proliferation/Regulation
ENSG00000128482	RNF112	ring finger protein 112	−1.258	5.99 × 10^−3^	Metabolism/Cell Proliferation/Regulation
ENSG00000137193	PIM1	Pim-1 proto-oncogene, serine/threonine kinase	−1.080	6.87 × 10^−11^	Metabolism/Cell Proliferation/Regulation
ENSG00000116133	DHCR24	24-dehydrocholesterol reductase	−1.042	2.13 × 10^−3^	Metabolism/Cell Proliferation/Regulation
ENSG00000148773	MKI67	marker of proliferation Ki-67	−1.255	4.78 × 10^−3^	Metabolism/Cell Proliferation/Regulation
ENSG00000092621	PHGDH	phosphoglycerate dehydrogenase	−1.408	5.84 × 10^−11^	Metabolism/Cell Proliferation/Regulation
ENSG00000108387	SEPT4	septin 4	−1.274	3.41 × 10^−2^	Metabolism/Cell Proliferation/Regulation
ENSG00000125354	SEPT6	septin 6	−1.003	4.95 × 10^−4^	Metabolism/Cell Proliferation/Regulation
ENSG00000136531	SCN2A	sodium voltage-gated channel alpha subunit 2	−1.828	1.68 × 10^−4^	Metabolism/Cell Proliferation/Regulation
ENSG00000139211	AMIGO2	adhesion molecule with Ig like domain 2	−1.127	1.26 × 10^−11^	Metabolism/Cell Proliferation/Regulation
ENSG00000120738	EGR1	early growth response 1	−1.344	4.50 × 10^−16^	Metabolism/Cell Proliferation/Regulation
ENSG00000165959	CLMN	calmin	−2.182	1.51 × 10^−2^	Metabolism/Cell Proliferation/Regulation
ENSG00000090539	CHRD	chordin	−1.001	4.24 × 10^−2^	Metabolism/Cell Proliferation/Regulation
ENSG00000106003	LFNG	LFNG O-fucosylpeptide 3-beta-N-acetylglucosaminyltransferase	−1.029	3.52 × 10^−2^	Metabolism/Cell Proliferation/Regulation
ENSG00000164920	OSR2	odd-skipped related transciption factor 2	−1.249	3.36 × 10^−8^	Metabolism/Cell Proliferation/Regulation
ENSG00000115252	PDE1A	phosphodiesterase 1A	−2.989	1.15 × 10^−5^	Metabolism/Cell Proliferation/Regulation
ENSG00000111341	MGP	matrix Gla protein	−1.673	2.70 × 10^−2^	Metabolism/Cell Proliferation/Regulation
ENSG00000187098	MITF	melanogenesis associated transcription factor	−1.741	1.10 × 10^−16^	Metabolism/Cell Proliferation/Regulation
ENSG00000169184	MN1	MN1 proto-oncogene, transcriptional regulator	−1.857	3.34 × 10^−5^	Metabolism/Cell Proliferation/Regulation
ENSG00000120278	PLEKHG1	pleckstrin homology and RhoGEF domain containing G1	−1.308	2.49 × 10^−4^	Metabolism/Cell Proliferation/Regulation
ENSG00000240694	PNMA2	paraneoplastic Ma antigen 2	−1.284	1.77 × 10^−3^	Metabolism/Cell Proliferation/Regulation
ENSG00000143125	PROK1	prokineticin 1	−1.352	1.04 × 10^−2^	Metabolism/Cell Proliferation/Regulation
ENSG00000106772	PRUNE2	prune homolog 2	−1.792	1.75 × 10^−10^	Metabolism/Cell Proliferation/Regulation
ENSG00000160801	PTH1R	parathyroid hormone 1 receptor	−1.442	3.49 × 10^−7^	Metabolism/Cell Proliferation/Regulation
ENSG00000107551	RASSF4	Ras association domain family member 4	−1.457	6.36 × 10^−6^	Metabolism/Cell Proliferation/Regulation
ENSG00000141068	KSR1	kinase suppressor of ras 1	−1.157	2.05 × 10^−3^	Metabolism/Cell Proliferation/Regulation
ENSG00000117155	SSX2IP	SSX family member 2 interacting protein	−1.190	2.62 × 10^−16^	Metabolism/Cell Proliferation/Regulation
ENSG00000162595	DIRAS3	DIRAS family GTPase 3	−1.371	6.86 × 10^−5^	Metabolism/Cell Proliferation/Regulation
ENSG00000179981	TSHZ1	teashirt zinc finger homeobox 1	−1.094	3.79 × 10^−5^	Metabolism/Cell Proliferation/Regulation
ENSG00000088756	ARHGAP28	Rho GTPase activating protein 28	−1.425	3.47 × 10^−23^	Metabolism/Cell Proliferation/Regulation
ENSG00000137727	ARHGAP20	Rho GTPase activating protein 20	−2.101	1.84 × 10^−5^	Metabolism/Cell Proliferation/Regulation
ENSG00000172346	CSDC2	cold shock domain containing C2	−2.139	2.20 × 10^−14^	Metabolism/Cell Proliferation/Regulation
ENSG00000181444	ZNF467	zinc finger protein 467	−1.210	2.94 × 10^−3^	Metabolism/Cell Proliferation/Regulation
ENSG00000206538	VGLL3	vestigial like family member 3	−1.052	9.47 × 10^−3^	Metabolism/Cell Proliferation/Regulation
ENSG00000112246	SIM1	single-minded family bHLH transcription factor 1	−1.095	1.58 × 10^−2^	Metabolism/Cell Proliferation/Regulation
ENSG00000173227	SYT12	synaptotagmin 12	−2.160	1.04 × 10^−2^	Metabolism/Cell Proliferation/Regulation
ENSG00000258818	RNASE4	ribonuclease A family member 4	−1.050	4.07 × 10^−3^	Metabolism/Cell Proliferation/Regulation
ENSG00000153993	SEMA3D	semaphorin 3D	−1.488	9.82 × 10^−5^	Metabolism/Cell Proliferation/Regulation
ENSG00000012171	SEMA3B	semaphorin 3B	−1.515	5.39 × 10^−11^	Metabolism/Cell Proliferation/Regulation
ENSG00000182175	RGMA	repulsive guidance molecule family member a	−1.225	8.69 × 10^−6^	Metabolism/Cell Proliferation/Regulation
ENSG00000120693	SMAD9	SMAD family member 9	−1.359	8.99 × 10^−5^	Metabolism/Cell Proliferation/Regulation
ENSG00000112984	KIF20A	kinesin family member 20A	−1.607	1.70 × 10^−4^	Metabolism/Cell Proliferation/Regulation
ENSG00000211448	DIO2	iodothyronine deiodinase 2	−1.249	7.48 × 10^−6^	Metabolism/Cell Proliferation/Regulation
ENSG00000197406	DIO3	iodothyronine deiodinase 3	−1.593	3.51 × 10^−2^	Metabolism/Cell Proliferation/Regulation
ENSG00000010932	FMO1	flavin containing monooxygenase 1	−3.430	1.99 × 10^−4^	Metabolism/Cell Proliferation/Regulation
ENSG00000076258	FMO4	flavin containing monooxygenase 4	−1.825	1.53 × 10^−9^	Metabolism/Cell Proliferation/Regulation
ENSG00000131781	FMO5	flavin containing monooxygenase 5	−1.218	4.49 × 10^−2^	Metabolism/Cell Proliferation/Regulation
ENSG00000162496	DHRS3	dehydrogenase/reductase 3	−1.489	1.03 × 10^−4^	Metabolism/Cell Proliferation/Regulation
ENSG00000159167	STC1	stanniocalcin 1	1.249	7.50 × 10^−13^	Response to stimulus
ENSG00000133805	AMPD3	adenosine monophosphate deaminase 3	1.087	8.42 × 10^−9^	Response to stimulus
ENSG00000143786	CNIH3	cornichon family AMPA receptor auxiliary protein 3	1.380	4.64 × 10^−7^	Response to stimulus
ENSG00000170373	CST1	cystatin SN	2.721	3.26 × 10^−6^	Response to stimulus
ENSG00000134769	DTNA	dystrobrevin alpha	1.118	4.90 × 10^−7^	Response to stimulus
ENSG00000120875	DUSP4	dual specificity phosphatase 4	1.040	3.70 × 10^−11^	Response to stimulus
ENSG00000060558	GNA15	G protein subunit alpha 15	1.725	2.90 × 10^−2^	Response to stimulus
ENSG00000181773	GPR3	G protein-coupled receptor 3	1.345	4.88 × 10^−4^	Response to stimulus
ENSG00000180875	GREM2	gremlin 2, DAN family BMP antagonist	1.351	8.05 × 10^−7^	Response to stimulus
ENSG00000148834	GSTO1	glutathione S-transferase omega 1	1.653	1.22 × 10^−3^	Response to stimulus
ENSG00000128285	MCHR1	melanin concentrating hormone receptor 1	2.294	3.65 × 10^−2^	Response to stimulus
ENSG00000169715	MT1E	metallothionein 1E	1.731	6.29 × 10^−3^	Response to stimulus
ENSG00000104490	NCALD	neurocalcin delta	1.428	1.58 × 10^−2^	Response to stimulus
ENSG00000171208	NETO2	neuropilin and tolloid like 2	1.272	2.96 × 10^−8^	Response to stimulus
ENSG00000154217	PITPNC1	phosphatidylinositol transfer protein, cytoplasmic 1	1.257	2.90 × 10^−5^	Response to stimulus
ENSG00000087494	PTHLH	parathyroid hormone like hormone	1.458	9.51 × 10^−5^	Response to stimulus
ENSG00000041353	RAB27B	RAB27B, member RAS oncogene family	1.385	2.72 × 10^−6^	Response to stimulus
ENSG00000136237	RAPGEF5	Rap guanine nucleotide exchange factor 5	1.659	2.41 × 10^−2^	Response to stimulus
ENSG00000181625	SLX1B	SLX1 homolog B, structure-specific endonuclease subunit	1.410	3.68 × 10^−3^	Response to stimulus
ENSG00000101463	SYNDIG1	synapse differentiation inducing 1	1.162	7.33 × 10^−4^	Response to stimulus
ENSG00000011347	SYT7	synaptotagmin 7	1.678	1.56 × 10^−2^	Response to stimulus
ENSG00000184292	TACSTD2	tumor associated calcium signal transducer 2	1.229	1.93 × 10^−2^	Response to stimulus
ENSG00000105825	TFPI2	tissue factor pathway inhibitor 2	1.940	2.13 × 10^−7^	Response to stimulus
ENSG00000178726	THBD	thrombomodulin	2.311	2.52 × 10^−4^	Response to stimulus
ENSG00000147573	TRIM55	tripartite motif containing 55	2.227	2.68 × 10^−6^	Response to stimulus
ENSG00000081181	ARG2	arginase 2	1.398	3.08 × 10^−2^	Response to stimulus
ENSG00000169627	BOLA2B	bolA family member 2B	1.289	3.05 × 10^−2^	Response to stimulus
ENSG00000144476	ACKR3	atypical chemokine receptor 3	−2.220	3.17 × 10^−14^	Response to stimulus
ENSG00000114698	PLSCR4	phospholipid scramblase 4	−1.101	6.91 × 10^−15^	Response to stimulus
ENSG00000137959	IFI44L	interferon induced protein 44 like	−1.666	1.47 × 10^−8^	Response to stimulus
ENSG00000069431	ABCC9	ATP binding cassette subfamily C member 9	−1.279	7.83 × 10^−6^	Response to stimulus
ENSG00000132561	MATN2	matrilin 2	−1.966	5.43 × 10^−21^	Response to stimulus
ENSG00000162551	ALPL	alkaline phosphatase, liver/bone/kidney	−2.494	8.25 × 10^−25^	Response to stimulus
ENSG00000184979	USP18	ubiquitin specific peptidase 18	−1.057	3.81 × 10^−2^	Response to stimulus
ENSG00000172264	MACROD2	MACRO domain containing 2	−2.645	6.64 × 10^−3^	Response to stimulus
ENSG00000250722	SELENOP	selenoprotein P	−2.805	1.50 × 10^−36^	Response to stimulus
ENSG00000069482	GAL	galanin and GMAP prepropeptide	2.335	3.68 × 10^−3^	Inflammatory process
ENSG00000151651	ADAM8	ADAM metallopeptidase domain 8	1.251	5.04 × 10^−7^	Inflammatory process
ENSG00000011201	ANOS1	anosmin 1	1.877	9.50 × 10^−16^	Inflammatory process
ENSG00000108691	CCL2	C-C motif chemokine ligand 2	1.339	5.92 × 10^−3^	Inflammatory process
ENSG00000164400	CSF2	colony stimulating factor 2	2.520	3.09 × 10^−5^	Inflammatory process
ENSG00000108342	CSF3	colony stimulating factor 3	2.372	3.36 × 10^−17^	Inflammatory process
ENSG00000163739	CXCL1	C-X-C motif chemokine ligand 1	1.612	1.83 × 10^−5^	Inflammatory process
ENSG00000081041	CXCL2	C-X-C motif chemokine ligand 2	1.713	8.35 × 10^−5^	Inflammatory process
ENSG00000163734	CXCL3	C-X-C motif chemokine ligand 3	1.149	7.49 × 10^−6^	Inflammatory process
ENSG00000163735	CXCL5	C-X-C motif chemokine ligand 5	2.107	3.74 × 10^−36^	Inflammatory process
ENSG00000169429	CXCL8	C-X-C motif chemokine ligand 8	2.340	3.78 × 10^−17^	Inflammatory process
ENSG00000123496	IL13RA2	interleukin 13 receptor subunit alpha 2	1.324	9.69 × 10^−6^	Inflammatory process
ENSG00000125538	IL1B	interleukin 1 beta	1.940	2.34 × 10^−14^	Inflammatory process
ENSG00000162892	IL24	interleukin 24	3.105	2.62 × 10^−16^	Inflammatory process
ENSG00000136244	IL6	interleukin 6	1.654	2.51 × 10^−6^	Inflammatory process
ENSG00000134070	IRAK2	interleukin 1 receptor associated kinase 2	1.446	1.34 × 10^−24^	Inflammatory process
ENSG00000119714	GPR68	G protein-coupled receptor 68	1.839	5.85 × 10^−6^	Inflammatory process
ENSG00000175040	CHST2	carbohydrate sulfotransferase 2	1.046	1.22 × 10^−9^	Inflammatory process
ENSG00000271605	MILR1	mast cell immunoglobulin like receptor 1	1.339	8.66 × 10^−4^	Inflammatory process
ENSG00000130203	APOE	apolipoprotein E	−1.092	4.88 × 10^−9^	Inflammatory process
ENSG00000196639	HRH1	histamine receptor H1	−1.494	6.54 × 10^−10^	Inflammatory process
ENSG00000172156	CCL11	C-C motif chemokine ligand 11	−1.416	4.72 × 10^−2^	Inflammatory process
ENSG00000168952	STXBP6	syntaxin binding protein 6	−1.283	3.40 × 10^−3^	Inflammatory process
ENSG00000129009	ISLR	immunoglobulin superfamily containing leucine rich repeat	−1.501	9.35 × 10^−20^	Inflammatory process
ENSG00000170989	S1PR1	sphingosine-1-phosphate receptor 1	−1.683	3.54 × 10^−8^	Inflammatory process
ENSG00000235568	NFAM1	NFAT activating protein with ITAM motif 1	−2.546	3.56 × 10^−4^	Inflammatory process
ENSG00000124212	PTGIS	prostaglandin I2 synthase	−2.422	1.80 × 10^−22^	Inflammatory process
ENSG00000185432	METTL7A	methyltransferase like 7A	−1.523	7.00 × 10^−10^	Inflammatory process
ENSG00000169432	SCN9A	sodium voltage-gated channel alpha subunit 9	−1.245	1.42 × 10^−4^	Inflammatory process
ENSG00000163701	IL17RE	interleukin 17 receptor E	−1.421	2.73 × 10^−2^	Inflammatory process
ENSG00000167191	GPRC5B	G protein-coupled receptor class C group 5 member B	−1.650	1.14 × 10^−9^	Inflammatory process
ENSG00000162645	GBP2	guanylate binding protein 2	−2.361	2.44 × 10^−34^	Inflammatory process
ENSG00000170323	FABP4	fatty acid binding protein 4	−4.620	4.47 × 10^−6^	Inflammatory process
ENSG00000162692	VCAM1	vascular cell adhesion molecule 1	−2.990	7.32 × 10^−9^	Inflammatory process
ENSG00000144730	IL17RD	interleukin 17 receptor D	−1.157	4.32 × 10^−2^	Inflammatory process
ENSG00000116106	EPHA4	EPH receptor A4	−1.815	2.62 × 10^−7^	Inflammatory process
ENSG00000182580	EPHB3	EPH receptor B3	−1.264	6.89 × 10^−5^	Inflammatory process
ENSG00000106123	EPHB6	EPH receptor B6	−1.367	1.33 × 10^−10^	Inflammatory process
ENSG00000238271	IFNWP19	interferon omega 1 pseudogene 19	−1.612	4.34 × 10^−2^	Inflammatory process
ENSG00000133056	PIK3C2B	phosphatidylinositol-4-phosphate 3-kinase catalytic subunit type 2 beta	−1.449	2.38 × 10^−2^	Inflammatory process
ENSG00000145675	PIK3R1	phosphoinositide-3-kinase regulatory subunit 1	−1.017	1.15 × 10^−2^	Inflammatory process
ENSG00000164099	PRSS12	protease, serine 12	−1.081	1.70 × 10^−4^	Inflammatory process
ENSG00000090889	KIF4A	kinesin family member 4A	−1.560	3.32 × 10^−2^	Inflammatory process
ENSG00000099998	GGT5	gamma-glutamyltransferase 5	1.009	9.64 × 10^−14^	Response to oxidative stress
ENSG00000148834	GSTO1	glutathione S-transferase omega 1	1.653	1.22 × 10^−3^	Response to oxidative stress
ENSG00000198763	MT-ND2	mitochondrially encoded NADH:ubiquinone oxidoreductase core subunit 2	1.118	3.23 × 10^−2^	Response to oxidative stress
ENSG00000086991	NOX4	NADPH oxidase 4	1.604	2.24 × 10^−5^	Response to oxidative stress
ENSG00000158125	XDH	xanthine dehydrogenase	1.317	2.20 × 10^−4^	Response to oxidative stress
ENSG00000122378	FAM213A	family with sequence similarity 213 member A	1.054	9.19 × 10^−7^	Response to oxidative stress
ENSG00000117592	PRDX6	peroxiredoxin 6	1.014	1.90 × 10^−2^	Response to oxidative stress
ENSG00000188906	LRRK2	leucine rich repeat kinase 2	−1.082	2.49 × 10^−2^	Response to oxidative stress
ENSG00000196139	AKR1C3	aldo-keto reductase family 1 member C3	−1.442	3.29 × 10^−7^	Response to oxidative stress
ENSG00000123453	SARDH	sarcosine dehydrogenase	−1.445	1.03 × 10^−2^	Response to oxidative stress
ENSG00000196616	ADH1B	alcohol dehydrogenase 1B (class I), beta polypeptide	−3.297	1.15 × 10^−47^	Response to oxidative stress
ENSG00000165092	ALDH1A1	aldehyde dehydrogenase 1 family member A1	−1.811	7.67 × 10^−56^	Response to oxidative stress
ENSG00000109819	PPARGC1A	PPARG coactivator 1 alpha	−1.062	2.51 × 10^−6^	Response to oxidative stress
ENSG00000155962	CLIC2	chloride intracellular channel 2	−1.595	2.65 × 10^−2^	Response to oxidative stress
ENSG00000240069	GPAA1P1	glycosylphosphatidylinositol anchor attachment 1 pseudogene 1	2.788	5.44 × 10^−4^	Communication
ENSG00000230596	GPAA1P2	glycosylphosphatidylinositol anchor attachment 1 pseudogene 2	2.360	2.70 × 10^−3^	Communication
ENSG00000115756	HPCAL1	hippocalcin like 1	1.011	3.56 × 10^−3^	Communication
ENSG00000180332	KCTD4	potassium channel tetramerization domain containing 4	2.199	5.29 × 10^−4^	Communication
ENSG00000166562	SEC11C	SEC11 homolog C, signal peptidase complex subunit	1.536	1.19 × 10^−3^	Communication
ENSG00000139364	TMEM132B	transmembrane protein 132B	1.465	1.59 × 10^−7^	Communication
ENSG00000249992	TMEM158	transmembrane protein 158 (gene/pseudogene)	3.051	2.25 × 10^−6^	Communication
ENSG00000157111	TMEM171	transmembrane protein 171	1.540	4.72 × 10^−2^	Communication
ENSG00000095209	TMEM38B	transmembrane protein 38B	1.152	1.87 × 10^−5^	Communication
ENSG00000179431	FJX1	four jointed box 1	1.432	2.05 × 10^−4^	Communication
ENSG00000143341	HMCN1	hemicentin 1	−2.093	1.63 × 10^−6^	Communication
ENSG00000144681	STAC	SH3 and cysteine rich domain	−3.241	3.30 × 10^−12^	Communication
ENSG00000171992	SYNPO	synaptopodin	−1.400	2.83 × 10^−3^	Communication
ENSG00000172403	SYNPO2	synaptopodin 2	−3.016	9.64 × 10^−14^	Communication
ENSG00000111452	ADGRD1	adhesion G protein-coupled receptor D1	−1.390	1.36 × 10^−3^	Communication
ENSG00000126016	AMOT	angiomotin	−2.152	4.31 × 10^−12^	Communication
ENSG00000113209	PCDHB5	protocadherin beta 5	−1.191	4.99 × 10^−2^	Communication
ENSG00000113212	PCDHB7	protocadherin beta 7	−1.428	1.70 × 10^−2^	Communication
ENSG00000163531	NFASC	neurofascin	−1.183	1.11 × 10^−2^	Communication
ENSG00000007944	MYLIP	myosin regulatory light chain interacting protein	−1.078	1.18 × 10^−6^	Communication
ENSG00000169760	NLGN1	neuroligin 1	−1.523	3.37 × 10^−5^	Communication
ENSG00000144857	BOC	BOC cell adhesion associated, oncogene regulated	−1.222	6.43 × 10^−5^	Communication
ENSG00000162849	KIF26B	kinesin family member 26B	−1.849	5.18 × 10^−6^	Communication
ENSG00000226237	GAS1RR	GAS1 adjacent regulatory RNA	−1.923	4.31 × 10^−3^	Communication
ENSG00000185565	LSAMP	limbic system-associated membrane protein	−1.733	2.28 × 10^−3^	Communication
ENSG00000158258	CLSTN2	calsyntenin 2	−2.251	3.02 × 10^−8^	Communication
ENSG00000163520	FBLN2	fibulin 2	−1.135	8.04 × 10^−5^	Communication
ENSG00000153707	PTPRD	protein tyrosine phosphatase, receptor type D	−1.549	1.33 × 10^−4^	Communication
ENSG00000175356	SCUBE2	signal peptide, CUB domain and EGF like domain containing 2	−1.335	8.72 × 10^−4^	Communication
ENSG00000182010	RTKN2	rhotekin 2	−2.248	3.82 × 10^−3^	Communication
ENSG00000068831	RASGRP2	RAS guanyl releasing protein 2	−1.188	1.22 × 10^−2^	Communication
ENSG00000171408	PDE7B	phosphodiesterase 7B	−1.071	1.94 × 10^−3^	Communication
ENSG00000103710	RASL12	RAS like family 12	−1.189	2.06 × 10^−6^	Communication
ENSG00000145934	TENM2	teneurin transmembrane protein 2	−1.527	6.39 × 10^−3^	Communication
ENSG00000166448	TMEM130	transmembrane protein 130	−2.267	5.15 × 10^−6^	Communication
ENSG00000178401	DNAJC22	DnaJ heat shock protein family (Hsp40) member C22	−1.357	1.14 × 10^−4^	Communication
ENSG00000185442	FAM174B	family with sequence similarity 174 member B	−1.141	2.79 × 10^−7^	Communication
ENSG00000177363	LRRN4CL	LRRN4 C-terminal like	−1.072	2.24 × 10^−5^	Communication
ENSG00000170153	RNF150	ring finger protein 150	−2.208	3.10 × 10^−13^	Communication
ENSG00000185561	TLCD2	TLC domain containing 2	−1.134	2.38 × 10^−5^	Communication
ENSG00000151690	MFSD6	major facilitator superfamily domain containing 6	−1.024	2.12 × 10^−7^	Communication
ENSG00000226887	ERVMER34-1	endogenous retrovirus group MER34 member 1, envelope	−2.372	5.16 × 10^−3^	Communication
ENSG00000005379	TSPOAP1	TSPO associated protein 1	−2.017	1.32 × 10^−5^	Communication
ENSG00000013297	CLDN11	claudin 11	−1.263	1.23 × 10^−5^	Communication
ENSG00000172348	RCAN2	regulator of calcineurin 2	−2.240	9.99 × 10^−8^	Communication
ENSG00000198542	ITGBL1	integrin subunit beta like 1	−1.045	1.55 × 10^−8^	Communication
ENSG00000113805	CNTN3	contactin 3	−1.285	1.94 × 10^−4^	Communication
ENSG00000165124	SVEP1	sushi, von Willebrand factor type A, EGF and pentraxin domain containing 1	−1.546	1.36 × 10^−4^	Communication
ENSG00000162804	SNED1	sushi, nidogen and EGF like domains 1	−1.163	8.26 × 10^−4^	Communication
ENSG00000187244	BCAM	basal cell adhesion molecule (Lutheran blood group)	−1.088	3.79 × 10^−5^	Communication
ENSG00000064309	CDON	cell adhesion associated, oncogene regulated	−2.033	2.00 × 10^−13^	Communication
ENSG00000095203	EPB41L4B	erythrocyte membrane protein band 4.1 like 4B	−1.389	6.09 × 10^−3^	Communication
ENSG00000157193	LRP8	LDL receptor related protein 8	1.448	9.21 × 10^−4^	Lipids(barrier function)
ENSG00000073756	PTGS2	prostaglandin-endoperoxide synthase 2	1.274	1.20 × 10^−8^	Lipids(barrier function)
ENSG00000123689	G0S2	G0/G1 switch 2	1.382	7.06 × 10^−5^	Lipids(barrier function)
ENSG00000176170	SPHK1	sphingosine kinase 1	1.014	2.42 × 10^−4^	Lipids(barrier function)
ENSG00000105499	PLA2G4C	phospholipase A2 group IVC	1.145	7.84 × 10^−11^	Lipids(barrier function)
ENSG00000101188	NTSR1	neurotensin receptor 1	1.895	2.44 × 10^−2^	Lipids(barrier function)
ENSG00000076555	ACACB	acetyl-CoA carboxylase beta	−1.405	3.86 × 10^−3^	Lipids(barrier function)
ENSG00000100342	APOL1	apolipoprotein L1	−1.841	2.88 × 10^−4^	Lipids(barrier function)
ENSG00000157399	ARSE	arylsulfatase E (chondrodysplasia punctata 1)	−1.230	1.09 × 10^−3^	Lipids(barrier function)
ENSG00000106484	MEST	mesoderm specific transcript	−1.829	6.91 × 10^−15^	Lipids(barrier function)
ENSG00000169255	B3GALNT1	beta-1,3-N-acetylgalactosaminyltransferase 1 (globoside blood group)	−1.071	9.26 × 10^−4^	Lipids(barrier function)
ENSG00000120915	EPHX2	epoxide hydrolase 2	−1.744	1.48 × 10^−3^	Lipids(barrier function)
ENSG00000169174	PCSK9	proprotein convertase subtilisin/kexin type 9	−2.368	3.78 × 10^−2^	Lipids(barrier function)
ENSG00000162407	PLPP3	phospholipid phosphatase 3	−1.408	1.06 × 10^−15^	Lipids(barrier function)
ENSG00000133321	RARRES3	retinoic acid receptor responder 3	−1.892	1.00 × 10^−2^	Lipids(barrier function)
ENSG00000137642	SORL1	sortilin related receptor 1	−1.848	2.32 × 10^−2^	Lipids(barrier function)
ENSG00000172296	SPTLC3	serine palmitoyltransferase long chain base subunit 3	−2.612	3.34 × 10^−5^	Lipids(barrier function)
ENSG00000129951	PLPPR3	phospholipid phosphatase related 3	−1.639	2.62 × 10^−2^	Lipids(barrier function)
ENSG00000134343	ANO3	anoctamin 3	−3.413	5.11 × 10^−5^	Lipids(barrier function)
ENSG00000172594	SMPDL3A	sphingomyelin phosphodiesterase acid like 3A	−1.331	9.50 × 10^−5^	Lipids(barrier function)
ENSG00000147465	STAR	steroidogenic acute regulatory protein	−1.107	4.63 × 10^−2^	Lipids(barrier function)
ENSG00000099204	ABLIM1	actin binding LIM protein 1	−1.299	2.25 × 10^−6^	Cytoskeleton organization
ENSG00000167549	CORO6	coronin 6	−1.256	4.91 × 10^−4^	Cytoskeleton organization
ENSG00000146122	DAAM2	dishevelled associated activator of morphogenesis 2	−1.158	3.78 × 10^−2^	Cytoskeleton organization
ENSG00000139734	DIAPH3	diaphanous related formin 3	−1.223	2.20 × 10^−2^	Cytoskeleton organization
ENSG00000082397	EPB41L3	erythrocyte membrane protein band 4.1 like 3	−1.296	2.38 × 10^−8^	Cytoskeleton organization
ENSG00000078018	MAP2	microtubule associated protein 2	−3.102	1.25 × 10^−11^	Cytoskeleton organization
ENSG00000116141	MARK1	microtubule affinity regulating kinase 1	−1.291	9.50 × 10^−5^	Cytoskeleton organization
ENSG00000133026	MYH10	myosin heavy chain 10	−1.297	2.58 × 10^−3^	Cytoskeleton organization
ENSG00000099864	PALM	paralemmin	−1.255	3.72 × 10^−9^	Cytoskeleton organization
ENSG00000118898	PPL	periplakin	−2.858	4.50 × 10^−16^	Cytoskeleton organization
ENSG00000126785	RHOJ	ras homolog family member J	−1.184	1.02 × 10^−2^	Cytoskeleton organization
ENSG00000168477	TNXB	tenascin XB	−1.431	1.19 × 10^−3^	Cytoskeleton organization
ENSG00000137285	TUBB2B	tubulin beta 2B class IIb	−1.460	4.75 × 10^−9^	Cytoskeleton organization
ENSG00000148700	ADD3	adducin 3	−1.009	2.68 × 10^−6^	Cytoskeleton organization
ENSG00000231789	PIK3CD-AS2	PIK3CD antisense RNA 2	−1.395	1.54 × 10^−2^	Cytoskeleton organization
ENSG00000152527	PLEKHH2	pleckstrin homology, MyTH4 and FERM domain containing H2	−1.017	4.84 × 10^−3^	Cytoskeleton organization
ENSG00000188783	PRELP	proline and arginine rich end leucine rich repeat protein	−1.594	2.61 × 10^−29^	Cytoskeleton organization
ENSG00000222047	C10orf55	chromosome 10 open reading frame 55	1.712	4.81 × 10^−3^	Unknown
ENSG00000163814	CDCP1	CUB domain containing protein 1	1.621	2.86 × 10^−17^	Unknown
ENSG00000154319	FAM167A	family with sequence similarity 167 member A	1.082	5.30 × 10^−10^	Unknown
ENSG00000166578	IQCD	IQ motif containing D	1.637	1.34 × 10^−6^	Unknown
ENSG00000253522	MIR3142HG	MIR3142 host gene	1.555	1.41 × 10^−2^	Unknown
ENSG00000164929	BAALC	brain and acute leukemia, cytoplasmic	1.215	1.50 × 10^−2^	Unknown
ENSG00000239704	CDRT4	CMT1A duplicated region transcript 4	3.443	2.85 × 10^−2^	Unknown
ENSG00000260549	MT1L	metallothionein 1L, pseudogene	2.582	6.86 × 10^−5^	Unknown
ENSG00000187193	MT1X	metallothionein 1X	1.349	4.34 × 10^−2^	Unknown
ENSG00000125148	MT2A	metallothionein 2A	2.408	4.51 × 10^−5^	Unknown
ENSG00000217930	PAM16	presequence translocase associated motor 16 homolog	1.400	3.06 × 10^−2^	Unknown
ENSG00000241749	RPSAP52	ribosomal protein SA pseudogene 52	1.471	8.60 × 10^−6^	Unknown
ENSG00000186577	SMIM29	small integral membrane protein 29	1.077	3.37 × 10^−2^	Unknown
ENSG00000157111	TMEM171	transmembrane protein 171	1.540	4.72 × 10^−2^	Unknown
ENSG00000185262	UBALD2	UBA like domain containing 2	1.329	2.38 × 10^−3^	Unknown
ENSG00000240476	LINC00973	long intergenic non-protein coding RNA 973	2.326	5.72 × 10^−3^	Unknown
ENSG00000257219	LINC02407	long intergenic non-protein coding RNA 2407	2.301	7.04 × 10^−3^	Unknown
ENSG00000235385	LINC02154	long intergenic non-protein coding RNA 2154	2.177	2.30 × 10^−3^	Unknown
ENSG00000164236	ANKRD33B	ankyrin repeat domain 33B	−1.240	3.89 × 10^−3^	Unknown
ENSG00000264230	ANXA8L1	annexin A8 like 1	−2.813	4.43 × 10^−4^	Unknown
ENSG00000198624	CCDC69	coiled-coil domain containing 69	−1.320	6.43 × 10^−13^	Unknown
ENSG00000055813	CCDC85A	coiled-coil domain containing 85A	−2.721	1.19 × 10^−2^	Unknown
ENSG00000105516	DBP	D-box binding PAR bZIP transcription factor	−1.355	3.37 × 10^−2^	Unknown
ENSG00000258498	DIO3OS	DIO3 opposite strand/antisense RNA (head to head)	−1.035	2.02 × 10^−3^	Unknown
ENSG00000229847	EMX2OS	EMX2 opposite strand/antisense RNA	−1.016	1.90 × 10^−4^	Unknown
ENSG00000133106	EPSTI1	epithelial stromal interaction 1	−1.184	3.04 × 10^−13^	Unknown
ENSG00000162636	FAM102B	family with sequence similarity 102 member B	−1.024	3.21 × 10^−4^	Unknown
ENSG00000255052	FAM66D	family with sequence similarity 66 member D	−1.599	4.85 × 10^−3^	Unknown
ENSG00000265962	GACAT2	gastric cancer associated transcript 2 (non-protein coding)	−1.788	7.03 × 10^−3^	Unknown
ENSG00000072163	LIMS2	LIM zinc finger domain containing 2	−1.501	3.09 × 10^−17^	Unknown
ENSG00000225783	MIAT	myocardial infarction associated transcript (non-protein coding)	−2.924	4.61 × 10^−11^	Unknown
ENSG00000220785	MTMR9LP	myotubularin related protein 9-like, pseudogene	−1.153	3.36 × 10^−3^	Unknown
ENSG00000183486	MX2	MX dynamin like GTPase 2	−1.475	1.38 × 10^−4^	Unknown
ENSG00000143850	PLEKHA6	pleckstrin homology domain containing A6	−1.190	4.27 × 10^−3^	Unknown
ENSG00000230487	PSMG3-AS1	PSMG3 antisense RNA 1 (head to head)	−1.008	2.36 × 10^−3^	Unknown
ENSG00000203727	SAMD5	sterile alpha motif domain containing 5	−2.663	2.06 × 10^−14^	Unknown
ENSG00000111907	TPD52L1	tumor protein D52 like 1	−1.743	1.79 × 10^−2^	Unknown
ENSG00000183801	OLFML1	olfactomedin like 1	−2.388	6.30 × 10^−18^	Unknown
ENSG00000116774	OLFML3	olfactomedin like 3	−1.813	3.10 × 10^−27^	Unknown
ENSG00000103145	HCFC1R1	host cell factor C1 regulator 1	−1.254	9.23 × 10^−4^	Unknown
ENSG00000156042	CFAP70	cilia and flagella associated protein 70	−1.335	4.93 × 10^−2^	Unknown
ENSG00000147642	SYBU	syntabulin	−1.633	2.50 × 10^−2^	Unknown
ENSG00000165507	C10orf10	chromosome 10 open reading frame 10	−1.072	1.75 × 10^−6^	Unknown
ENSG00000110002	VWA5A	von Willebrand factor A domain containing 5A	−1.182	3.26 × 10^−7^	Unknown
ENSG00000082497	SERTAD4	SERTA domain containing 4	−1.686	8.66 × 10^−9^	Unknown
ENSG00000130518	KIAA1683	KIAA1683	−1.007	1.23 × 10^−2^	Unknown
ENSG00000152217	SETBP1	SET binding protein 1	−1.253	7.10 × 10^−4^	Unknown
ENSG00000198846	TOX	thymocyte selection associated high mobility group box	−1.792	1.29 × 10^−4^	Unknown
ENSG00000139714	MORN3	MORN repeat containing 3	−1.540	4.27 × 10^−2^	Unknown
ENSG00000164309	CMYA5	cardiomyopathy associated 5	−1.344	3.12 × 10^−2^	Unknown
ENSG00000072952	MRVI1	murine retrovirus integration site 1 homolog	−1.251	7.52 × 10^−5^	Unknown
ENSG00000204789	ZNF204P	zinc finger protein 204, pseudogene	−2.032	4.81 × 10^−3^	Unknown
ENSG00000116667	C1orf21	chromosome 1 open reading frame 21	−1.219	3.04 × 10^−12^	Unknown
ENSG00000120262	CCDC170	coiled-coil domain containing 170	−1.921	8.42 × 10^−9^	Unknown
ENSG00000204396	VWA7	von Willebrand factor A domain containing 7	−2.105	1.73 × 10^−2^	Unknown

Notes: FC, fold change. The differential expressed genes (DEGs) in the H_2_O_2_-induced oxidative damage study were screened using the criteria of |log_2_FC| > 2 and *P*-adjust < 0.05. The DEGs were functionally annotated using Blast2go (Version 2.5) and goatools (Version 0.6.5) for gene ontology (GO) annotation and clustering analysis.

**Table 5 foods-12-00806-t005:** Contents of total and reduction sugars and phenolic compounds analyzed by spectrophotometric methods by RFB, HBFB, OFB.

	Samples	RFB	HBFB	OFB
Matter Content (%)				
Total sugar		43.36 ± 0.14	12.16 ± 0.11	9.69 ± 0.17
Reducing sugar		3.18 ± 0.01	0.91 ± 0.03	0.12 ± 0.02
*β*-glucan		0.78 ± 0.05	0.84 ± 0.08	2.74 ± 0.04
Flavonoids		1.11 ± 0.01	1.31 ± 0.06	1.17 ± 0.01
Total phenol		1.46 ± 0.09	1.84 ± 0.05	1.75 ± 0.03

**Table 6 foods-12-00806-t006:** The substance content of RFB, HBFB, OFB.

	Samples	RFB	HBFB	OFB
Matter Content (%)				
Fat		0.6	1.1	0.9
Protein proportion		7.83	14.9	27.3
Ash		0.35	4.0	4.6
Moisture		7.3	6.71	7.95
Carbohydrate		83.9	73.3	59.2

## Data Availability

The data are available from the corresponding author upon reasonable request.

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
