# Peer review of "Role of PI3K-AKT Pathway in Ultraviolet Ray and Hydrogen Peroxide-Induced Oxidative Damage and Its Repair by Grain Ferments"

_foods, 2023, doi:10.3390/foods12040806_

Round 1
Reviewer 1 Report
The manuscript entitled “Role of PI3K-AKT pathway in UVA and H2O2-induced oxidative damages and the verification by grain ferments” by Cheng et al., studied three different fermented grain products (RFB, HBFB and OFB) and evaluated their anti-oxidative properties. The authors found out that these products alleviate the oxidative damage mechanism by UVA and H2O2 and operate through the PI3K-AKT pathway.
Specific Concerns:
1. How do the authors measure the reactive oxygen species (ROS) levels?
2. Do reactive nitrogen species (RNS) have any contribution to the oxidative damage?
3. What is the extent of DNA damage associated with UVA and hydrogen peroxide?
4. Does UVA and hydrogen peroxide activate any DNA damage repair pathways?
5. Figure 1: Axis labels, data labels and legends must be clearly represented. There is a repetition of the same text in figure legend.
6. Table 3 and 4 is missing.
7. Figure 2 and 3: Axis labels and data labels should be clearly represented.
8. Figure 4: Check figure numbering.
9. Specify the modifications that the authors have mentioned for RFB, HBFB and OFB with respect to the published literature?
10. Why the concentrations mentioned for further studies are different in figure 4 and in the text?
11. Figure 5: Axis labels and data labels should be clearly represented.
12. How are the reported products better compared to the available alternatives?
13. Elaborate on the relationship between ECM and PI3K.
14. How do RFB, HBFB and OFB affect skin chemistry such as proteins, lipids, proteoglycans, and collagen etc.
15. Figure 6: Figure labels should be clearly represented.
16. The limitations of the study should be discussed in detail.
17. The significance of the present study mentioning the advantages compared to the reported alternatives should be elaborated.
18. Does UVA and hydrogen peroxide mediate oxidative stress that can lead to cancer?
19. Do the reported pathway further active mTOR?
20. Is there any role of PTEN in this mechanism?
Author Response
Detailed response to reviewers’ comments
January. 26, 2023
Manuscript ID: foods-2123019
Type of manuscript: Article
The revised title: Role of PI3K-AKT pathway in UVA and H2O2-induced oxidative damages and the verification by grain ferments
Authors: Wenjing Cheng, Xiuqin Shi, Jiachan Zhang *, Luyao Li, Fei qian Di, Meng Li, Changtao Wang, Quan An and Dan Zhao
Dear Editor and referees,
First of all, we are very grateful to the journal and reviewers for giving us this opportunity to revise the manuscript. We also thank the editors for their work on our manuscript. The comments and suggestions are very helpful. Detailed point-by-point responses to the comments are provided in the following pages. Note that the comments are presented in Italics, and our responses are in Roman and blue font. In addition, we addressed all these major points and other issues carefully and revised the manuscript accordingly. Besides, we highlighted the revised parts in the article in red.
Please let me know if you have any further questions.
Sincerely,
Jiachan Zhang
Beijing Key Lab of Plant Resource Research and Development, Beijing Technology and Business University, Fucheng Road, Beijing 100048, China
Tel.: +86-13426258535
E-mail: xiao-chan8787@163.com
To the referee’s comments, we make the following responses and changes in the manuscript:
Referee #1:
- How do the authors measure the reactive oxygen species (ROS) levels?
Reply:
ROS levels are measured by DCFDA staining. A DCFDA cellular ROS detection assay can be used following the instructions.
In our previous researches [1,2,3], intracellular ROS levels were measured using ROS detection kits produced by Nanjing Jiancheng Institute of Bioengineering or Beyotime Biotechnology. In brief, after incubation with DCFH-DA fluorescent probe for 20 min, the medium was changed, and the fluorescence intensity of ROS in HSF cells was observed under an inverted fluorescence microscope.
References
[1] Su, Y.; Zhang, Y.; Fu, H.; Yao, F.; Liu, P.; Mo, Q.; Wang, D.; Zhao, D.; Wang, C.; Li, M. Physicochemical and Anti-UVB-Induced Skin Inflammatory Properties of Lacticaseibacillus paracasei Subsp. paracasei SS-01 Strain Exopolysaccharide. Fermentation 2022, 8, 198.
[2] Fu, H.; Zhang, Y.; An, Q.; Wang, D.; You, S.; Zhao, D.; Zhang, J.; Wang, C.; Li, M. Anti-Photoaging Effect of Rhodiola rosea Fermented by Lactobacillus plantarum on UVA-Damaged Fibroblasts. Nutrients 2022, 14, 2324.
[3] Sun, Q.; Fang, J.; Wang, Z.; Song, Z.; Geng, J.; Wang, D.; Wang, C.; Li, M. Two Laminaria japonica Fermentation Broths Alleviate Oxidative Stress and Inflammatory Response Caused by UVB Damage: Photoprotective and Reparative Effects. Mar. Drugs 2022, 20, 650.
- Do reactive nitrogen species (RNS) have any contribution to the oxidative damage?
Reply:
Reactive nitrogen species (RNS) are another group of important chemically reactive species, which can damage cells via nitrosative stress. RNS include nitric oxide (NO), nitroxyl (HNO), nitrogen dioxide (NO2) and peroxynitrite (ONOO-) [1].
Oxidative stress occurs from the imbalance between reactive oxygen and nitrogen species production and these antioxidant defenses [2]. The levels of ROS and RNS such as O2•-, H2O2, HO•, NO, and ONOO- increase in cells under oxidative stress [1,3]. These reactive species can readily oxidize many cellular components, while oxidation of DNA can be most profound in the long term [4,5]. Oxidized DNA bases can result in mutations to the genome in the absence of faithful DNA repair that are passed on to future generations [6,7].
References
[1] Dedon P. C.; Tannenbaum S. R. Arch. Reactive nitrogen species in the chemical biology of inflammation. Biochem. Biophys.2004, 423, 12–22. 10.1016/j.abb.2003.12.017.
[2] Liguori, I.; Russo, G.; Curcio, F.; Bulli, G.; Aran, L.; Della-Morte, D.; Gargiulo, G.; Testa, G.; Cacciatore, F.; Bonaduce, D.; et al. Oxidative Stress, Aging, and Diseases. Clin Interv Aging. 2018, 13, 757–772.
[3] Wiseman H.; Halliwell B. Damage to DNA by reactive oxygen and nitrogen species: role in inflammatory disease and progression to cancer. Biochem. J. 1996, 313, 17–29. 10.1042/bj3130017.
[4] Cadet J.; Wagner J. R.; Shafirovich V.; Geacintov N. E. One-electron oxidation reactions of purine and pyrimidine bases in cellular DNA Int. J. Radiat. Biol. 2014, 90, 423–432. 10.3109/09553002.2013.877176.
[5] Fleming A. M.; Burrows C. J. Formation and processing of DNA damage substrates for the hNEIL enzymes. Free Radical Biol. Med. 2017, 107, 35–52. 10.1016/j.freeradbiomed.2016.11.030.
[6] Pfeifer G.; Besaratinia A. Mutational spectra of human cancer. Hum. Genet. 2009, 125, 493–506. 10.1007/s00439-009-0657-2.
[7] David S. S.; O’Shea V. L.; Base-excision repair of oxidative DNA damage. Kundu S. Nature 2007, 447, 941–950. 10.1038/nature05978.
- What is the extent of DNA damage associated with UVA and hydrogen peroxide?
Reply: Thank you for your comments. Both DNA and H2O2 can induce ROS levels and oxidative damage.
Sun’s UV radiation consisting of 95% UVA (320–400 nm) and 5% UVB (290–320 nm), is a major environmental risk factor for skin cancer [1,2]. UVB is directly absorbed by DNA which causes molecular rearrangements forming the specific photoproducts such as cyclobutane dimers and 6–4 photoproducts. Mutations and cancer can result from many of these modifications to DNA [3]. Differently, UVA is efficient at generating ROS that can damage DNA via indirect photosensitizing reactions [3]. It is a potent driver of oxidative free radical damage to DNA and other macromolecules [4]. UVA causes mutations by generating reactive oxygen species (ROS) such as superoxide anion, hydrogen peroxide and the hydroxyl radical [5,6]. Nucleotides are highly susceptible to free radical injury. Oxidation of nucleotide bases promotes mispairing outside of normal Watson-Crick parameters, causing mutagenesis [7]. The transversion guanine→thymine, for example, is a well-characterized mutation caused by ROS by oxidizing guanine at the 8th position to produce 8-hydroxy-2′-deoxyguanine (8-OHdG) [8]. 8-OHdG tends to pair with an adenine instead of cytosine and therefore this oxidative change mutates a G/C pair into an A/T pair. Such mutations can be found in tumors isolated from the skin, suggesting that oxidative injury can be carcinogenic [9].
Hydrogen peroxide is one type of ROS. It can be obtained by the dismutation of the superoxide anion (O2•-). H2O2 can even mimic the G1 to S phase transition induced by the exposure to growth factors [10]. However, high concentrations of H2O2 activate cell death through the activation of peroxidation reactions and come into equilibrium with Bcl2, an antiapoptotic member of Bcl family, which exerts antioxidant activity [11].
In our study, the decrease in the expression levels of COL1A1, COL1A2, COL4A5, FN1, IGF2, and PIK3R1 was detected in the damaged cells induced by UVA/H2O2, indicating a similar change between them.
References
[1]. Ravanat JL, Douki T, Cadet J. Direct and indirect effects of UV radiation on DNA and its components. J Photochem Photobiol B. 2001;63:88–102.
[2] de Gruijl FR. Photocarcinogenesis: UVA vs UVB. Methods Enzymol. 2000;319:359–366.
[3]D'Orazio J., Jarrett S., Amaro-Ortiz A., Scott T. UV radiation and the skin. International Journal of Molecular Sciences. 2013;14(6):12222–12248. doi: 10.3390/ijms140612222.
[4] Tyrrell R.M. Ultraviolet radiation and free radical damage to skin. Biochem. Soc. Symp. 1995;61:47–53.
[5] Meyskens F.L., Jr, Farmer P., Fruehauf J.P. Redox regulation in human melanocytes and melanoma. Pigment. Cell Res. 2001;14:148–154.
[6] Jaszewska, E.; Soin, M.; Filipek, A.; Naruszewicz, M. UVA-induced ROS generation inhibition by Oenothera paradoxa defatted seeds extract and subsequent cell death in human dermal fibroblasts. J. Photochem. Photobiol. B Biol. 2013, 126, 42–46.
[7] Schulz I., Mahler H.C., Boiteux S., Epe B. Oxidative DNA base damage induced by singlet oxygen and photosensitization: Recognition by repair endonucleases and mutagenicity. Mutat. Res. 2000;461:145–156.
[8] Nishimura S. Involvement of mammalian OGG1(MMH) in excision of the 8-hydroxyguanine residue in DNA. Free Radic. Biol. Med. 2002;32:813–821.
[9] Agar N.S., Halliday G.M., Barnetson R.S., Ananthaswamy H.N., Wheeler M., Jones A.M. The basal layer in human squamous tumors harbors more UVA than UVB fingerprInt. mutations: A role for UVA in human skin carcinogenesis. Proc. Natl. Acad. Sci. USA. 2004;101:4954–4959.
[10] Burdon R. H. Superoxide and hydrogen peroxide in relation to mammalian cell proliferation. Free Radical Biology and Medicine. 1995;18(4):775–794. doi: 10.1016/0891-5849(94)00198-S.
[11] Hockenbery D. M., Oltvai Z. N., Yin X.-M., Milliman C. L., Korsmeyer S. J. Bcl-2 functions in an antioxidant pathway to prevent apoptosis. Cell. 1993;75(2):241–251. doi: 10.1016/0092-8674(93)80066-N.
- Does UVA and hydrogen peroxide activate any DNA damage repair pathways?
Reply:
Cellular DNA is under constant insult by various endogenous and exogenous factors, such as ROS from cellular respiration and UV radiation from the sun [1].To combat the resultant DNA damage, eukaryotic cells employ a number of different DNA damage repair pathways.
These pathways include base excision repair (BER), nucleotide excision repair (NER), interstrand crosslink repair, mismatch repair, ribonucleotide removal pathway, direct damage reversal and several double strand break repair pathways. Among them, NER and BER are most likely to mend PAH-induced DNA damage, since the NER pathway repairs bulky DNA adducts and the BER pathway repairs oxidative DNA damage [2-4].
In the study, DEGs were screened between Model and Control. In both of the oxidative damage studies, there were plenty of genes involved in the response to external stimulus, oxidative stress response, etc.
References
[1] Barnes D.E., Lindahl T.. Repair and genetic consequences of endogenous DNA base damage in mammalian cells. Annu. Rev. Genet. 2004; 38:445–476.
[2] Braithwaite E, Wu X, Wang Z. Repair of DNA lesions induced by polycyclic aromatic hydrocarbons in human cell-free extracts: involvement of two excision repair mechanisms in vitro. Carcinogenesis. 1998;19:1239–1246.
[3] Goode EL, Ulrich CM, Potter JD. Polymorphisms in DNA repair genes and associations with cancer risk. Cancer Epidemiol Biomarkers Prev. 2002;11:1513–1530.
[4] Robertson AB, Klungland A, Rognes T, Leiros DNA repair in mammalian cells: base excision repair, the long and short of it. Cell Mol Life Set. 2009;66:981–993.
- Figure 1: Axis labels, data labels and legends must be clearly represented. There is a repetition of the same text in figure legend.
- Table 3 and 4 is missing.
- Figure 2 and 3: Axis labels and data labels should be clearly represented.
- Figure 4: Check figure numbering.
Reply to 5 to 8:
Thank you for your comments and great suggestions. we have checked and revised these parts in the manuscript. Thank you again for your suggestions, which are very important for us to improve this manuscript.
- Specify the modifications that the authors have mentioned for RFB, HBFB and OFB with respect to the published literature?
Reply:
The fermentation method of RFB, HBFB and OFB was following Ref.16,26 and 27, with a little modifications. The differences have been added in the manuscript in red. Thank you again for your suggestions, which are very important for us to improve this manuscript.
- Why the concentrations mentioned for further studies are different in figure 4 and in the text?
Reply: Thank you for your comments. We have checked and revised this part. Sorry for the mistakes.
- Figure 5: Axis labels and data labels should be clearly represented.
Reply: Thank you for your comments and great suggestions. We have checked and revised Figure 5 in the manuscript. Some detailed information has also been added in the legend.
- How are the reported products better compared to the available alternatives?
Reply: Thank you for your comments.
In the study, three of the ferments are used to verify whether they follow the findings by RNA-Seq or not. We hope to obtain a proper screening method to find out the potential actives which have anti-oxidant abilities. While, we cannot know the order of antioxidant strength of these actives. Therefore, it is imperative that efforts continue to further improve outcomes.
- Elaborate on the relationship between ECM and PI3K.
Reply: Thank you for your comments.
ECMs are important components of human tissue, and play a very important role in maintaining cell morphology, structure, and function. The PI3K-AKT signaling pathway is a survival pathway that has recently become known as a key regulator of cell proliferation, migration and apoptosis [1,2]. The PI3K/AKT pathway also plays a key role in regulating ECM synthesis [3]. What’s more, Integrin-mediated attachment to ECM activates pro-survival signaling pathways, such as PI3K–AKT cascade [4].
In both UVA and H2O2 injury models, we found that the decline in ECM was a common factor for the activation of PI3K. This indicated that both photo-damage and oxidative damage can interrupt the secretion, synthesis, and normal operation of ECM. The activation of PI3K/AKT pathway is an effective regulatory pathway to restore such damage.
References
[1] Sang H, Li T, Li H, Liu J. Gab1 regulates proliferation and migration through the PI3K/Akt signaling pathway in intrahepatic cholangiocarcinoma. Tumour Biol. 2015;36: 8367–8377. doi: 10.1007/s13277-015-3590-0.;
[2] Zhou H, Li D, Shi C, Xin T, Yang J, Zhou Y, Hu S, Tian F, Wang J, Chen Y. Effects of exendin-4 on bone marrow mesenchymal stem cell proliferation, migration and apoptosis in vitro. Sci Rep. 2015; 5:12898. doi: 10.1038/srep12898.
[3] Zeng R, Xiong Y, Zhu F, Ma Z, Liao W, He Y, He J, Li W, Yang J, Lu Q, et al. Fenofibrate attenuated glucose-induced mesangial cells proliferation and extracellular matrix synthesis via PI3K/AKT and ERK1/2. PLoS One. 2013;8: e76836. doi: 10.1371/journal.pone.0076836.
[4] Mehlen P, Puisieux A. 2006. Metastasis: a question of life or death. Nat. Rev. Cancer 6:449–458
- How do RFB, HBFB and OFB affect skin chemistry such as proteins, lipids, proteoglycans, and collagen etc.
Reply: Thank you for your comments and great suggestions.
According to Figure 4 in the manuscript, RFB, HBFB and OFB showed excellent proliferation effects on the damaged cells caused by UVA and H2O2. The expressions of collagen related genes, such as COL1A1, COL1A2, and COL4A5 were also accelerated (Figure 5), which we usually consider that the protein levels are also increased when genes are accelerated. Besides, polysaccharides in RFB, HBFB and OFB have good effects on skin, such as moisturizing, maintaining skin barrier, antioxidant, etc. However, further studies need to be performed to explore the specific roles and to verify their functions on clinical trials.
Thank you again for your suggestions, which are very important for us to improve this manuscript.
- Figure 6: Figure labels should be clearly represented.
Reply: Thank you for your comments and great suggestions. Figure labels (RFB, HBFB and OFB) are represented in the legend of Figure 6.
- The limitations of the study should be discussed in detail.
Reply: Thank you for your comments and great suggestions.
The limitations of the study are listed in the end of 4. Discussion, and marked in red.
Thank you again for your suggestions, which are very important for us to improve this manuscript.
- The significance of the present study mentioning the advantages compared to the reported alternatives should be elaborated.
Reply: The significance of the present study has been mentioned in last of the 4. Discussion. Thank you for your comments and great suggestions.
- Does UVA and hydrogen peroxide mediate oxidative stress that can lead to cancer?
Reply:
Yes, it does. The damage of UVA and H2O2 will affect the stability of ROS levels in cells, resulting in DNA damage. DNA damage is one of the most serious effects of skin. Excessive exposure to UV radiation plays a major role in the induction of photocarcinogenesis [1]. Nucleotides are highly susceptible to free radical injury. Oxidation of nucleotide bases promotes mispairing outside of normal Watson-Crick parameters, causing mutagenesis [2]. Such mutations can be found in tumors isolated from the skin, suggesting that oxidative injury can be carcinogenic [3].
References
[1] Agar, N.S.; Halliday, G.M.; Barnetson, R.S.C.; Ananthaswamy, H.N.; Wheeler, M.; Jones, A.M. The basal layer in human squamous tumors harbors more UVA than UVB fingerprint mutations: A role for UVA in human skin carcinogenesis. Proc. Natl. Acad. Sci. USA 2004, 101, 4954–4959.
[2] Schulz I., Mahler H.C., Boiteux S., Epe B. Oxidative DNA base damage induced by singlet oxygen and photosensitization: Recognition by repair endonucleases and mutagenicity. Mutat. Res. 2000; 461:145–156.
[3] Agar N.S., Halliday G.M., Barnetson R.S., Ananthaswamy H.N., Wheeler M., Jones A.M. The basal layer in human squamous tumors harbors more UVA than UVB fingerprInt. mutations: A role for UVA in human skin carcinogenesis. Proc. Natl. Acad. Sci. USA. 2004;101: 4954–4959.
- Do the reported pathway further active mTOR?
Reply: Thank you for your comments and great suggestions.
In this study, the PI3K-AKT pathway was selected to discuss both in UVA and in H2O2 induced damage studies. Figure 3 showed the expression changes of the genes involved in this signaling pathway induced by UVA (Figure 3A) and H2O2(Figure 3B). Although the screened DEGs were not involved in the regulation of mTOR pathway directly, the key genes of PI3K-AKT pathway have the possibilities to accelerate mTOR signaling pathway. As shown in Figure 3A, the increased LKB1 and REDD1 induced by UVA (Figure 3A) and the inhibited AMPK induced by H2O2(Figure 3B) could further activate mTOR.
- Is there any role of PTEN in this mechanism?
Reply: Thank you for your comments and great suggestions.
PTEN is the most important negative regulator of the PI3K signaling pathway. The PTEN network regulates a broad spectrum of biological functions, modulating the flow of information from membrane bound growth factor receptors to nuclear transcription factors, occurring in concert with other tumor suppressors and oncogenic signaling pathways [1-4]. However, no influence on PTEN expression was found in the study. The screening criteria of DEGs may play roles in the result.
References
[1] Lee, Y.R.; Chen, M.; Pandolfi, P.P. The Functions and Regulation of the PTEN Tumour Suppressor: New Modes and Prospects. Nat. Rev. Mol. Cell Bio. 2018, 19, 547–562.
[2] Song, M.S.; Salmena, L.; Pandolfi, P.P. The Functions and Regulation of the PTEN Tumour Suppressor. Nat. Rev. Mol. Cell Bio. 2012, 13, 283–296.
[3] Milella, M.; Falcone, I.; Conciatori, F.; Incani, U.C.; Del Curatolo, A.; Inzerilli, N.; Nuzzo, C.M.A.; Vaccaro, V.; Vari, S.; Cognetti, F.; et al. PTEN: Multiple Functions in Human Malignant Tumors. Front. Oncol. 2015, 5.
[4] Wang, L.; Wang, L.; Shi, X.; Xu, S. Chlorpyrifos Induces the Apoptosis and Necroptosis of L8824 Cells through the ROS/PTEN/PI3K/AKT Axis. J. Hazard Mater. 2020, 398.
Reviewer 2 Report
The information described in the manuscript is interesting and novel, however, it is necessary to attend to the comments attached (recommendations and questions) in the pdf file

Author Response
Detailed response to reviewers’ comments
January. 26, 2023
Manuscript ID: foods-2123019
Type of manuscript: Article
The revised title: Role of PI3K-AKT pathway in UVA and H2O2-induced oxidative damages and the verification by grain ferments
Authors: Wenjing Cheng, Xiuqin Shi, Jiachan Zhang *, Luyao Li, Fei qian Di, Meng Li, Changtao Wang, Quan An and Dan Zhao
Dear Editor and referees,
First of all, we are very grateful to the journal and reviewers for giving us this opportunity to revise the manuscript. We also thank the editors for their work on our manuscript. The comments and suggestions are very helpful. Detailed point-by-point responses to the comments are provided in the following pages. Note that the comments are presented in Italics, and our responses are in Roman and blue font. In addition, we addressed all these major points and other issues carefully and revised the manuscript accordingly. Besides, we highlighted the revised parts in the article in red.
Please let me know if you have any further questions.
Sincerely,
Jiachan Zhang
Beijing Key Lab of Plant Resource Research and Development, Beijing Technology and Business University, Fucheng Road, Beijing 100048, China
Tel.: +86-13426258535
E-mail: xiao-chan8787@163.com
Referee #:
Lines 8-10; 13,14: add authors initials in each parenthesis.
Line 9: delete space…. Wang) ;
Line 12,15: remove semicolon at end of line.
Line 14: modify….+86-
Line 17: The abstract exceeds the total number of words (200) indicated in the authors' guide.
Line 20: modify…. In our study, RNA-seq…
Line 28: modify…. Results indicate that DEGs were…
Line 33: modify…. S. commune-grain ferments…
Line 36,39: RNA-Seq or RNA-seq like in line 21; it is necessary to homogenize terms throughout the document.
Line 36,37: delete space between paragraphs.
Line 42: change Antioxidant by antioxidant, Grain by grain
Line 45: mention ROS examples in the text
Line 48: …damage [references missed?]
Line 48,49: Edit format between both paragraphs (No Spacing), apparently, they are separated. See example of lines 65 and 66. Review format throughout the document
Line 51: change [2-5] by [2–5]
Line 54: change [7-10] by [7–10]
Line 55-59: [references missed?]
Line 73: modify…. In our study,
Line 79: modify [16 by [16]
Line 73-77: paragraph text formatting appears left-aligned, needs to be corrected through the manuscript.
Line 88: change [20,22-24] by [20,22–24]
Line 95: modify…. results, in a previous study,
Line 101: change 2. Experimental by 2. Materials and Methods
Line 106: insert space 1×105 U/L
Line 112: If an incubator was used, it is necessary to indicate the model, brand, and country.
Line 113: insert space 37 °C
Line 115: change [27-29] by [27–29]
Line 118: modify…. 0 to 5 mg/mL; 0 to 2.5 mg/mL; 0 to 1.25 mg/mL
Line 118: modify…. 24 h
Line 119: modify…. 4 h
Line 120: change…. Biorigin, Beijing, Inc
Line 121: indicate the model, brand and country of the equipment.
Line 126: modify…. 1×106
Line 127: insert space 0.01 M
Line 128: insert space 365 nm
Line 130: insert the country of the equipment.
Line 131,134: modify…. 2,000 µmol/L
Line 132: change 0.5 h by 30 minutes.
Line 133: use the abbreviated form of hydrogen peroxide like in the abstract section.
Line 134: change 0.5 h by 30 min.
Line 142: insert space 280 = 1.8
Line 143: insert space 230 ≥ 2.0
Line 143: insert space RIN ≥ 6.5
Line 143: insert space 8S ≥ 1.0
Line 143: insert space > 2 µg
Line 153: insert space > 98.5%
Line 157: insert space > 98.9%
Line 170: insert space | > 1.2
Line 170,172: insert space P-adjust < 0.05
Line 171: insert space | > 2.0
Line 172: insert space Version 2.5
Line 174: did you mean PB or BP?
Line 191: insert space 94 °C
Line 192: insert space 60 °C
Line 192,193: insert space 72 °C
Line 200: insert space 9.0 (GraphPad
Line 201: insert space < 0.05
Line 201: insert space < 0.01
Line 203: In the information contained in figure 1, section A, insert a space in the figure caption.... Intensity (J
Line 203: In the information contained in figure 1, section B, insert space in the figure caption.... Concentration (umol/L)
Line 203: If possible, it is necessary to improve the dpi resolution of the graphic so that the figure is not distorted when increasing the zoom.
Line 203: the text included in figure 1, section a, b, c, and d, appears distorted.
Line 206: repeated text appears.
Line 207: insert space 15 J/cm
Line 207: insert space H2O2 (1,000
Line 207: modify…. 1,000 µmol/L for 30 min
Line 208: modify….
Line 218: use the abbreviated form for hydrogen peroxide.
Line 219: modify…. 1,000 µmol/L
Line 219: change 0.5 h by 30 min
Line 223: insert space | > 1.2
Line 224,238: insert space P-adjust < 0.05
Line 237: insert space | > 2
Line 258: insert space score = 3.11
Line 259: insert space score = 3.09
Line 263: if possible, it is necessary to modify the resolution of the graph. The text contained in the figure appears distorted.
Line 293: figure 2 appears before being mentioned in the text.
Line 293: modify (Figure 2).
Line 303: you cannot include references in the results section, unless both sections are in a single section.
Line 316: if possible, it is necessary to modify the resolution of the figure 3. The text contained in the figure appears distorted.
Line 322: H2O2 induced or H2O2-induced; it is necessary to homogenize terms throughout the document.
Line 328-335: you cannot include references in the results section, unless both sections are in a single section.
Line 335: P-value
Line 376: insert point… survival. Furthermore
Line 388: use two values after point, xx ± xx
Line 388: change total phenol by Total phenol
Line 388: delete space…. β-glucan
Line 393: Regarding the following statement...There are 6 replicates in each group, and the OD value of each well is measured with a microplate reader at a wavelength of 450 nm.... the information should appear in the materials and methods section. Modify the footer of tables and figures where it appears.
Line 395,429: Regarding the following statement... Statistical significance was determined by ANOVA test..... the information should appear in the materials and methods section. Modify the footer of tables and figures where it appears.
Line 395,396,429: insert space p < 0.01 (it is necessary to homogenize terms throughout the manuscript).
Line 396: insert space p < 0.05
Line 397-402: you cannot include references in the results section, unless both sections are in a single section.
Line 403-405: information to be included in the materials and methods section
Line 409: modify…. β-glucan
Line 411: HSFs or HSF; it is necessary to homogenize terms throughout the document.
Line 413: insert space 69.55 ± 2.30%
Line 413: insert space 1.25 mg/mL
Line 414: insert space 57.83 ± 3.81%.
Line 422: check the font size <
Line 429: insert space H2O2 (1000
Line 429: modify…. 1,000 µmol/L for 30 min
Line 446-450: references missed?
Line 451-455: references missed?
Line 446: mention examples of which oxidative molecules
Line 447: mention examples of which antioxidants are depleted
Line 468: change [42-46] by [42–46]
Line 505,520: cite the correct form of the reference in the text
Line 529: ….that HSFs upon…
Line 532: Dan et al. or Hao et al.? cite the correct form of the reference in the text.
Line 566: mention examples of the type of phenolic acids and polysaccharides reported for this fungus.
Line 574: [52-56] by [52–56]
Line 580: delete space…. with the
Line 616: no formatting spaces should appear between lines in the references section.
Line 618: …. Circ. Res. 1999, 84,
Line 618: 113-121 or 113–121
Line 619,623,625,627,629, etc: The title must appear in lowercase text format. Correct through references.
Line 620,651,etc: Scientific names must appear in italic text format. Correct through references.
Line 254: Reference volume must appear in italic text format. Correct through references.
Line 636: Can. J. Physiol. Pharmacol.
Line 639: Korean. Circ. J.
Line 641,644: the journal of the references must appear in abbreviation format. Check through references.
Line 654,656,660,662,664,667,669,673,681,683,692,etc: fix cited reference journal abbreviation format
Note: it is necessary to check the correct format for references section
Reply to the above comments:
Sorry about the mistakes. The format problems have been modified in the manuscript. The lost information has also been added. Figures and tables are revised too. All the changes are marked in red. Thank you again for your suggestions, which are very important for us to improve this manuscript.
Line 121: a microscope was used to measure the wavelength of 450 nm?
Reply: Thank you for your comments and great suggestions. Due to our negligence, the microplate reader was written as a microscope. The optical densities of the solutions were quantified at a wavelength of 450 nm using a microplate reader. We have checked and revised in the manuscript. Thank you again.
Line 388: the sum of the components in each column of the table is greater than 100%, are two sections Total sugar to total phenol and Fat to Carbohydrate?
Reply: Thank you for your comments and great suggestions. Because of our wrong representation, the table content is not clearly displayed. The table should be divided into two parts, we have split into two tables. Sugars and phenols were analyzed separately in Table 5. The total amount of fat, protein, ash, moisture and carbohydrate was 100% Table 6. We have checked and revised in the manuscript. Thank you again for your suggestions, which are very important for us to improve this manuscript.
Line 467,468: Do these materials only have antioxidant power, or do they act as pro-oxidants under certain conditions?
Reply: Thank you for your comments and great suggestions. In this study, RFB,HBFB and OFB were schizophylla fermentation broth, which contained rich polysaccharides. According to our previous research methods [1-7], polysaccharides, microbial fermentation liquid and other substances can play an antioxidant role by scavenging free radicals and increasing MDA content in damaged cells. Coix lachryma-jobi L. Seed Lactobacillus reuteri fermentation broth, Rhodiola rosea fermented by Lactobacillus plantarum, Lacticaseibacillus paracasei Subsp. paracasei SS-01 strain exopolysaccharide, G. Lucidum polysaccharide exert their antioxidant capacity by increasing SOD activity and decreasing ROS content by MDA activity. However, as to whether they act as pro-oxidants under certain conditions, we have not discussed this aspect. In this study, the antioxidant effects of RFB, HBFB and OFB were found through data. Thank you again for your suggestions, which are very important for us to improve this manuscript.
[1] Su, Y.; Zhang, Y.; Fu, H.; Yao, F.; Liu, P.; Mo, Q.; Wang, D.; Zhao, D.; Wang, C.; Li, M. Physicochemical and Anti-UVB-Induced Skin Inflammatory Properties of Lacticaseibacillus paracasei Subsp. paracasei SS-01 Strain Exopolysaccharide. Fermentation 2022, 8, 198.
[2] Fu, H.; Zhang, Y.; An, Q.; Wang, D.; You, S.; Zhao, D.; Zhang, J.; Wang, C.; Li, M. Anti-Photoaging Effect of Rhodiola rosea Fermented by Lactobacillus plantarum on UVA-Damaged Fibroblasts. Nutrients 2022, 14, 2324.
[3] Sun, Q.; Fang, J.; Wang, Z.; Song, Z.; Geng, J.; Wang, D.; Wang, C.; Li, M. Two Laminaria japonica Fermentation Broths Alleviate Oxidative Stress and Inflammatory Response Caused by UVB Damage: Photoprotective and Reparative Effects. Mar. Drugs 2022, 20, 650.
[4] Shi, X.; Cheng, W.; Wang, Q.; Zhang, J.; Wang, C.; Li, M.; Zhao, D.; Wang, D.; An, Q. Exploring the Protective and Reparative Mechanisms of G. Lucidum Polysaccharides Against H2O2-Induced Oxidative Stress in Human Skin Fibroblasts. CLINICAL COSMETIC AND INVESTIGATIONAL DERMATOLOGY 2021, 14, 1481–1496, doi:10.2147/CCID.S334527.
[5] Zhao, J.; Fu, H.; Zhang, Y.; Li, M.; Wang, D.; Zhao, D.; Zhang, J.; Wang, C. Protective Effects of Lactobacillus Reuteri SJ-47 Strain Exopolysaccharides on Human Skin Fibroblasts Damaged by UVA Radiation. BIORESOURCES AND BIOPROCESSING 2022, 9, doi:10.1186/s40643-022-00617-0.
[6] Zhang, Y.; Fu, H.; Zhang, Y.; Wang, D.; Zhao, D.; Zhang, J.; Li, M.; Wang, C. Taraxasterol Repairs UVB-Induced Skin Barrier Injury through MAPK/NF-Kappa B Signaling Pathways. FOOD AND AGRICULTURAL IMMUNOLOGY 2022, 33, 604–616, doi:10.1080/09540105.2022.2107619.
[7] Fang, J.; You, S.; Sun, Q.; Wang, Z.; Wang, C.; Wang, D.; Li, M. Protection Impacts of Coix Lachryma-Jobi L. Seed Lactobacillus Reuteri Fermentation Broth on Hydrogen Peroxide-Induced Oxidative Stress in Human Skin Fibroblasts. APPLIED SCIENCES-BASEL 2023, 13, doi:10.3390/app13010540.

Round 2
Reviewer 1 Report
The manuscript entitled “Role of PI3K-AKT pathway in UVA and H2O2-induced oxidative damages and the verification by grain ferments” by Cheng et al., had been revised thoroughly. Most of the queries were addressed in the new version which provides relevant information to the readers. There were still a few grammatical problems and typographical errors. Some of them are as follows,
1. Abstract: Line 24.
2. Line 27-29: Grammatical error.
Author Response
Detailed response to reviewers’ comments
February. 4, 2023
Manuscript ID: foods-2123019
Type of manuscript: Article
The revised title: Role of PI3K-AKT pathway in ultraviolet ray and hydrogen peroxide-induced oxidative damage and its repair by grain ferments
Authors: Wenjing Cheng, Xiuqin Shi, Jiachan Zhang *, Luyao Li, Fei qian Di, Meng Li, Changtao Wang, Quan An and Dan Zhao
Dear Editor and referees,
We would like to thank the journals and reviewers for suggesting revisions to our manuscript again. We also thank the editors for their work on our manuscript. These comments and suggestions are very useful. The following pages provide detailed, point-by-point responses to these comments. Note that the comments are presented in Italics, and our responses are in Roman and blue font. In addition, we addressed all these major points and other issues carefully and revised the manuscript accordingly. Besides, we highlighted the revised parts in the article in red.
Please let me know if you have any further questions.
Sincerely,
Jiachan Zhang
Beijing Key Lab of Plant Resource Research and Development, Beijing Technology and Business University, Fucheng Road, Beijing 100048, China
Tel.: +86-13426258535
E-mail: xiao-chan8787@163.com
To the referee’s and academic editor’s comments, we make the following responses and changes in the manuscript:
Referee #1:
The manuscript entitled “Role of PI3K-AKT pathway in UVA and H2O2-induced oxidative damages and the verification by grain ferments” by Cheng et al., had been revised thoroughly. Most of the queries were addressed in the new version which provides relevant information to the readers. There were still a few grammatical problems and typographical errors. Some of them are as follows,
- Abstract: Line 24.
- Line 27-29: Grammatical error.
Reply 1 to 2:
Thank you for your comments and great suggestions. We have revised the grammatical problems and typographical errors in the manuscript. The differences have been added in the manuscript in red.